# UrbanFeel: A Comprehensive Benchmark for Temporal and Perceptual Understanding of City Scenes through Human Perspective

**Jun He**[1], **Yi Lin**[1], **Zilong Huang**[1], **Jiacong Yin**[1], **Junyan Ye**[1], **Yuchuan Zhou**[1],
**Weijia Li**[1,2†], **Xiang Zhang**[1†]
[1]Sun Yat-sen University  [2]SIGS, Tsinghua University

## Abstract

Urban development impacts over half of the global population, making human-centered understanding of its structural and perceptual changes essential for sustainable development. While Multimodal Large Language Models (MLLMs) have shown remarkable capabilities across various domains, existing benchmarks that explore their performance in urban environments remain limited, lacking systematic exploration of temporal evolution and subjective perception of urban environment that aligns with human perception. To address these limitations, we propose UrbanFeel, a comprehensive benchmark designed to evaluate the performance of MLLMs in urban development understanding and subjective environmental perception. UrbanFeel comprises 14.3K carefully constructed visual questions spanning three cognitively progressive dimensions: Static Scene Perception, Temporal Change Understanding, and Subjective Environmental Perception. We collect multi-temporal single-view and panoramic street-view images from 11 representative cities worldwide, and generate high-quality question-answer pairs through a hybrid pipeline of spatial clustering, rule-based generation, model-assisted prompting, and manual annotation. Through extensive evaluation of 20 state-of-the-art MLLMs, we observe that Gemini-2.5 Pro achieves the best overall performance, with its accuracy approaching human expert levels and narrowing the average gap to just 1.5%. Most models perform well on tasks grounded in scene understanding. In particular, some models even surpass human annotators in pixel-level change detection. However, performance drops notably in tasks requiring temporal reasoning over urban development. Additionally, in the subjective perception dimension, several models reach human-level or even higher consistency in evaluating dimension such as beautiful and safety. Our results suggest that MLLMs are demonstrating rudimentary emotion understanding capabilities. The code and dataset of this work will be released at https://github.com/Hejun0915/UrbanFeel.

## 1 Introduction

With over half of the global population now living in urban areas (World Bank, 2024), understanding the dynamics of urban development has become increasingly critical for designing sustainable governance strategies, guiding urban policy, and promoting human-centric smart cities (Yuan et al., 2024; Van Etten et al., 2021; Zhang et al., 2024b). Compared to satellite imagery, which provides macro-scale, top-down observations, street-view imagery offers fine-grained, street-level perspectives that are more aligned with human visual perception (Biljecki & Ito, 2021; Naik et al., 2017; Wang et al., 2025). This unique characteristic enables it to capture subtle environmental changes within cities, making it a valuable data source for analyzing intra-urban transformation.

Recent research has explored the use of deep learning models in conjunction with street-view imagery to assess urban development stages (Zhang et al., 2018; Alpherts et al., 2025), visual quality (Ito et al., 2024; Benidir et al., 2025), and perceived livability (Dubey et al., 2016; Yang et al.,

---

[†]Corresponding author(s). E-mail(s): {liweij29, zhangx795}@mail.sysu.edu.cn

2024; Li et al., 2025). However, these approaches face challenges in terms of generalization across modalities and cities. More importantly, they struggle to effectively quantify and interpret human subjective perception—an essential component of real-world urban understanding.

The advent of Large Language Models (LLMs) and Multimodal LLMs (MLLMs) has introduced new possibilities for tackling these limitations (Zhang et al., 2024a; Xuan et al., 2025; Ye et al., 2025). By leveraging massive amounts of multimodal pretraining data, MLLMs exhibit strong capabilities in spatial reasoning, visual-linguistic alignment, and commonsense inference. Initial attempts have applied these models to urban imagery tasks, such as vehicle trajectory prediction (Liu et al., 2025; Lai et al., 2025) or scene understanding (Yan et al., 2024; Feng et al., 2025b;a; Huang et al., 2025), and several early benchmarks have emerged to evaluate their performance on objective tasks such as image geolocalization (Zhou et al., 2025) and infrastructure inference (Feng et al., 2025c).

Prior work has largely been confined to static snapshots, focusing on objective recognition tasks such as autonomous driving or urban planning, while overlooking the historical dynamics of cities and thus failing to capture trajectories of development, renewal, and transformation. At the same time, physical changes in the built environment—such as renovation or decay—often reshape human perceptions of safety, beauty, and liveliness. However, existing benchmarks rarely examine how these perceptual shifts are linked to temporal urban evolution, leaving a critical gap in understanding the interaction between physical change and human experience.

To bridge these gaps, we present **UrbanFeel**, a novel human-centric benchmark for evaluating MLLMs in the context of urban change perception. UrbanFeel defines 11 tasks across three dimensions—*static scene perception*, *temporal change understanding*, and *subjective environmental perception*—to assess models' capabilities in recognition, reasoning, and alignment with human perception. Our benchmark emphasizes multi-view integration, temporal-spatial consistency, and perceptual alignment, aiming to push the boundaries of MLLMs toward more human-aligned urban understanding, to help MLLMs provide a reference for continuous monitoring and prediction capabilities in sustainable cities.

Our main contributions are summarized as follows:

- We introduce **UrbanFeel**, a multi-perspective, multi-dimensional benchmark designed to evaluate MLLMs' performance on tasks related to urban development and human perception. UrbanFeel carefully designs 11 subtasks, focusing on evaluating the model's perception and understanding capabilities in three dimensions: Static Scene Perception, Temporal Change Understanding, and Subjective Environmental Perception.
- We design a scalable and interpretable task-querying framework, incorporating a diverse range of evaluation formats including binary classification, multiple-choice, sorting, and open-ended reasoning. To enhance explainability, we introduced manual annotation based on local visual evidence into the benchmark management process.
- We conduct a comprehensive evaluation of 20 state-of-the-art MLLMs on UrbanFeel, quantifying model differences across task categories and revealing that current models still fall short of human-level performance in spatial reasoning and subjective perception within urban change scenarios.

## 2 RELATED WORK

### 2.1 URBAN TEMPORAL CHANGE ASSESSMENT FROM STREET-VIEW IMAGERY

With the accelerating pace of global urbanization, cities have undergone profound spatial and environmental transformations, prompting growing research interest in urban evolution (Pandey & Seto, 2015; Hatab et al., 2019; Follmann et al., 2021). In recent years, street-view imagery has emerged as a valuable data source for urban change detection due to its close alignment with human perspectives (Biljecki & Ito, 2021; Ito et al., 2024). For instance, ChangeScore (Naik et al., 2017) utilized deep networks to correlate visual changes with socioeconomic variables. Subsequent studies focused on quantifying the built environment's physical fabric: Street2Vec measured physical structural shifts via latent space embeddings (Stalder et al., 2024b), while CityPulse constructed semantic label sequences to detect binary environmental changes (Huang et al., 2024b). Similarly,

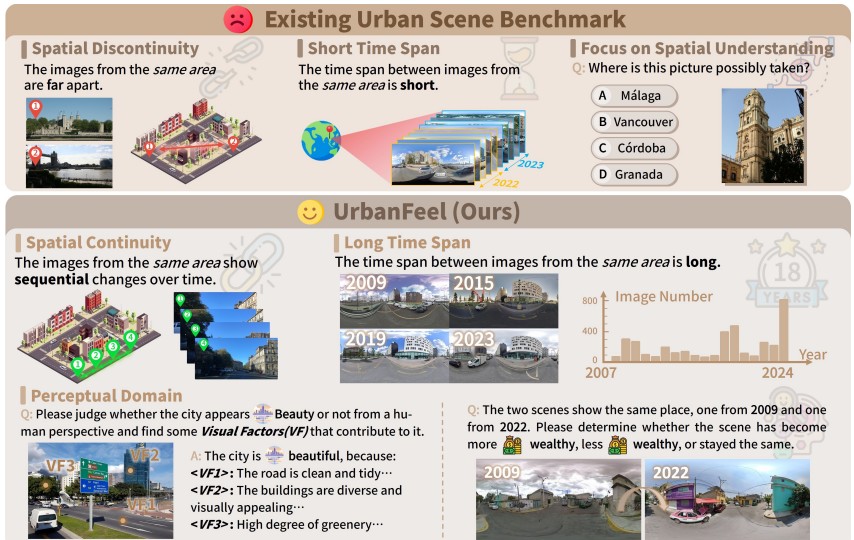

Figure 1: Comparison with existing urban scene benchmarks. UrbanFeel introduces three key innovations: (1) spatially continuous street-view data that includes both single-view and panoramic imagery, (2) long-term temporal coverage spanning over 15 years, and (3) a novel evaluation dimension focused on subjective human perception (e.g., safety, beauty), enabling human-centered assessment beyond conventional spatial understanding.

Stalder et al. (Stalder et al., 2024a) quantified urban decay dynamics using detected objects in different years as a proxy. While establishing street-view time series as high-resolution proxies for physical change, these methods predominantly frame urban evolution as binary classification. Recently, Visual Chronicles (Deng et al., 2025) employed VLMs to mine co-occurring visual trends from open-ended queries. However, existing research remains largely confined to describing objective visual elements change within limited regions, rather than systematically evaluating model capabilities in controlled temporal reasoning tasks across diverse urban contexts.

## 2.2 SUBJECTIVE PERCEPTION ASSESSMENT OF URBAN ENVIRONMENTS

Parallel to physical change detection, extensive research utilizes street-view imagery and deep learning to quantify subjective urban perceptions and their socioeconomic correlations (He et al., 2023; van Veghel et al., 2024; Rui & Xu, 2024; He et al., 2025b;a). The foundational Place Pulse project (Dubey et al., 2016) established this paradigm by crowdsourcing pairwise comparisons to train deep learning models for large-scale prediction. Subsequent studies expanded this frontier: Yao et al. (Yao et al., 2019) improved alignment via adversarial learning, while Wei et al. (Wei et al., 2022) and Fan et al. (Fan et al., 2023) linked perceptual attributes to planning metrics and socioeconomic outcomes. Notably, Wang et al. Wang et al. (2025) applied interpretable machine learning to uncover functional zone-dependent nonlinear associations and threshold effects between environmental features and perception. Crucially, however, these approaches remain confined to static image snapshots and traditional discriminative models. They lack a comprehensive benchmarking framework to evaluate MLLMs' capacity to capture how human subjective perception evolves dynamically alongside long-term physical transformations.

## 2.3 MULTIMODEL LARGE LANGUAGE MODELS

In recent years, Multimodal Large Language Models (MLLMs) such as Qwen (Bai et al., 2023), GPT-4o (OpenAI, 2024), and Gemini-2.5-pro (Comanici et al., 2025) have achieved remarkable progress in image generation (Anonymous, 2025), visual reasoning (Zhang et al., 2025a), and cross-modal alignment (Wu et al., 2024a). Leveraging large-scale pretraining and instruction tuning, these models show strong generalization in open-domain visual understanding (Li et al., 2024b; Wu et al., 2024b), though their performance on domain-specific applications remains limited. In urban contexts, recent studies explored MLLMs' zero-shot spatial reasoning. UrbanCLIP (Yan et al., 2024) aligns imagery with textual semantics via contrastive learning, while UrbanLLaVA (Feng et al.,

2025b) integrates street-view, structured data, and trajectories, achieving strong generalization on UBench. Despite these advances, a systematic framework for evaluating MLLMs on subjective urban perception or urban change assessment remains lacking. Prior efforts, such as (Zhang et al., 2025b), focus on isolated dimensions like safety, without addressing temporal coherence or perceptual consistency. Overall, existing work emphasizes static reasoning or functional classification, overlooking human-centric perceptual responses and their evolution over time.

## 2.4 MLLM Benchmarks in Urban Scene

With Multimodal Large Language Models (MLLMs) advance in image understanding (Ma et al., 2024) and cross-modal reasoning (Huang et al., 2024a), benchmark datasets have evolved accordingly. Early benchmarks centered on basic Visual Question Answering (VQA), but such tasks no longer capture the full potential of modern MLLMs. To address this, several expert-level benchmarks have been introduced for domain-specific tasks with greater semantic and spatial complexity, especially in urban contexts. For example, V-IRL (Yang et al., 2024) focuses on street-view navigation and recognition; CityBench (Feng et al., 2025c) targets urban identity and navigation, though with limited task diversity. UrBench (Zhou et al., 2025) incorporates multi-view imagery from street and remote sensing sources for spatial reasoning. CityLens (Liu et al., 2025) evaluates urban function modeling using socio-economic indicators, and USTBench (Lai et al., 2025) assesses spatial planning via traffic and road network data. Despite these advances in modeling objective urban scenarios, most existing benchmarks are limited to static snapshots in time. They lack a comprehensive evaluation of models' ability to capture the spatiotemporal evolution of urban environments, particularly how physical transformations affect human subjective perception responses.

# 3 URBANFEEL

## 3.1 OVERVIEW

We present **UrbanFeel**, a comprehensive benchmark designed to evaluate the capabilities of Multimodal Large Language Models (MLLMs) in both physical understanding and subjective perception within the context of urban development. Built upon multi-view and multi-temporal street-view imagery collected from diverse global cities over the past 18 years, UrbanFeel simulates real-world urban evolution by capturing fine-grained visual and perceptual changes. As illustrated in Fig. 2, UrbanFeel includes four types of question format—binary judgment, multiple-choice, open-ended reasoning, and a novel temporal sorting format—resulting in over 14,300 high-quality QA samples, with approximately 11.0 K for validation and 3.3 K for testing. Detailed statistics and examples can be found in Appendix A.

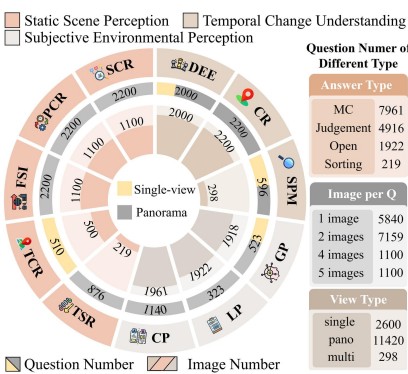

Figure 2: Data Statistics of UrbanFeel.

Unlike prior benchmarks that focus primarily on object detection or scene classification within urban imagery, as shown in Fig.1, UrbanFeel introduces several novel design dimensions. First, it systematically incorporates both single-view and panoramic street images captured in a certain sequence to evaluate models' ability to capture spatial context across viewpoints. Second, it integrates long-term urban development sequences—spanning more than a decade—to support tasks that require historical reasoning and temporal ordering. Third, UrbanFeel introduces human-centered affective perception tasks, covering four dimensions: beautiful, safety, wealthy, and lively. Each sample is additionally annotated with localized visual evidence, enabling the evaluation of model explainability and alignment with human perceptual cues. Although real-world urban analysis often involves iterative workflows, UrbanFeel deliberately targets the fundamental atomic perception and reasoning capabilities that serve as essential prerequisites for such complex decision-making. UrbanFeel thus offers a challenging and comprehensive evaluation framework for MLLMs in complex urban scenarios, laying a foundation for future studies on modeling the alignment and divergence between machine and human perception.

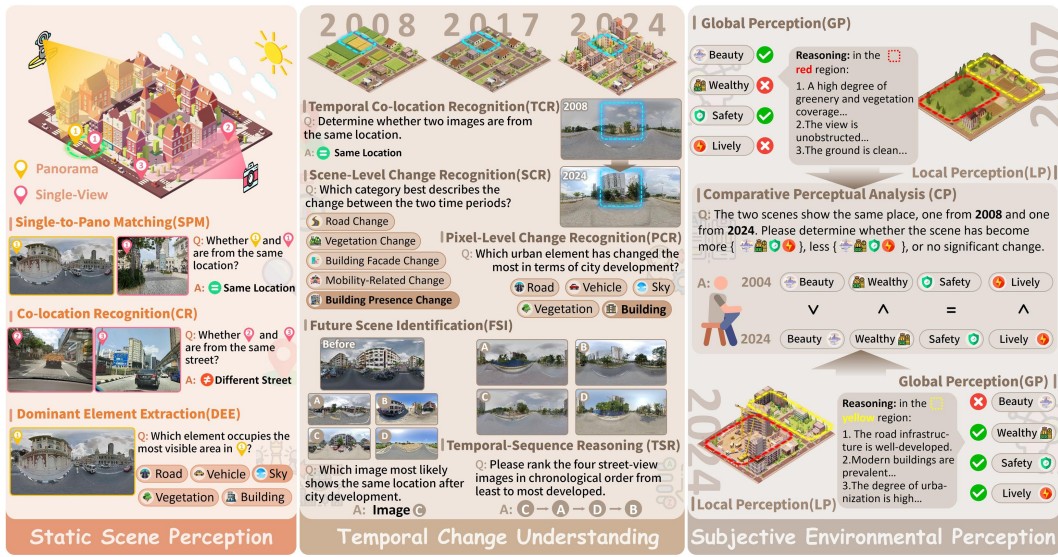

Figure 3: Overview of our UrbanFeel. UrbanFeel defines 11 sub-tasks spanning 3 cognitive dimensions: static scene perception, temporal change understanding, and subjective environmental perception.

## 3.2 BENCHMARK TASK

Guided by a cognitively progressive evaluation framework, we design 11 diverse tasks and construct **UrbanFeel**, a comprehensive benchmark for modeling urban development perception. As illustrated in Fig. 3, these tasks span three levels of cognitive depth—*Static Scene Perception*, *Temporal Change Understanding*, and *Subjective Environmental Perception*—enabling a multi-dimensional assessment of MLLMs across recognition, reasoning, and perceptual alignment.

**Static Scene Perception** focuses on evaluating models' ability to recognize salient visual elements and spatial consistency in a single time frame. Tasks under this category include identifying dominant visual components in a given image and determining whether a pair of images—single-view and panoramic—depict the same geographic location. This dimension retains some classic scene perception tasks and aims to assess models' capacity for snapshot-level spatial understanding and contextual matching.

**Temporal Change Understanding** targets the model's ability to detect, differentiate, and reason about visual changes over time. Beyond identifying structural variations across temporally aligned images, models are required to classify the type of urban evolution (e.g., façade renovation, road maintenance, or vegetation growth) and to perform temporal ordering of multiple images based on perceived development stages. These tasks simulate human-like reasoning about city progression and test the model's temporal-spatial integration abilities.

**Subjective Environmental Perception** emphasizes the alignment between MLLMs and human subjective evaluation. We construct affective perception tasks across four dimensions—*beautiful*, *safe*, *wealthy*, and *lively*—and require models not only to produce scalar judgments but also to provide localized visual justifications. In addition, we introduce before–after comparison tasks to examine whether models can detect perceptual shifts in changing environments. This dimension moves beyond objective recognition, probing whether MLLMs can simulate human affective responses in complex visual scenes.

## 3.3 BENCHMARK CURATION

**Data Collection and Pre-processing.** As shown in Figure 4, the UrbanFeel benchmark collects over 4,000 street-view images from 11 cities across four continents via Mapillary and the Google Street View API, covering both single-view and panoramic formats. The selected cities include representative locations from the Global South (e.g., Kuala Lumpur, Tolyatti) and the Global North

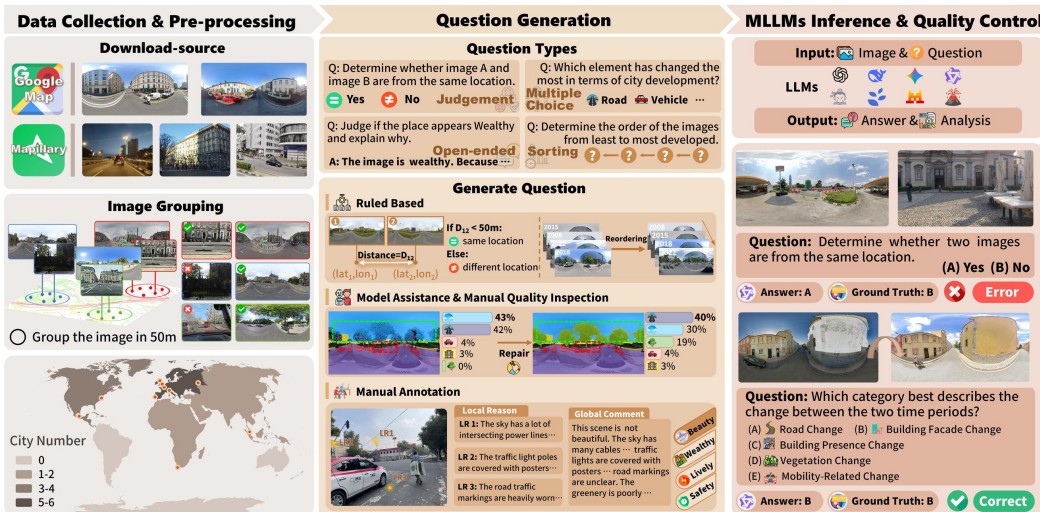

Figure 4: Benchmark construction process of UrbanFeel, including data collection and pre-processing, question generation, and MLLMs inference and quality control.

(e.g., Paris, Washington, D.C.), spanning a temporal range from 2007 to 2024 and capturing diverse stages of urban development.

During the data pre-processing stage, given the lack of precise spatial or temporal ordering in some images, we apply spatiotemporal clustering based on geolocation and timestamps to generate coherent urban evolution sequences. To ensure data quality, we use a pretrained segmentation model along with manual filtering to remove low-quality samples, such as indoor scenes, blurry captures, and heavily occluded images. Additional preprocessing details are provided in the Appendix C.

**Question Generation.** UrbanFeel supports four question formats: binary judgment, multiple choice, sorting, and open-ended QA. To efficiently generate diverse question sets, we adopt a hybrid strategy combining rule-based generation, model-assisted prompting, and manual authoring. For instance, tasks like same-place matching and future-view prediction are generated using temporal and spatial metadata. For change-type recognition, we initialize annotations with outputs from general-purpose segmentation models, followed by manual verification and correction. Subjective perception tasks are written entirely by human annotators and span four dimensions: beautiful, safety, wealthy, and lively. Annotators also mark localized visual evidence to support reasoning and explainability evaluation.

**MLLMs Inference and Quality Control.** To ensure annotation accuracy and evaluation reliability, we introduce a multi-stage validation pipeline. During model inference, strict output formatting constraints are enforced. Responses are automatically assessed using a separate language model to compare with human-provided ground truths. For ambiguous or illogical responses, manual review is conducted to reclassify or remove problematic samples. This filtering ensures the final evaluation metrics are robust and reproducible.

## 4 EXPERIMENTS

### 4.1 EVALUATED MODELS

We evaluate a total of 20 multimodal large models under a zero-shot setting using UrbanFeel, including 2 closed-source models and 18 open-source models. The closed-source models are GPT-4o (OpenAI, 2024) and Gemini 2.5 Pro (Comanici et al., 2025), accessed via their official APIs. The open-source models cover a diverse set of representative MLLM families, including DeepSeek-VL2 (Wu et al., 2024b), InternVL 3 (Zhu et al., 2025), LLaVA (Guo & Huang, 2025), Qwen2.5-VL (Yu et al., 2025), Phi (Abdin et al., 2024), Gemma-3 (Team et al., 2025), and Idefics3-8B (Laurençon et al., 2024), among others. A full list of model versions and configurations is provided in Appendix C.

Table 1: Quantitative results for 2 closed-source and 18 open-source MLLMs, as well as those for human and random guess across 11 tasks. The overall score is computed across all tasks. The maximum value and the second largest value of model performance in each task are indicated by the **bold** and underlined text, respectively. Task names are abbreviated for brevity.

| Model | Static Scene Perception | | | Temporal Change Perception | | | | | Subjective Perception Consistency | | | Overall |
|---|---|---|---|---|---|---|---|---|---|---|---|---|
| | DEE | CR | SPM | TCR | FSI | PCR | TSR | SCR | GP | LP | CP | |
| DeepSeek-vl2-tiny | 44.5 | 18.0 | 44.5 | 48.2 | 25.5 | 31.5 | 3.7 | 24.1 | 59.6 | **53.8** | 43.6 | 36.1 |
| DeepSeek-vl2 | 59.3 | 29.0 | 43.7 | 94.8 | 53.8 | 37.1 | 8.2 | 38.9 | 65.8 | 40.9 | 33.3 | 45.9 |
| MiniCPM-V 2.6-8B | 45.9 | 94.8 | 77.6 | 90.9 | 26.8 | 38.7 | 10.5 | 25.2 | 41.9 | 34.8 | 33.9 | 47.4 |
| Qwen2.5-vl-3B | 61.1 | 51.0 | 76.2 | 77.8 | 43.2 | 28.9 | 5.0 | 17.1 | 67.6 | 35.6 | 28.3 | 44.7 |
| Qwen2.5-vl-7B | 52.4 | **98.2** | 75.8 | 85.1 | 43.2 | 33.4 | 10.0 | 43.9 | 56.6 | 33.8 | 38.6 | 51.9 |
| Qwen2.5-vl-72B | 60.1 | 97.2 | 66.2 | 87.9 | 90.3 | **40.9** | 26.0 | 46.0 | 65.7 | 44.6 | 36.9 | 60.2 |
| LLaVA-1.5-7B | 26.3 | 51.0 | 49.0 | 88.3 | 25.8 | 24.6 | 3.7 | 17.0 | 59.5 | 39.2 | **53.2** | 39.8 |
| LLaVA-v1.6-mistral-7B | 34.9 | 51.0 | 45.5 | 65.0 | 17.5 | 28.0 | 3.7 | 16.1 | 67.2 | 38.7 | 51.0 | 38.1 |
| InternVL3-2B | 51.3 | 41.4 | 65.5 | 66.6 | 19.6 | 38.9 | 1.4 | 15.0 | 68.2 | 40.0 | 25.4 | 39.4 |
| InternVL3-8B | 39.8 | 67.4 | 53.8 | 78.4 | 31.8 | 33.5 | 7.8 | 32.2 | 69.7 | 34.1 | 38.8 | 44.3 |
| Phi-3.5 | 46.0 | 57.2 | 51.4 | 75.4 | 56.9 | 24.8 | 7.8 | 37.4 | 54.6 | 37.4 | 36.2 | 44.1 |
| Phi-4 | 37.7 | 26.2 | 69.7 | 82.9 | 57.4 | 32.1 | 2.3 | 39.9 | **71.0** | 42.2 | 48.1 | 46.3 |
| Idefics3-8B | 47.1 | 56.6 | 60.0 | 53.2 | 22.5 | 39.5 | 3.7 | 15.5 | 61.2 | 34.1 | 49.0 | 40.2 |
| Mistral-Small-3.1-24B | 19.6 | 91.6 | 67.9 | 86.6 | 64.3 | 28.9 | 16.0 | 46.7 | 63.3 | 42.4 | 39.9 | 51.6 |
| Aria | **64.8** | 90.0 | 71.7 | 89.0 | 42.7 | 38.9 | 10.5 | 38.1 | 67.7 | 42.1 | 45.9 | 54.7 |
| Aya-vision-8b | 18.8 | 51.0 | 38.6 | 51.5 | 26.2 | 24.6 | 3.2 | 33.3 | 69.6 | 43.8 | 39.5 | 36.4 |
| Gemma-3-4b | 42.5 | 81.6 | 64.1 | 59.5 | 25.9 | 39.0 | 8.2 | 36.5 | 53.1 | 46.1 | 26.7 | 43.9 |
| Gemma-3-27b | 50.8 | 80.2 | 66.9 | 58.5 | 76.2 | 37.5 | 18.7 | 44.8 | 64.7 | 39.6 | 41.0 | 52.7 |
| GPT-4o | 50.8 | 97.2 | **79.2** | 89.2 | 74.2 | 40.5 | 38.9 | 49.9 | 60.2 | 37.3 | 36.4 | 59.4 |
| Gemini-2.5-pro | **64.8** | 96.3 | 78.2 | **95.4** | 97.6 | 36.5 | **52.1** | 56.5 | 67.7 | 49.0 | 30.3 | **65.9** |
| Human | 71.3 | 88.1 | 76.7 | 96.4 | 96.4 | 21.2 | 70.0 | 69.5 | 66.6 | 32.9 | 53.1 | 67.4 |
| Random | 21.4 | 48.6 | 50.3 | 47.6 | 26.3 | 19.5 | 3.9 | 18.8 | 51.0 | 18.2 | 35.2 | 31.5 |

## 4.2 EVALUATION PROTOCOL

UrbanFeel includes four question types: binary judgment, multiple choice, sorting, and open-ended QA. Following the evaluation protocol of prior benchmarks such as MMMU (Yue et al., 2024) and UrBench (Zhou et al., 2025), we adopt a hybrid strategy combining exact matching, model verification, and semantic similarity evaluation.

For non-open-ended questions (i.e., judgment, multiple choice, and sorting), we first apply strict string matching—answers are considered correct only if they match the reference label. However, since some models generate verbose responses without clearly selecting an option, we employ an auxiliary language model to assess whether the prediction semantically aligns with the ground truth, ensuring fair evaluation of models that include rationale in their outputs. For open-ended questions, correctness is determined by measuring semantic similarity between the model-generated answer and reference answers.

For the human baseline, we recruited two independent groups of ten participants each, all with geography-related academic backgrounds (undergraduate, master's, or doctoral students). One group conducted the annotations, while the other performed the evaluations, ensuring no overlap between the two. More implementation details of evaluation protocol are described in Appendix B.

## 4.3 MAIN RESULTS

**Overall Challenge of UrbanFeel.** Table 1 summarizes the overall quantitative performance of mainstream Multimodal Large Language Models (MLLMs) on UrbanFeel, revealing the significant challenges posed by our benchmark. While closed-source models such as GPT-4o and Gemini-2.5-Pro demonstrate impressive capabilities on selected tasks, their overall performance remains substantially behind human-level accuracy—particularly in tasks that require compositional reasoning and spatiotemporal understanding. This performance gap is even more pronounced for open-source models, suggesting that current MLLMs still face major limitations in practical applications related to urban development, environmental perception, and city planning.

**Performance Across Task Dimensions.** Table 1 further disaggregates model performance across the 11 sub-tasks in UrbanFeel. Most MLLMs exhibit strong capabilities in basic visual recognition tasks; for instance, the majority of models achieve over 60% accuracy on the Time-Consistent Recognition (TCR) task, which requires only straightforward temporal identification.

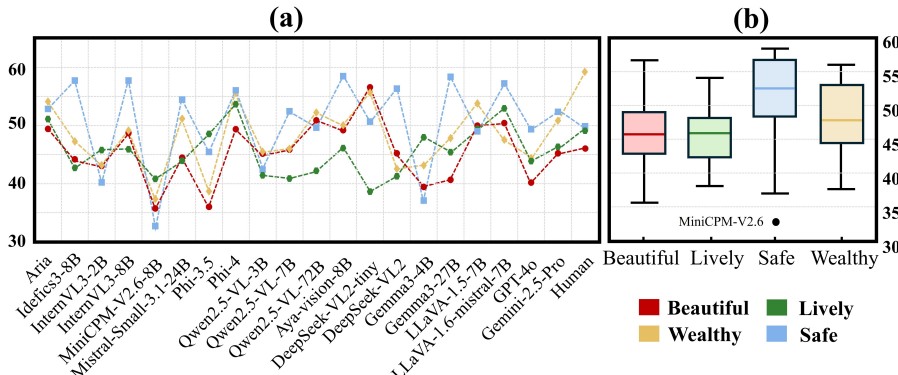

Figure 5: Quantitative comparison of MLLM performance on subjective environment perception. (a) Accuracy across four dimensions. (b) Box plots show model variance, where the horizontal lines in boxes indicate medians; box width indicates consistency.

However, model performance drops significantly when spatial and temporal reasoning must be integrated. In the TSR task, for example, most models score below 10% accuracy. Even the best-performing model—Gemini-2.5-Pro—still lags behind human performance by 17.9% in accuracy, revealing that existing models still struggle with long-range temporal ordering and scene-level alignment across time and space.

Interestingly, in the dimension of subjective environmental perception, many models show strong consistency with human judgment (CP, LP). Their visual justifications—such as cues used to infer aesthetic or safety—often align closely with those identified by human annotators, suggesting early potential for human-aligned perceptual reasoning. However, this alignment weakens substantially when temporal dynamics are introduced. On tasks involving perceptual comparison between before–after scenes, most models exhibit heightened sensitivity to visual changes, often overemphasizing fine-grained variations and diverging from human-level perceptual stability. More subjective analysis cases will be displayed in the Appendix D.

To our surprise, in the PCR task with panoramic inputs, MLLMs outperform human evaluators. This is because humans are less sensitive to pixel-level differences caused by panoramic distortions. While evaluators focus on salient foreground changes, mid- or long-range building variations occupy only a small pixel proportion and may be less noticeable than background shifts in sky or road caused by slight camera movements, leading to frequent misjudgments.

## 5 DISCUSSION

### 5.1 MODEL PERFORMANCE ACROSS SUBJECTIVE PERCEPTION DIMENSIONS

To further investigate model behavior in subjective environmental perception, Figure 5(a) presents accuracy distributions across four key dimensions. The results show that MLLMs exhibit considerable variation across dimensions. Most models achieve human-comparable or even superior accuracy in dimensions such as *Safe*, *Beautiful*, and *Lively*, suggesting promising potential for aligning with human perceptual judgments in urban scenes.

Among these, *Safe* is the dimension where most models perform best, reaching an average accuracy of 50.6%. However, this dimension also reveals the largest performance gap between models, indicating substantial inconsistency in safety-related judgments. In contrast, *Lively* displays more stable accuracy across models, despite having a slightly lower average performance, suggesting that models more consistently capture liveliness, likely by relying on broad visual cues such as vehicles and crowds. However, in the *Wealthy* dimension, the models still underperform human evaluators by an average margin of 10.1%, implying that wealth perception may involve more nuanced or culturally specific visual cues that current models struggle to capture.

The box plot in Figure 5(b) further supports these observations. Although the *Safe* dimension has the highest median accuracy (52.4%), it also exhibits the widest interquartile range and largest overall variance, confirming the inconsistent model behavior. Conversely, the *Lively* dimension has

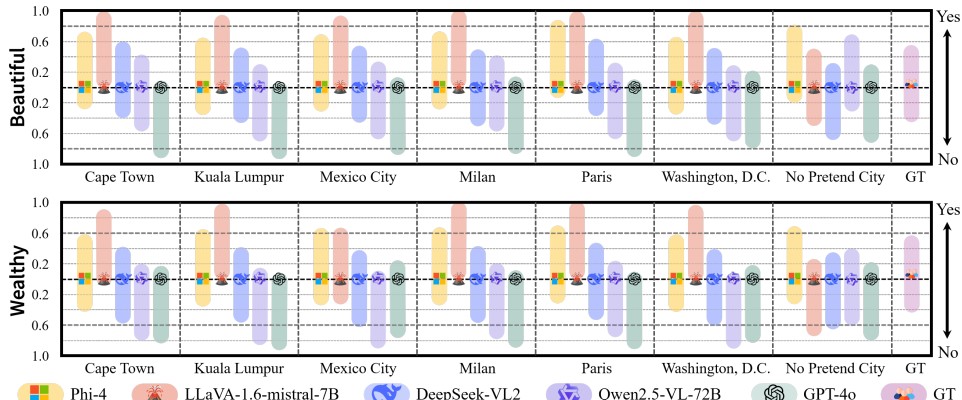

Figure 6: Quantitative comparison of different models under the assumed city identity setting. "Yes" indicates the proportion of positive evaluations made by MLLMs for the given perceptual dimension, while "No" represents the proportion of negative evaluations. The results show that LLaVA-1.6-mistral-7B and DeepSeek-vl2 yield more positive evaluations across most cities, while Qwen2.5-VL and GPT-4o show a decline under assumed city identity.

the most concentrated distribution, indicating higher inter-model agreement. This consistency suggests that current MLLMs may rely on more universal or easily detectable signals when evaluating liveliness, whereas dimensions like safety and wealth require finer-grained perceptual reasoning or socio-cultural understanding.

## 5.2 DOES CITY IDENTITY AFFECT MLLMS' SUBJECTIVE ENVIRONMENTAL JUDGMENTS?

To examine whether MLLMs exhibit geographic bias in subjective perception, we conducted a "city identity intervention" experiment. We randomly selected 100 street-view images from the GP validation set and assigned each one of six hypothetical city identities (Cape Town, Kuala Lumpur, Mexico City, Milan, Paris, Washington, D.C.), comparing the results to those without any assigned identity ("No Pretend City"). Figure 6 shows the distributions of positive (e.g., "beautiful") and negative (e.g., "not beautiful") judgments under the *Beautiful* and *Wealthy* dimensions. Results for *Lively* and *Safe* are included in the Appendix C.

Overall, most models exhibit varying degrees of change in their subjective judgments when city identity is introduced. LLaVA-1.6 and DeepSeek-vl2 tend to produce more positive evaluations across most cities, suggesting a tendency to interpret identity labels favorably. In contrast, Phi-4 demonstrates high stability, indicating greater reliance on image content and robustness to added semantic labels. Notably, GPT-4o and Qwen2.5-VL show a general decline in positive judgments when city identity is provided, implying a more "cautious" or even "conservative" evaluation behavior, potentially triggered by the activation of learned stereotypes or expectations.

When comparing "Global North" and "Global South" city identities, we observe that **a northern identity does not necessarily lead to more favorable evaluations.** Although the average score for northern cities is slightly higher, GPT-4o's perception of wealth for "Paris" and "Milan" drops significantly—sometimes even below that of cities like "Cape Town." This counterintuitive result may stem from a mismatch between the semantic label and the actual image content; for example, ordinary or aged urban scenes labeled as "Paris" may result in greater expectation gaps, prompting the model to generate more negative evaluations. Conversely, GPT-4o's slightly increased positive judgments for "Mexico City" may be attributed to positive visual signals such as modern buildings, clean streets, and bright lighting—combined with the lack of strong negative priors associated with the label "Mexico City" in the model's pretraining corpus.

## 5.3 DO MLLMS PERCEIVE SINGLE-VIEW AND PANORAMA DIFFERENTLY?

To evaluate whether the differences in viewpoint coverage and information organization between single-view and panorama introduce perceptual biases in MLLMs, we compare model performance across the two perspectives. As shown in Figure 7, the majority of models consistently perform

better on single-view images than on their panorama counterparts. On average, single-view inputs yield an accuracy improvement of 11.7% over panoramic inputs. Notably, Gemini-2.5-Pro achieves the highest accuracy on single-view images at 69.4%, closely by Qwen2.5-VL-72B with 64.9%. In terms of panoramic images, Aria shows the best performance with 55.3%, while Gemini-2.5-Pro follows closely with with 54.9%.

This performance gap suggests that although panorama offer greater spatial coverage and denser visual information, their inherent geometric distortions and contextual blending may increase the "cognitive burden" on MLLMs. It also shows that MLLMs have perspective data imbalance and bias during training process.

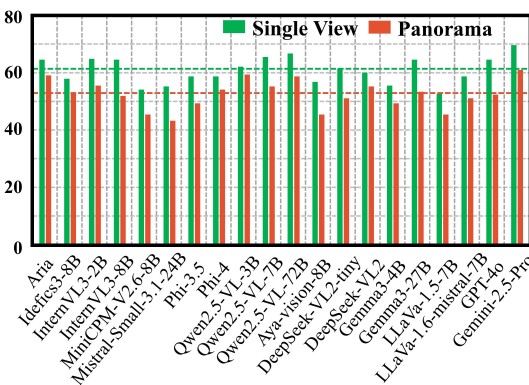

Figure 7: Quantitative results of MLLMs performance from different perspectives.

### 5.4 DOES EXPLICIT REASONING ENHANCE TEMPORAL UNDERSTANDING?

To investigate whether guiding MLLMs with explicit reasoning steps enhances performance in complex urban temporal tasks, we conducted a controlled ablation study on the Temporal Sequence Reasoning (TSR) task, comparing Direct Sorting, General CoT, and Re-Thinking strategies. Contrary to the expectation that explicit reasoning consistently improves performance, our results reveal a counter-intuitive trend: for state-of-the-art models such as GPT-4o, Qwen2.5-VL-72B, and Gemini-2.5-Pro, **the Direct Sorting strategy consistently yields the highest accuracy**. Introducing verbose reasoning steps often leads to performance degradation; for instance, GPT-4o's accuracy drops from 46.1% to 37.0% when using the Re-Thinking strategy. Qualitative analysis suggests this stems from an "over-reasoning" pitfall, where models hallucinate subtle details or over-interpret transient noise to justify an incorrect linear progression. We provide detailed quantitative results and specific case studies of this phenomenon in Appendix C.4 and D.

## 6 CONCLUSION

In this study, we introduces UrbanFeel, a new benchmark for evaluating the capabilities of Multimodal Large Language Models (MLLMs) in urban development understanding and subjective perception. The benchmark includes over 14.3K questions across 11 tasks, covering static scene perception, temporal change understanding, and subjective environmental perception. It is constructed using single-view and panoramic street-view images from 11 cities, spanning more than 15 years. We evaluate 20 MLLMs and identify key limitations. Current models underperform in tasks requiring joint spatial–temporal reasoning. We also observe geographic bias in subjective perception tasks, where predictions vary with city identity. In addition, models show perceptual inconsistencies across different viewpoints, particularly between single-view and panoramic inputs. We envision Urban-Feel advancing perception-aware urban intelligence. By bridging temporal evolution and human perception, this work positions MLLMs as scalable tools for the continuous monitoring and assessment required to achieve sustainable urban development goals.

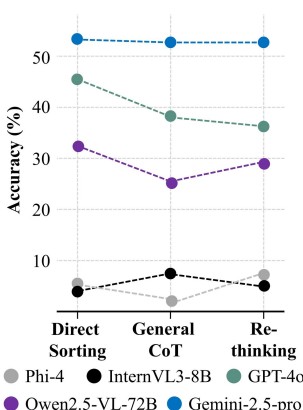

Figure 8: Quantitative results of different prompt input on TSR.

### ACKNOWLEDGEMENT

This research is funded by the National Natural Science Foundation of China (Grant No. 62571560); Natural Science Foundation of Guangdong Province (grant No. 2024A1515012083) and the National Key Research and Development Program of China (grant No. 2022YFB3903402) .

ETHICS STATEMENT

We developed UrbanFeel guided by principles of responsible and ethical AI research. We acknowledge that, despite efforts to ensure coverage and diversity, our dataset and annotations may still carry biases from source imagery, annotator backgrounds, or cultural contexts—especially in subjective perception tasks such as beauty, safety, wealth, and liveliness. Users should remain vigilant to these limitations. To mitigate bias, we provided annotators with standardized guidelines and training before labeling. The annotations are intended as reference labels that reflect the consensus inclinations of the annotator group, not as absolute ground truth. Moreover, we recognize the possibility that models built on UrbanFeel might be misused to influence public perceptions or aesthetic judgments. Our intent is for positive applications—urban analysis, perceptual model evaluation, and human–machine alignment research—and we explicitly disavow any malicious uses. By releasing our data, code, and evaluation tools, we hope to foster transparency, accountability, and further work toward fairer, cross-cultural, and ethically grounded AI in urban contexts.

REPRODUCIBILITY STATEMENT

We are committed to ensuring the full reproducibility of our work. The core concepts and methodology of our benchmark design are detailed in Section 3, including the three task dimensions and the benchmark curation. Our experimental setup, including datasets, evaluation procedures, and baseline MLLMs, is described in Section 4 and Appendix C. Additional details on the manual annotation process, human evaluation settings, and operational guidelines for subjective perception tasks are provided in Appendix B. To facilitate the reproduction of our results and to support further research in urban multimodal perception, we will release the complete codebase, benchmark dataset (UrbanFeel), and evaluation scripts, along with detailed documentation and pretrained model predictions.

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

**UrbanFeel: A Comprehensive Benchmark for Temporal and Perceptual Understanding of City Scenes through Human Perspective**

Supplementary Material

In this appendix, we present supplementary materials that could not be included in the main paper due to space constraints. These materials offer extended details on the benchmark construction, evaluation protocols, and additional experimental results to support reproducibility and inspire future research. Specifically, we provide:

- **Additional Benchmark Statistics:** Including detailed question distributions and comparative analysis with existing urban perception benchmarks.

- **Benchmark Construction Details:** Covering the entire pipeline from data collection and pre-processing to manual annotation and evaluation protocols.

- **Experiment Details and Additional Results:** Listing all evaluated baseline models, presenting extended experimental results, and analyzing whether reasoning-augmented models outperform standard baselines on UrbanFeel. We also provide detailed results on how assigned city identities influence model perception across different dimensions.

- **Case Studies:** Illustrative examples showcasing model responses and human annotations across UrbanFeel's 11 task types.

- **Limitations and Future Work:** Discussing current constraints of UrbanFeel and outlining directions for further expansion and refinement.

- **Use of Large Language Models:** Describe the main uses of LLM in the writing of this manuscript.

## A    ADDITIONAL BENCHMARK STATISTICS

### A.1    QUESTION STATISTICS

Table 2 summarizes the number of test and validation instances across different subtasks. Urban-Feel comprises a carefully curated set of 14.3K visual questions, with 11K in the test set and 3.3K in the validation set. Our benchmark provides a comprehensive suite of representative questions under three evaluation dimensions, enabling a holistic assessment of MLLMs' capabilities in understanding spatiotemporal urban dynamics and aligning with human perception in complex urban development scenarios.

Table 2: Question Distribution in Test and Val Sets

| Evaluation Dimension | Task | Test | Val |
|---|---|---|---|
| **Static Scene Perception** | DEE | 1600 | 400 |
| | CR | 400 | 100 |
| | SPM | 238 | 60 |
| **Temporal Change Perception** | TCR | 1760 | 440 |
| | FSI | 880 | 220 |
| | PCR | 880 | 220 |
| | TSR | 160 | 40 |
| | SCR | 880 | 220 |
| **Subjective Perception Consistency** | GP | 1358 | 560 |
| | LP | 1638 | 280 |
| | CP | 1241 | 720 |
| **Total** | – | **11035** | **3260** |

Table 3: Comparison of Existing Urban Scene Benchmarks. Our UrbanFeel designs a variety of question types over a longer time span, comprehensively evaluating the perception ability of different MLLMs on the physical space and human subjective dimensions of the environment in urban development scenarios.

| Feature | CityBench | CityGPT | USTBench | Urbanch | UrbanFeel (Ours) |
|---|---|---|---|---|---|
| Judgements | ✗ | ✗ | ✗ | ✗ | ✓ |
| Multi-Choice | ✓ | ✓ | ✓ | ✓ | ✓ |
| Open-Ended | ✗ | ✗ | ✗ | ✓ | ✓ |
| Sorting | ✗ | ✗ | ✗ | ✗ | ✓ |
| Static Scene | ✓ | ✓ | ✓ | ✓ | ✓ |
| Historical Data | ✗ | ✗ | ✗ | ✗ | ✓ |
| Subjective Perception | ✗ | ✗ | ✗ | ✗ | ✓ |
| Temporal Resolution | Hourly or Event-based | Hourly or event-based | Hourly with planning loops | Event-based | Yearly sequence |

## A.2 BENCHMARK COMPARISON

Table 3 presents a systematic comparison between UrbanFeel and existing urban perception benchmarks. Current benchmarks such as CityBench (Feng et al., 2025c), CityGPT (Feng et al., 2025a), and USTBench (Lai et al., 2025) primarily focus on single-timestamp static images or tasks at the hourly or event level, aiming to assess models' understanding of urban infrastructure or real-time dynamics. These tasks are often constructed using short-term trajectories, single-frame imagery, or synthetic datasets, and lack modeling or reasoning over real-world urban evolution.

In contrast, UrbanFeel emphasizes evaluating MLLMs' multimodal perception capabilities within the long-term context of urban development. It introduces street-view image sequences spanning 17 years, capturing visual transformations across phases of urban planning, expansion, and renovation. More importantly, UrbanFeel goes beyond physical spatial changes by systematically incorporating subjective environmental perception, designing tasks that assess perceptions of *beautiful*, *lively*, *safe*, and *wealthy*—thus exploring how human perceptions of environmental quality shift across different urban contexts, and how well models align with such perceptions.

Additionally, UrbanFeel supports diverse task formats—including multiple choice, open-ended reasoning, and ranking—and covers both static scene understanding and dynamic urban transformations. It fills critical gaps in spatiotemporal reasoning and human-centric perception evaluation, establishing the first comprehensive multimodal benchmark framed from the perspective of human–city interaction.

## B BENCHMARK CONSTRUCTION DETAILS

### B.1 DATA COLLECTION DETAILS

This section outlines the data collection pipeline for all street-view imagery used in **UrbanFeel**. The dataset comprises two main components: *single-view images* and *panoramic images*. We first obtained the vector boundaries of 11 representative cities using *OpenStreetMap*.

For single-view imagery, we utilized the *Mapillary API* provided by the Global Streetscapes dataset to download images. Each image was renamed using a standardized format based on the provided timestamp, geographic coordinates, and image ID: {lat,lon}_{year}_{month}_{image_id}.jpg, to facilitate streamlined preprocessing and file management. We strictly adhere to the CC BY-SA license for Mapillary data, using official tools for acquisition and ensuring that all released content satisfies the attribution and share-alike requirements.

For panoramic imagery, we employed the *Google Street View API* to retrieve data and aligned the naming convention with that of the single-view images. This ensures consistency across spatial

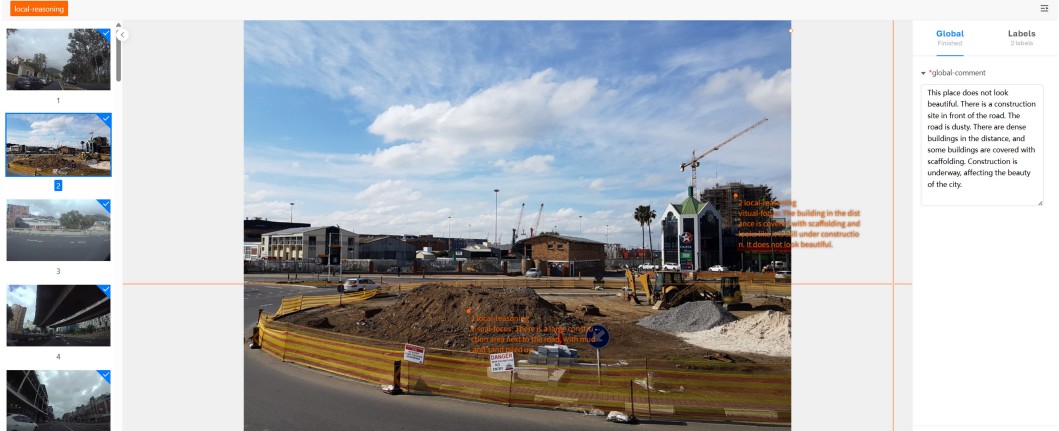

Figure 9: An example of the user interface of LabelU.

and temporal dimensions, enabling cross-view and temporal comparisons in subsequent benchmark tasks.To ensure full compliance with Google's Terms of Service and copyright regulations, the open-source version of UrbanFeel does not distribute raw Google Street View imagery. Instead, we release only the unique Panorama IDs and metadata, accompanied by a retrieval script that allows researchers to legally fetch the images via the official API using their own credentials.

## B.2 DATA PRE-PROCESSING DETAILS

During data preprocessing, we first standardized the orientation of all street-view images by rotating them to face true north based on camera heading metadata, minimizing the impact of viewpoint variation in panoramic images on downstream MLLM perception. Since the images lack explicit spatial relationships, we performed spatial clustering by calculating pairwise geodesic distances using image coordinates, with a 50-meter threshold to identify multi-view, multi-temporal image sequences from the same urban location.

We then applied the OneFormer(Jain et al., 2023) semantic segmentation model to preprocess the images, discarding those with less than 5% sky coverage, which were likely captured indoors. Finally, we manually filtered out low-quality images affected by motion blur, overexposure, or other visual defects.

## B.3 MANUAL ANNOTATION AND EVALUATION DETAILS

During the annotation phase, acknowledging the inherent subjectivity of perception-related tasks, we recruited a group of 10 undergraduate and master's students with geography-related academic backgrounds. Annotators were provided with standardized written guidelines (as shown in Table 4) and a short training session with representative examples prior to the formal labeling process. They were instructed to identify localized visual evidence from the images that supported their global perceptual judgments. The resulting annotations are treated as *reference labels* rather than absolute ground truth, reflecting the consensus tendencies of this annotator group. All annotation work was conducted using the **LabelU** platform. Fig. 9 illustrates an annotation case in the Local Perception task of the *Beauty* dimension.

For the evaluation phase, we recruited an independent group of 10 volunteers, entirely distinct from the annotators, comprising undergraduate, master's, and doctoral students in geography-related fields. This separation ensured that annotation and evaluation were performed by different populations, thereby reducing potential bias introduced by overlapping roles. All human assessments were conducted on the **LabelLLM** platform, which provided a standardized interface for task interaction and response collection.

It is important to note that subjective concepts such as *beauty*, *safety*, or *wealth* are inevitably influenced by cultural and personal perspectives. While the provision of operational guidelines and

Table 4: Annotation guidelines for subjective perception dimensions. These guidelines were provided to annotators as operational references rather than absolute criteria.

| Dimension | Description (Instruction for Annotators) |
|---|---|
| **Beauty** | Annotators were instructed to focus on the overall aesthetic impression of the scene, considering whether the environment appears visually harmonious, orderly, and pleasant. They were asked to pay attention to greenery, landscaping, architectural style and maintenance, cleanliness of streets, balance of colors, and whether the layout looks uncluttered. *Example:* a tree-lined avenue with well-maintained modern buildings should be labeled as more beautiful than a cluttered street with graffiti and broken infrastructure. |
| **Safety** | Annotators were instructed to evaluate whether the environment gives a sense of security, especially from a pedestrian's perspective. They were asked to check for cues such as adequate street lighting, visible sidewalks, orderly traffic, the presence of surveillance cameras or other visible security measures, and the absence of disorder (e.g., litter, vandalism). *Example:* a well-lit commercial street with open shops and visible security cameras should be labeled as safer than a dark, narrow alley with poor visibility and signs of decay. |
| **Wealth** | Annotators were instructed to judge the degree of perceived economic prosperity in the environment. They were asked to consider the quality and modernity of buildings, the presence of commercial activity (e.g., branded shops), the maintenance of infrastructure, and visible indicators of affluence (e.g., luxury cars). *Example:* a district with glass office towers and upscale retail should be labeled as wealthier than a neighborhood with dilapidated housing and cracked pavements. |
| **Liveliness** | Annotators were instructed to assess the vibrancy and human activity in the scene. They were asked to pay attention to pedestrians, cyclists, vehicles, open businesses, street vendors, or public events, as well as infrastructure supporting activity (benches, playgrounds). *Example:* a busy marketplace with crowds and open shops should be labeled as more lively than an empty street with little visible activity. |

training sessions was intended to minimize ambiguity and promote consistency across participants, these annotations should be regarded as references produced by a specific annotator population, rather than universal ground truth. Future work will extend this framework through cross-cultural annotation campaigns and inter-annotator agreement analyses to further address cultural bias and subjective variability.

## B.4 EVALUATION PROTOCOLS

Given the varying structures and formats of different question types in UrbanFeel, we adopt tailored evaluation strategies for each to ensure fairness and reproducibility.

To provide a rigorous quantitative assessment, we adopt Accuracy (ACC) as the primary evaluation metric across all tasks. The accuracy is computed as:

$$ACC = \frac{N_{\text{Correct Predictions}}}{N_{\text{Total Predictions}}} \tag{1}$$

where Total Predictions denotes the total number of evaluated instances, and Correct Predictions denotes the number of instances for which the model output satisfies the task-specific correctness criterion.

**1. Exact Matching for Objective Tasks (MCQ, Binary, Sorting).** For multiple-choice, binary judgment, and sorting tasks, the correctness criterion corresponds to strictly matching the objective ground truth label or sequence. To handle verbose model outputs (e.g., when a model outputs reasoning alongside the option), we employ a two-step normalization process: first, we attempt rule-based parsing to extract the option label; if this fails, we use a lightweight LLM call solely to extract the predicted label (e.g., "Option A") without altering the semantic content. The extracted label is then compared against the ground truth using exact string matching.

**2. Semantic Similarity for Open-Ended Tasks.** For open-ended questions where exact string matching is too rigid, we employ a semantic similarity metric. We consider a prediction correct

if the cosine similarity between the Sentence-BERT embedding of the predicted answer $\hat{y}_i$ and the ground-truth text $y_i$ exceeds a pre-defined threshold $\tau$. This is formally expressed as:

$$\text{Correct}(\hat{y}_i) = \mathbf{1}\big(\text{sim}(\hat{y}_i, y_i) > \tau\big) \tag{2}$$

where $\mathbf{1}(\cdot)$ is the indicator function returning 1 if the condition is satisfied and 0 otherwise, and $\text{sim}(\cdot, \cdot)$ denotes the cosine similarity computed using a pre-trained Sentence-BERT model. In our experiments, we set the threshold $\tau = 0.6$ ensuring that correct but phrased-differently answers are accepted while irrelevant hallucinations are rejected.

# C EXPERIMENT DETAILS & ADDITIONAL RESULTS

## C.1 BASELINE MODELS

We evaluate a total of 20 state-of-the-art Multimodal Large Language Models (MLLMs), encompassing both open-source and closed-source models with diverse model sizes and capabilities. All models are capable of processing visual inputs and are assessed under a unified evaluation pipeline. The list of baseline models used in UrbanFeel includes:

1. **GPT** (OpenAI, 2024): We adopt the latest version of GPT-4o as the representative model from the GPT series.
2. **Gemini**(Comanici et al., 2025): Gemini-2.5-Pro is selected as the representative of the Gemini family.
3. **Qwen** (Yu et al., 2025): We evaluate three variants of Qwen2.5-VL, including the 3B, 7B, and 72B checkpoints.
4. **InternVL** (Zhu et al., 2025): InternVL3-2B and InternVL3-8B are included as vision-language expert models.
5. **MiniCPM** (Yao et al., 2024): MiniCPM-V-2.6 is used to represent lightweight MLLMs.
6. **DeepSeek** (Wu et al., 2024b): Both DeepSeek-VL2-tiny and DeepSeek-VL2 are included to explore performance scaling trends.
7. **LLaVA** (Li et al., 2024b): We include LLaVA-1.5 (7B) and LLaVA-1.6-mistral as representatives of this popular open-source family.
8. **Mistral** (Jiang et al., 2023): Mistral-Small-3.1-24B-Instruct is evaluated as a strong language-focused baseline.
9. **Gemma** (Team et al., 2025): Both Gemma-3-4B and Gemma-3-27B checkpoints are assessed.
10. **Aya** (Dash et al., 2025): The Aya-vision-8B checkpoint is included.
11. **Phi** (Abdin et al., 2024): We evaluate multimodal variants of Phi-3.5 and Phi-4.
12. **Aria** (Li et al., 2024a): We also include the open-source Aria model.

All models are accessed via their official APIs or released checkpoints, and evaluated using a standardized prompt structure and visual input protocol to ensure fairness and consistency. To ensure reproducibility, we set the temperature to 0 and perform greedy decoding.

## C.2 DO REASONING-AUGMENTED MODELS OUTPERFORM BASELINES ON URBANFEEL?

To systematically assess the benefits of reasoning capabilities for urban perception tasks, we conduct a comparative analysis of various base MLLMs and their reasoning-augmented counterparts across five sub-tasks of the UrbanFeel benchmark(SPM, TCR, TSR, SCR, LP). As shown in Table 5, these tasks span a wide spectrum, from static scene understanding and cross-view matching to temporal semantic consistency reasoning and fine-grained local perception.

Experimental results indicate that reasoning-augmented models (e.g., QVQ, GPT-o3, Gemini-2.5-Pro-thinking) generally perform better on tasks emphasizing spatial understanding. For instance, in SPM and TCR—tasks that require scene consistency and localized judgment—GPT-o3 achieves accuracy rates of 86.2% and 93.8%, respectively. In the subjective perception task (LP), QVQ brought

Table 5: Quantitative comparison results of reasoning-augmented model and the non-reasoning-augmented model.The maximum value and the second largest value of model performance in each task are indicated by the **bold** and underlined text, respectively. Task names are abbreviated for brevity.

| Model | SPM | TCR | TSR | SCR | LP |
|---|---|---|---|---|---|
| Qwen2.5VL-72B | 66.2 | 87.9 | 26.0 | 46.0 | 36.9 |
| QVQ | 68.3 | 76.4 | 17.5 | 24.5 | 48.7 |
| GPT-4o | 81.4 | 89.2 | 38.9 | 49.9 | 37.3 |
| o3 | **86.2** | 93.8 | 37.0 | 25.5 | 37.0 |
| Gemini-2.5-Pro | 80.3 | 95.4 | **52.1** | **56.5** | **49.0** |
| Gemini-2.5-Pro-thinking | 75.9 | **96.3** | 39.5 | 24.4 | 42.6 |
| Human | 76.7 | 96.4 | 70.0 | 69.5 | 32.9 |

an 11.8% increase in accuracy compared to Qwen2.5-VL-72B, suggesting a stronger alignment with human perception.

However, reasoning does not consistently lead to performance gains across all tasks. In the temporally ordered TSR task, the reasoning-augmented models exhibit varying degrees of performance degradation, which may be attributed to the extended reasoning span required when processing multiple images, thereby limiting the models' ability to effectively capture and model the relationships among these images. In the scene change-sensitive SCR task, several reasoning-augmented models even show significant declines, with an average accuracy drop of 51.2% compared to their non-reasoning counterparts. This suggests that reasoning models may overemphasize fine-grained differences when facing abrupt scene transitions, thereby overlooking global semantic coherence and resulting in perceptual misjudgments.

### C.3 Do MLLMs Possess Robust Generalization Across Diverse Cities?

Due to space limitations, this appendix provides the quantitative results of MLLMs on different perception dimensions across the six cities mentioned in the main paper, under the Global Perception (GP) task without city identity intervention, as shown in Table 10 to Table 13.

To assess model robustness across diverse urban environments, we analyzed performance variations at the city level, revealing two distinct generalization patterns. In beautiful dimension (table 10) of global perception (GP task), leading closed-source models exhibit an "inverse geographic bias"; for instance, Gemini-2.5-Pro aligns more closely with human perception in Global South cities (averaging 67.1%) than in the Global North (52.4%), suggesting a reduced tendency to idealize Western aesthetics. Conversely, open-source models demonstrate severe city-specific overfitting. LLaVA-1.5-7B achieves near-perfect accuracy in Washington (98.1%) but drops drastically to 38.9% in Kuala Lumpur, indicating a reliance on US-centric training data rather than true perceptual generalization.

This geographic variance extends to objective recognition tasks (PCR and DEE), distinguishing robust generalists from brittle systems. As shown in table 8 and 9 In pixel-level detection (PCR), while models like Aria maintain consistency (e.g., 53% in Washington vs. 47% in Tolyatti), others suffer catastrophic collapse; notably, Aya-vision-8b plummets from 41% to 6%. Furthermore, the Dominant Element Extraction (DEE) task reinforces the "inverse bias" phenomenon even in objective settings: GPT-4o surprisingly achieves 70% accuracy in Cape Town versus just 35% in Washington. These sharp contrasts underscore the critical value of UrbanFeel's multi-city framework, as aggregate metrics frequently mask significant regional failures that only granular geographic evaluation can reveal.

### C.4 Additional City Identity Intervention Results

In Discussion section of the main paper, we analyzed how different models' perceptions of the *Beautiful* and *Wealthy* dimensions shift under hypothetical city identity interventions. Due to space limitations, this appendix provides the quantitative results of MLLMs on different perception dimensions across the six cities mentioned in the main paper, under the Global Perception (GP) task

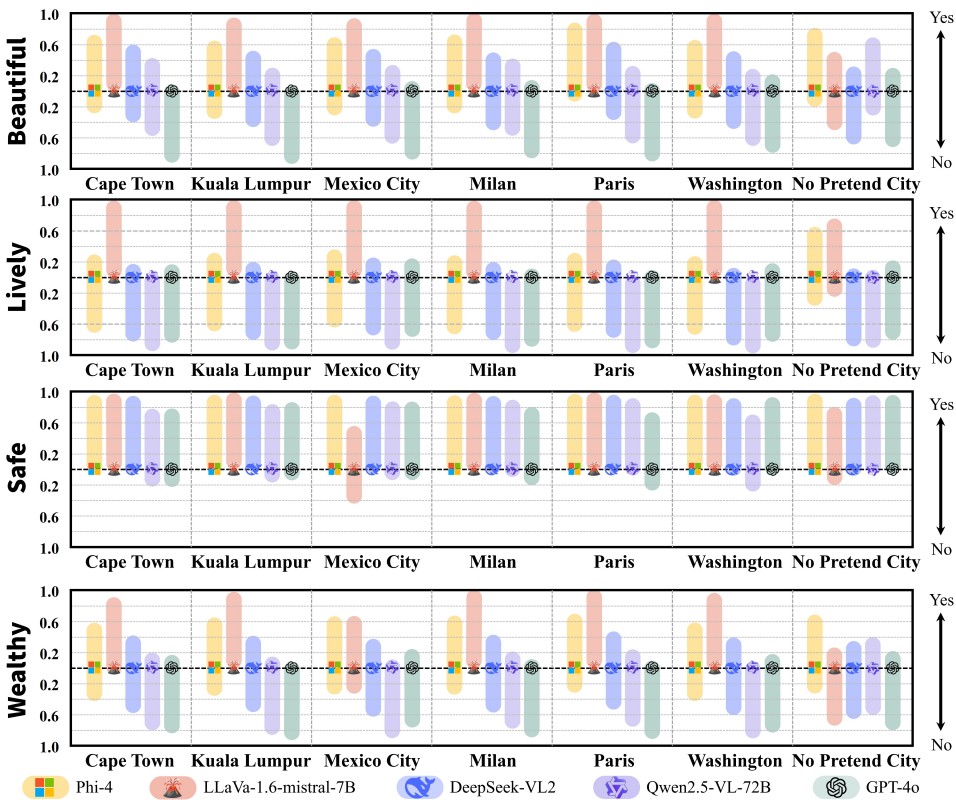

Figure 10: Quantitative comparison of different models under the assumed city identity setting."Yes" indicates the proportion of positive evaluations made by the model for the given perceptual dimension, while "No" represents the proportion of negative evaluations.

without city identity intervention, as shown in Table 10 to Table 13. Fig. 10 supplements the experimental results of City Identity Intervention under the *Lively* and *Safe* dimensions.

Our findings reveal significant inter-dimensional differences in model sensitivity to identity prompts. For the *Lively* dimension, most models exhibit a consistent tendency toward negative judgments. GPT-4o, Qwen2.5-VL-72B, and DeepSeek-VL2 remain largely stable before and after city identity assignment, suggesting minimal perceptual bias. In contrast, Phi-4 shows a notable decline in positive evaluations—dropping from around 60% to below 30% after identity intervention. Interestingly, LLaVA-1.6 demonstrates the opposite trend, labeling almost all images as "Lively," indicating high susceptibility to identity cues.

For the *Safe* dimension, most models maintain a high rate of positive judgments regardless of city identity, suggesting more robust safety perception. The only exception is LLaVA-1.6, which shows a marked decrease when the identity "Mexico City" is assigned—potentially reflecting latent safety-related stereotypes learned from training data.

## C.5 ADDITIONAL DETAILS ON REASONING ABLATION STUDY

This section provides the experimental details and supplementary data for the reasoning ablation study discussed in Section 5.4. As noted in the main text, we compared three prompting strategies to evaluate the efficacy of explicit reasoning in the Temporal Sequence Reasoning (TSR) task.

The specific contents of the prompts used for each strategy are illustrated in Figure 11. The strategies are designed as follows:

- **Direct Sorting (P0):** A concise prompt asking only for the final chronological sequence (e.g., "[Image A → Image B → ...]") without intermediate reasoning steps.

**Direct Sorting Prompt**

Question: You are given four street-view images (Image A, B, C, and D), each captured at a different point in time from the same location. These images reflect different stages of urban development.
Your task is to determine the correct chronological order of the images, from the earliest captured (oldest) to the latest captured (most recent).
Please respond only with the image order in this format:
[Image X → Image Y → Image Z → Image W]

**General CoT Prompt**

Question: You are given four street-view images (Image A, B, C, and D), each taken at a different point in time from the same location. These images reflect different stages of urban development. Your task is to determine the correct chronological order of the images from the earliest captured (oldest) to the latest captured (most recent).

To complete this task, first analyze each image based on visual cues such as building construction, road quality, greenery, public infrastructure, and signs of modernization.

Please follow this format in your response:

1. **Answer**: List the image order from the earliest captured (oldest) to the latest captured (most recent), using this format:
   [Image X → Image Y → Image Z → Image W]
2. **Reasoning**: Briefly explain why you chose this order, referring to the key urban development features you observed in the images.
Let's think step by step.

**Re-thinking Prompt**

Question: You are given four street-view images (Image A, B, C, and D) from the same location, each captured at a different time. They show different stages of urban development.
Your task is to determine the correct chronological order of the images, from the earliest captured (oldest) to the latest captured (most recent).

Step 1 — Initial order:
   Propose an initial order of the images from earliest to latest.
   Base your decision on permanent features such as building construction, infrastructure, and long-term changes in greenery.
   Ignore transient factors such as cars, pedestrians, or weather.

Step 2 — Consistency check:
   Check whether any image in your order appears to reverse development in time (e.g., a partly demolished building after a fully completed one, or a newly built structure appearing before an empty lot).
   If such cases exist, adjust the order and briefly explain why.

Final response format:
Initial order: [Image X → Image Y → Image Z → Image W]
Checked order: [Image X → Image Y → Image Z → Image W]
Reasoning: a short explanation referring to permanent features and any non-monotonic changes you considered when deciding the final earliest-to-latest order.

Figure 11: Different prompt type of TSR ablation study.

- **General CoT (P2):** A structured prompt requiring the model to first list "permanent features" (buildings, infrastructure) and explicitly ignore "transient factors" (cars, weather) before deriving the order.

- **Re-Thinking (P3):** A multi-step prompt where the model first proposes an initial order, then performs a "consistency check" for non-monotonic changes (e.g., reverse development), and finally outputs the corrected sequence.

Table 6 lists the detailed accuracy scores for all seven evaluated models across the three strategies. The results confirm that while reasoning strategies (P2, P3) offer marginal gains for smaller models like Phi-4 (improving from 5.5% to 7.3%), they consistently degrade the performance of high-capacity models (GPT-4o, Qwen2.5-VL-72B) compared to the Direct Sorting baseline.

To further illustrate the "over-reasoning" phenomenon, Figure 12 visualizes a representative failure case. Despite correctly identifying some features, the General CoT prompt leads the model to hallucinate a "linear growth" narrative—specifically, claiming that vegetation in Image A is "visibly larger" than in Image D to justify a later timestamp—while ignoring the definitive structural evidence of a new blue building in Image D. This confirms that verbose reasoning can induce confirmation bias, overriding visual evidence with plausible-sounding but incorrect logical chains.

Table 6: Full ablation results on the TSR task. Consistent with the discussion in Section 5.4, the **Direct Sorting** strategy yields the highest accuracy for all SOTA models.

| Model | Direct Sorting | General CoT | Re-Thinking |
|---|---|---|---|
| Phi-4 | 5.5 | 4.1 | 7.3 |
| InternVL3-8B | 4.6 | 3.2 | 5.0 |
| Qwen2.5-VL-72B | 32.0 | 24.7 | 29.7 |
| GPT-4o | 46.1 | 38.4 | 37.0 |
| Gemini-2.5-Pro | 52.5 | 52.1 | 52.1 |
| o3 | 60.3 | 59.8 | 60.7 |
| Gemini-2.5-Pro-Thinking | 54.8 | 49.8 | 51.1 |

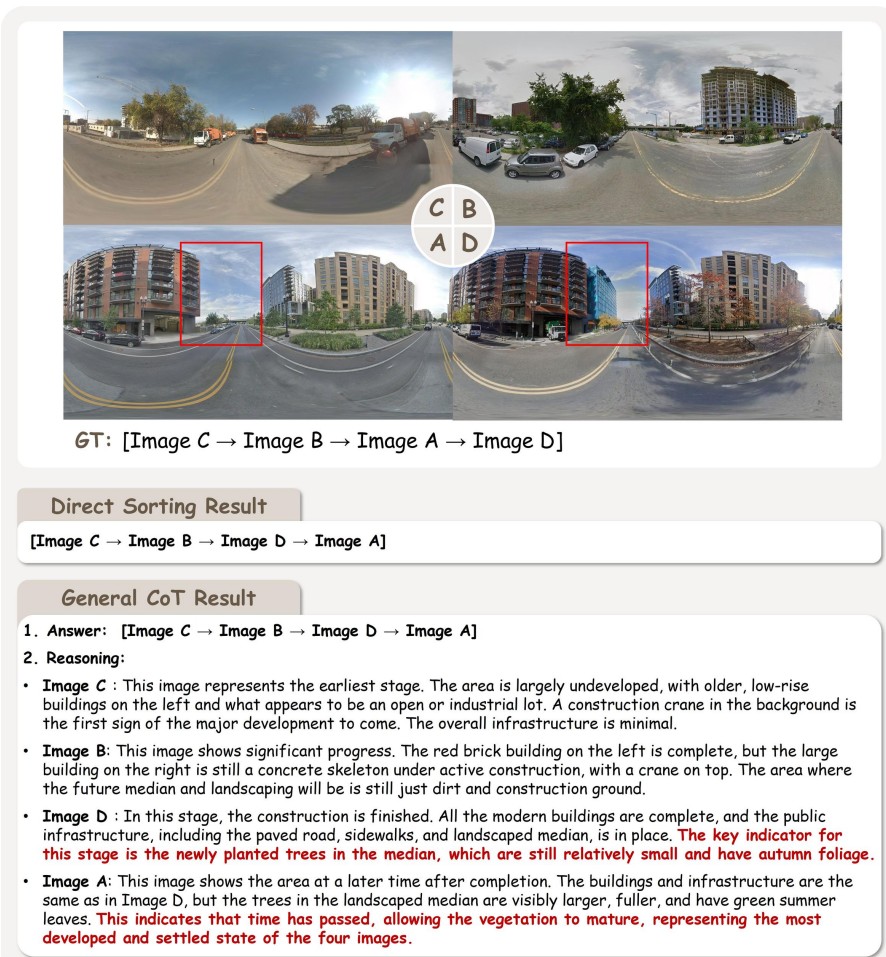

Figure 12: A faliure case of TSR tasks using different type of prompt. The red text indicates incorrect reasoning results from Gemini-2.5-pro.

## C.6 STATISTICAL SIGNIFICANCE ANALYSIS

To validate the robustness of our findings regarding model performance gaps and human-level consistency, we conducted a formal statistical analysis based on the sample sizes reported in Table 2. We calculated 95% Confidence Intervals (CIs) for accuracy using the Wilson score interval and performed two-proportion z-tests to assess the significance of performance differences.

Table 7 summarizes the statistical comparison between Human evaluators and the best-performing model (Gemini-2.5-Pro) across two representative tasks: Temporal Sequence Reasoning (TSR) and Global Perception (GP).

Table 7: Statistical significance analysis of performance gaps between Human evaluators and the best-performing MLLMs. The analysis reveals three distinct distinct capability regimes: **Significant Inferiority** in temporal reasoning (TSR), **Statistical Parity** in subjective perception (GP), and **Significant Superiority** in pixel-level detection (PCR).

| Task | Sample Size ($N$) | Subject | Accuracy (%) | 95% CI | Significance |
|------|------|------|------|------|------|
| TSR | 219 | Human
Gemini-2.5-pro | 70.0
52.1 | $[64.0, 76.0]$
$[45.5, 58.7]$ | $p < 0.01$ |
| GP | 1,918 | Human
Phi-4 | 66.6
67.7 | $[64.5, 68.7]$
$[65.6, 69.8]$ | Not Significant |
| PCR | 1,100 | Human
Qwen2.5-vl-72B | 21.2
40.9 | $[18.8, 23.6]$
$[38.0, 43.8]$ | $p < 0.001$ |

**1. Temporal Reasoning Gap (TSR):** For the TSR task ($N = 219$), the 95% CIs for Humans and Gemini-2.5-Pro are clearly separated (Human: $[64.0, 76.0]$ vs. Gemini: $[45.5, 58.7]$). The non-overlapping intervals and a z-test ($p < 0.01$) indicate that the 17.9% performance difference is statistically significant, statistically supporting our conclusion that current MLLMs achieve substantially lower accuracy than humans on long-range temporal ordering.

**2. Human-Level Parity in Perception (GP):** For the GP task ($N = 1,918$), the large sample size yields narrow error margins ($\approx \pm 2.1\%$). The overlapping CIs indicate that the accuracies of Gemini-2.5-Pro and human annotators are statistically indistinguishable in this setting. This validates our statement that state-of-the-art MLLMs have reached a level of consistency comparable to human annotators for general scene perception.

**3. Surpassing Humans in detection (PCR):** Interestingly, for the PCR task ($N = 1,100$), models significantly outperform humans (40.9% vs. 21.2%). The distinct separation of CIs ($[38.0, 43.8]$ vs. $[18.8, 23.6]$) and a $z$-test ($p < 0.001$) provide robust statistical evidence of this advantage. As discussed, this is likely because human evaluators struggle to identify subtle pixel-level changes obscured by the geometric distortions inherent in panoramic imagery, whereas MLLMs maintain high sensitivity to such fine-grained variations.

# D  CASE STUDY

In this section, we present illustrative examples of model responses and corresponding ground-truth labels across the 11 distinct sub-tasks designed in UrbanFeel (Figure 13 to Figure 41). The examples span a wide range of perception categories—including static scene understanding, temporal change understanding, and subjective enviromental consistency—revealing both the strengths and limitations of current models in handling real-world urban dynamics.

## D.1  QUALITATIVE ERROR ANALYSIS

Based on the qualitative breakdown of model reasoning chains across these tasks, we identify four primary categories of failure modes that limit the performance of current MLLMs in urban contexts:

**Over-reliance on surface semantics over spatial invariance (Temporal Context).** In tasks requiring temporal context understanding, models often prioritize salient surface-level semantic features while neglecting spatial geometric invariance. For instance, in the Temporal Co-location Recognition task (Figure 19), when an empty lot evolved into a developed residential block, models such as Mistral-Small and Aria incorrectly classified the pair as "Different Locations" solely due to the emergence of new buildings. This indicates a deficiency in utilizing invariant cues—such as road layout and curvature—to recognize that the images depict the same geographic location despite drastic semantic shifts over time.

**"Linear development" assumption in urban evolution and neglect of urban decay (Temporal Reasoning).** Regarding temporal reasoning, models frequently exhibit a "linear development" bias, often operating under the heuristic that newer or cleaner infrastructure always corresponds to a later timestamp. This leads to failures in accounting for urban decay or complex maintenance cycles.

In the sorting example shown in Figure 28, GPT-4o chronologically misplaced an image showing worn road markings before a pristine one, ignoring critical alignment cues like speed bumps. This limitation is further illustrated in the urban renewal scenario in Figure 29. Here, models consistently identified the intermediate demolition/construction phase as the earliest stage, placing it chronologically before the original standing building. This reveals a rigid "Tabula Rasa" heuristic—assuming that any construction site represents the genesis of development—thereby failing to recognize non-monotonic processes where established neighborhoods undergo decline or renewal.

**Spatial–perspective misalignment (Static Spatial Perception).** In static spatial perception, models demonstrate significant difficulties in cross-view mapping, particularly between single-view and panoramic imagery. In the Single-to-Pano Matching task (Figure 13), models like Aria failed to match a single-view crop to its corresponding panorama. This failure suggests that models treat the geometric distortions inherent in panoramic projections as semantic differences rather than perspectival variations, revealing limitations in performing spatial transformations on 360° imagery.

**Hallucinated visual evidence in subjective reasoning (Subjective Perception).** Finally, in subjective perception tasks, models occasionally hallucinate negative visual cues to justify conservative or biased classifications. For example, in the Local Perception task (Figure 35), GPT-4o categorized a scene as "not wealthy" by citing non-existent "shuttered storefronts" and "lack of investment," despite visual evidence of well-maintained infrastructure. This reveals a disconnect between the reasoning chain and the actual pixel data, where models generate plausible-sounding but factually incorrect evidence to support a high-level prior impression.

# E    LIMITATIONS & FUTURE WORK

While *UrbanFeel* provides a comprehensive benchmark for evaluating MLLMs in urban development understanding and human-centered perception, several limitations remain. First, although the dataset spans 11 cities across different continents, it suffers from geographic imbalance, with underrepresentation of regions such as Africa and South America. This may affect the generalizability of models in culturally sensitive subjective perception tasks. Second, the annotations for affective perception tasks are statically defined, potentially failing to capture the temporal diversity and evolving nature of human opinions. Moreover, despite covering 18 years of visual urban change, Urban-Feel lacks explicit causal labels or structured socio-environmental metadata, limiting its capacity to support deeper reasoning about the underlying drivers of urban transformation.

In future work, we plan to address these limitations through a concrete roadmap focused on geographic inclusivity and causal depth:

**Geographic Expansion and Cultural Grounding.** We are actively expanding UrbanFeel to include cities with emerging temporal coverage, focusing on urban development tasks to mitigate regional representation bias. Furthermore, to address cultural bias in subjective perception, future iterations will aim to diversify the annotator pool to include local residents and refine annotation guidelines with region-specific examples. We plan to report perception scores in a stratified manner, ensuring that labels like "safety" or "beauty" reflect locally grounded interpretations rather than a single cultural perspective.

**Towards Causal Spatiotemporal Reasoning.** While UrbanFeel currently emphasizes visually verifiable changes, we aim to bridge the gap between pixel-level observation and socio-economic causality. Inspired by recent trends in urban analytics, we plan to align street-view segments with external urban datasets—such as land-use layers, POIs, and census statistics—to enable models to contextualize physical changes. For cities with accessible records, we will introduce coarse-grained causal event annotations (e.g., new transit line openings, policy-driven redevelopment projects) to support reasoning about the drivers of urban evolution. Building on this, we will also explore how Urban-Feel tasks can be composed into multi-turn, scenario-based evaluations that more closely resemble real planning and policy workflows.

# F    USE OF LARGE LANGUAGE MODELS

During the preparation of this manuscript, Large Language Models (LLMs) were used solely for grammar checking and language polishing. The use of LLMs was strictly limited to improving

the clarity, readability, and overall presentation quality of the text. All aspects related to research idea, experimental code development, and result analysis were strictly conceived and completed independently by the authors.

Table 8: Quantitative comparison of different MLLMs in the Pixel-level Change Recognition (PCR) task across cities. City names are replaced with abbreviations.

| Model | Cam. | Cap. | Edi. | K.L. | Lis. | Mex. | Mil. | Par. | Tol. | Tyl. | Was. |
|---|---|---|---|---|---|---|---|---|---|---|---|
| DeepSeek-vl2-tiny | 31 | 29 | 33 | 35 | 26 | 30 | 26 | 39 | 21 | 29 | 48 |
| DeepSeek-vl2 | 34 | 36 | 33 | 35 | 25 | 43 | 30 | 44 | 27 | 48 | 54 |
| MiniCPM-V 2.6-8B | 38 | 49 | 35 | 40 | 34 | 34 | 44 | 30 | 42 | 53 | 37 |
| Qwen2.5-vl-3B | 35 | 30 | 35 | 35 | 31 | 33 | 31 | 41 | 22 | 33 | 47 |
| Qwen2.5-vl-7B | 39 | 30 | 31 | 33 | 25 | 41 | 37 | 47 | 26 | 33 | 50 |
| Qwen2.5-vl-72B | 37 | 39 | 36 | 34 | 32 | 37 | 45 | 48 | 42 | 51 | 49 |
| LLaVA-1.5-7B | 22 | 20 | 21 | 31 | 21 | 38 | 16 | 38 | 17 | 6 | 41 |
| LLaVA-v1.6-mistral-7B | 26 | 19 | 28 | 33 | 23 | 40 | 21 | 43 | 19 | 12 | 44 |
| InternVL3-2B | 41 | 44 | 29 | 36 | 28 | 42 | 32 | 45 | 27 | 53 | 51 |
| InternVL3-8B | 34 | 33 | 27 | 34 | 24 | 40 | 19 | 47 | 23 | 43 | 48 |
| Phi-3.5 | 23 | 18 | 21 | 31 | 21 | 40 | 15 | 40 | 17 | 6 | 41 |
| Phi-4 | 30 | 27 | 27 | 34 | 25 | 40 | 19 | 43 | 25 | 40 | 43 |
| Idefics3-8B | 39 | 32 | 39 | 40 | 35 | 35 | 34 | 44 | 26 | 51 | 59 |
| Mistral-Small-3.1-24B | 25 | 22 | 29 | 30 | 23 | 40 | 19 | 45 | 20 | 21 | 43 |
| Aria | 45 | 33 | 42 | 36 | 28 | 39 | 30 | 44 | 30 | 47 | 53 |
| Aya-vision-8b | 22 | 18 | 21 | 31 | 21 | 40 | 15 | 39 | 17 | 6 | 41 |
| Gemma-3-4b | 39 | 30 | 40 | 42 | 36 | 36 | 30 | 46 | 28 | 50 | 54 |
| Gemma-3-27b | 37 | 41 | 32 | 35 | 28 | 26 | 41 | 42 | 34 | 47 | 49 |
| GPT-4o | 30 | 41 | 30 | 40 | 29 | 46 | 43 | 49 | 43 | 48 | 47 |
| Gemini-2.5-pro | 30 | 35 | 25 | 33 | 35 | 38 | 34 | 54 | 34 | 42 | 41 |

Table 9: Quantitative comparison of different MLLMs in the dominant element extraction (DEE) task across cities. City names are replaced with abbreviations.

| Model | Cam. | Cap. | Edi. | K.L. | Lis. | Mex. | Mil. | Par. | Tol. | Tyl. | Was. |
|---|---|---|---|---|---|---|---|---|---|---|---|
| DeepSeek-vl2-tiny | 42 | 32 | 40 | 49 | 35 | 66 | 41 | 45 | 31 | 45 | 59 |
| DeepSeek-vl2 | 57 | 51 | 58 | 56 | 48 | 79 | 67 | 62 | 49 | 51 | 68 |
| MiniCPM-V 2.6-8B | 38 | 55 | 52 | 45 | 34 | 58 | 37 | 42 | 33 | 38 | 54 |
| Qwen2.5-vl-3B | 56 | 70 | 51 | 59 | 61 | 72 | 69 | 61 | 56 | 45 | 62 |
| Qwen2.5-vl-7B | 38 | 65 | 61 | 60 | 43 | 62 | 47 | 43 | 48 | 45 | 45 |
| Qwen2.5-vl-72B | 53 | 62 | 61 | 64 | 46 | 74 | 64 | 54 | 50 | 54 | 67 |
| LLaVA-1.5-7B | 23 | 21 | 29 | 29 | 18 | 33 | 23 | 27 | 20 | 29 | 33 |
| LLaVA-v1.6-mistral-7B | 35 | 26 | 28 | 42 | 31 | 52 | 40 | 37 | 26 | 23 | 41 |
| InternVL3-2B | 35 | 39 | 60 | 41 | 48 | 71 | 50 | 51 | 46 | 41 | 70 |
| InternVL3-8B | 37 | 55 | 43 | 44 | 27 | 45 | 28 | 35 | 27 | 36 | 41 |
| Phi-3.5 | 46 | 31 | 43 | 46 | 39 | 70 | 40 | 47 | 33 | 40 | 66 |
| Phi-4 | 37 | 31 | 35 | 41 | 32 | 46 | 37 | 36 | 26 | 43 | 47 |
| Idefics3-8B | 41 | 60 | 50 | 47 | 37 | 49 | 34 | 41 | 46 | 56 | 42 |
| Mistral-Small-3.1-24B | 6 | 27 | 30 | 24 | 13 | 11 | 18 | 22 | 10 | 32 | 11 |
| Aria | 58 | 65 | 62 | 57 | 57 | 75 | 72 | 68 | 63 | 59 | 66 |
| Aya-vision-8b | 8 | 29 | 25 | 24 | 13 | 12 | 16 | 21 | 10 | 26 | 9 |
| Gemma-3-4b | 40 | 54 | 58 | 33 | 28 | 48 | 26 | 31 | 40 | 48 | 41 |
| Gemma-3-27b | 42 | 65 | 66 | 41 | 41 | 50 | 45 | 41 | 50 | 58 | 41 |
| GPT-4o | 42 | 70 | 72 | 43 | 42 | 52 | 36 | 36 | 53 | 54 | 35 |
| Gemini-2.5-pro | 52 | 76 | 72 | 66 | 55 | 71 | 59 | 62 | 70 | 61 | 53 |

Table 10: Quantitative comparison results of different MLLMs in the Beautiful dimension under GP tasks. The maximum value and the second largest value of model performance in each city are indicated by the **bold** and underlined text, respectively.

| Model | Cape Town | Kuala Lumpur | Mexico City | Milan | Paris | Washington |
|---|---|---|---|---|---|---|
| DeepSeek-vl2-tiny | 57.5 | 44.4 | 62.9 | 69.5 | **69.7** | 94.3 |
| DeepSeek-vl2 | 55.0 | 72.2 | 74.2 | 48.8 | 63.6 | 71.7 |
| MiniCPM-V 2.6-8B | 42.5 | 66.7 | 54.8 | 39.0 | 39.4 | 20.8 |
| Qwen2.5-vl-3B | 57.5 | **88.9** | 71.0 | 53.7 | 65.2 | 81.1 |
| Qwen2.5-vl-7B | 72.5 | 66.7 | 69.4 | 64.6 | 63.6 | 81.1 |
| Qwen2.5-vl-72B | **75.0** | 50.0 | 69.4 | 64.6 | 65.2 | 84.9 |
| LLaVA-1.5-7B | 62.5 | 38.9 | 59.7 | 67.1 | 66.7 | **98.1** |
| LLaVA-v1.6-mistral-7B | 47.5 | **88.9** | 66.1 | 64.6 | 62.1 | 64.2 |
| InternVL3-2B | 60.0 | 72.2 | 72.6 | 69.5 | 68.2 | 96.2 |
| InternVL3-8B | 70.0 | **88.9** | **79.0** | 56.1 | 59.1 | 75.5 |
| Phi-3.5 | 47.5 | 83.3 | 66.1 | 25.6 | 50.0 | 41.5 |
| Phi-4 | 57.5 | 83.3 | 62.9 | 68.3 | 62.1 | 88.7 |
| Idefics3-8B | 60.0 | 77.8 | 74.2 | 50.0 | 50.0 | 58.5 |
| Mistral-Small-3.1-24B | 70.0 | **88.9** | **79.0** | 56.1 | 59.1 | 75.5 |
| Aria | 57.5 | 66.7 | 64.5 | **72.0** | 63.6 | 88.7 |
| Aya-vision-8b | 57.5 | 72.2 | 64.5 | 64.6 | 68.2 | 83.0 |
| Gemma-3-4b | 57.5 | 72.2 | 61.3 | 35.4 | 51.5 | 54.7 |
| Gemma-3-27b | 60.0 | 55.6 | 71.0 | 41.5 | 56.1 | 66.0 |
| GPT-4o | 60.0 | 72.2 | 72.6 | 42.7 | 53.0 | 47.2 |
| Gemini-2.5-pro | 55.0 | 72.2 | 74.2 | 41.5 | 59.1 | 56.6 |

Table 11: Quantitative comparison results of different MLLMs in the Lively dimension under GP tasks. The maximum value and the second largest value of model performance in each city are indicated by the **bold** and underlined text, respectively.

| Model | Cape Town | Kuala Lumpur | Mexico City | Milan | Paris | Washington |
|---|---|---|---|---|---|---|
| DeepSeek-vl2-tiny | 70.0 | **83.3** | 80.0 | 34.1 | 56.7 | 55.6 |
| DeepSeek-vl2 | 75.0 | 55.6 | 68.3 | 29.3 | 55.2 | 35.2 |
| MiniCPM-V 2.6-8B | 67.5 | 38.9 | 66.7 | 41.5 | 46.3 | 46.3 |
| Qwen2.5-vl-3B | 77.5 | 77.8 | 63.3 | 42.7 | 68.7 | 38.9 |
| Qwen2.5-vl-7B | 77.5 | 66.7 | 55.0 | 36.6 | 56.7 | 40.7 |
| Qwen2.5-vl-72B | 77.5 | 66.7 | 53.3 | 28.0 | 49.3 | 37.0 |
| LLaVA-1.5-7B | 75.0 | 33.3 | 75.0 | 26.8 | 50.7 | 46.3 |
| LLaVA-v1.6-mistral-7B | 70.0 | 72.2 | 63.3 | **81.7** | 70.1 | **63.0** |
| InternVL3-2B | 77.5 | 66.7 | 73.3 | 52.4 | 73.1 | 59.3 |
| InternVL3-8B | 80.0 | 66.7 | **83.3** | 39.0 | 65.7 | **63.0** |
| Phi-3.5 | 72.5 | 61.1 | 68.3 | 41.5 | 56.7 | 40.7 |
| Phi-4 | 57.5 | 66.7 | 70.0 | 78.0 | 76.1 | **63.0** |
| Idefics3-8B | 80.0 | 72.2 | 68.3 | 40.2 | 53.7 | 37.0 |
| Mistral-Small-3.1-24B | 72.5 | 72.2 | 70.0 | 25.6 | 47.8 | 46.3 |
| Aria | 75.0 | 77.8 | 80.0 | 50.0 | 67.2 | 48.1 |
| Aya-vision-8b | 70.0 | 72.2 | 75.0 | 57.3 | **80.6** | 57.4 |
| Gemma-3-4b | 65.0 | 55.6 | 81.7 | 56.1 | 76.1 | 57.4 |
| Gemma-3-27b | 77.5 | 55.6 | 73.3 | 31.7 | 59.7 | 44.4 |
| GPT-4o | 75.0 | 66.7 | 76.7 | 25.6 | 61.2 | 42.6 |
| Gemini-2.5-pro | **82.5** | 61.1 | 75.0 | 37.8 | 68.7 | 46.3 |

Table 12: Quantitative comparison results of different MLLMs in the Safe dimension under GP tasks. The maximum value and the second largest value of model performance in each city are indicated by the **bold** and underlined text, respectively.

| Model | Cape Town | Kuala Lumpur | Mexico City | Milan | Paris | Washington |
|---|---|---|---|---|---|---|
| DeepSeek-vl2-tiny | 32.5 | 27.8 | 24.2 | 19.5 | 25.4 | 7.7 |
| DeepSeek-vl2 | 70.0 | 61.1 | 68.1 | 82.9 | 74.6 | **87.2** |
| MiniCPM-V 2.6-8B | 25.0 | 33.3 | 20.9 | 22.0 | 23.9 | 15.4 |
| Qwen2.5-vl-3B | 65.0 | 66.7 | 68.1 | 81.7 | 71.6 | 85.9 |
| Qwen2.5-vl-7B | 70.0 | 66.7 | **70.3** | **84.1** | 73.1 | 85.9 |
| Qwen2.5-vl-72B | 70.0 | 66.7 | 69.2 | **84.1** | 74.6 | **87.2** |
| LLaVA-1.5-7B | 35.0 | 38.9 | 30.8 | 32.9 | 43.3 | 42.3 |
| LLaVA-v1.6-mistral-7B | 70.0 | 55.6 | 69.2 | 79.3 | 73.1 | 84.6 |
| InternVL3-2B | 67.5 | 61.1 | 65.9 | 76.8 | 70.1 | 82.1 |
| InternVL3-8B | 67.5 | 66.7 | 65.9 | 82.9 | 73.1 | 84.6 |
| Phi-3.5 | 65.0 | 66.7 | 53.8 | 47.6 | 64.2 | 61.5 |
| Phi-4 | 67.5 | 66.7 | **70.3** | 80.5 | 74.6 | 85.9 |
| Idefics3-8B | **75.0** | **72.2** | 60.4 | 78.0 | 67.2 | 85.9 |
| Mistral-Small-3.1-24B | 72.5 | 66.7 | 61.5 | 80.5 | 68.7 | **87.2** |
| Aria | 70.0 | **72.2** | 67.6 | 77.4 | **77.8** | 84.6 |
| Aya-vision-8b | 60.0 | 61.1 | 67.0 | 73.2 | 68.7 | 79.5 |
| Gemma-3-4b | 30.0 | 27.8 | 20.9 | 18.3 | 16.4 | 10.3 |
| Gemma-3-27b | 70.0 | 61.1 | 58.2 | 79.3 | 68.7 | 84.6 |
| GPT-4o | 65.0 | 66.7 | 59.3 | 74.4 | 68.7 | 76.9 |
| Gemini-2.5-pro | **75.0** | 66.7 | 64.8 | 79.3 | 73.1 | 80.8 |

Table 13: Quantitative comparison results of different MLLMs in the Wealthy dimension under GP tasks. The maximum value and the second largest value of model performance in each city are indicated by the **bold** and underlined text, respectively.

| Model | Cape Town | Kuala Lumpur | Mexico City | Milan | Paris | Washington |
|---|---|---|---|---|---|---|
| DeepSeek-vl2-tiny | 67.5 | 55.6 | 65.0 | 36.6 | 50.7 | 48.1 |
| DeepSeek-vl2 | 80.0 | **72.2** | 70.0 | 63.4 | 77.6 | 77.8 |
| MiniCPM-V 2.6-8B | 50.0 | 22.2 | 45.0 | 22.0 | 17.9 | 25.9 |
| Qwen2.5-vl-3B | 77.5 | 66.7 | 70.0 | 57.3 | 70.1 | 79.6 |
| Qwen2.5-vl-7B | 75.0 | **72.2** | 73.3 | 53.7 | 70.1 | 77.8 |
| Qwen2.5-vl-72B | 80.0 | 66.7 | 63.3 | 52.4 | 70.1 | 79.6 |
| LLaVA-1.5-7B | 77.5 | 61.1 | 71.7 | 67.1 | 71.6 | 64.8 |
| LLaVA-v1.6-mistral-7B | 70.0 | **72.2** | 61.7 | 52.4 | 64.2 | 61.1 |
| InternVL3-2B | 75.0 | **72.2** | 58.3 | 47.6 | 67.2 | 74.1 |
| InternVL3-8B | 77.5 | 66.7 | 73.3 | 75.6 | 74.6 | 70.4 |
| Phi-3.5 | 62.5 | 38.9 | 51.7 | 26.8 | 44.8 | 42.6 |
| Phi-4 | 70.0 | 55.6 | 58.3 | **87.8** | 80.6 | 74.1 |
| Idefics3-8B | 67.5 | 50.0 | 50.0 | 40.2 | 32.8 | 68.5 |
| Mistral-Small-3.1-24B | 70.0 | 66.7 | 73.3 | 41.5 | 62.7 | 75.9 |
| Aria | 60.0 | 61.1 | 66.7 | 52.4 | 71.6 | 72.2 |
| Aya-vision-8b | **82.5** | **72.2** | **76.7** | 68.3 | 79.1 | 68.5 |
| Gemma-3-4b | 70.0 | 61.1 | 53.3 | 42.7 | 70.1 | 74.1 |
| Gemma-3-27b | 66.7 | 64.7 | 56.8 | 57.9 | **81.2** | **86.5** |
| GPT-4o | 60.0 | 44.4 | 45.0 | 45.1 | 55.2 | 72.2 |
| Gemini-2.5-pro | 77.5 | **72.2** | 71.7 | 54.9 | 74.6 | 79.6 |

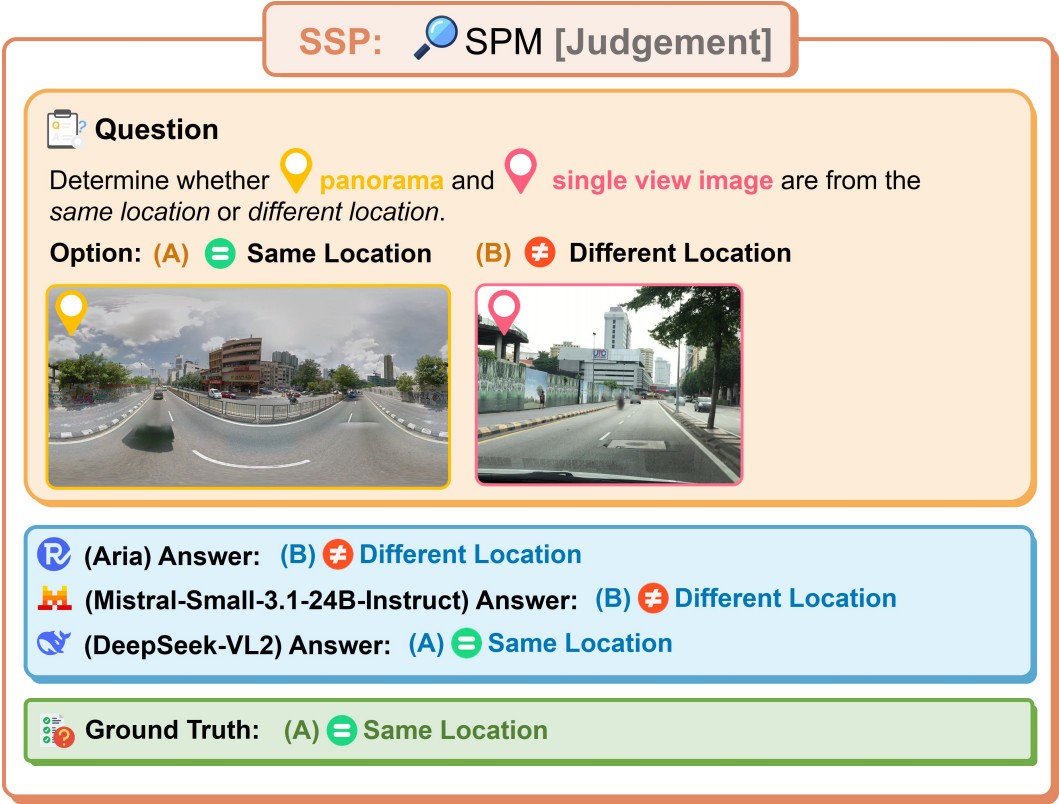

Figure 13: A question case of the **Single-to-Pano Matching(SPM)** task in UrbanFeel responses from *Aria*, *Mistral-Small-3.1-24B-Instruct*, *DeepSeek-VL2*

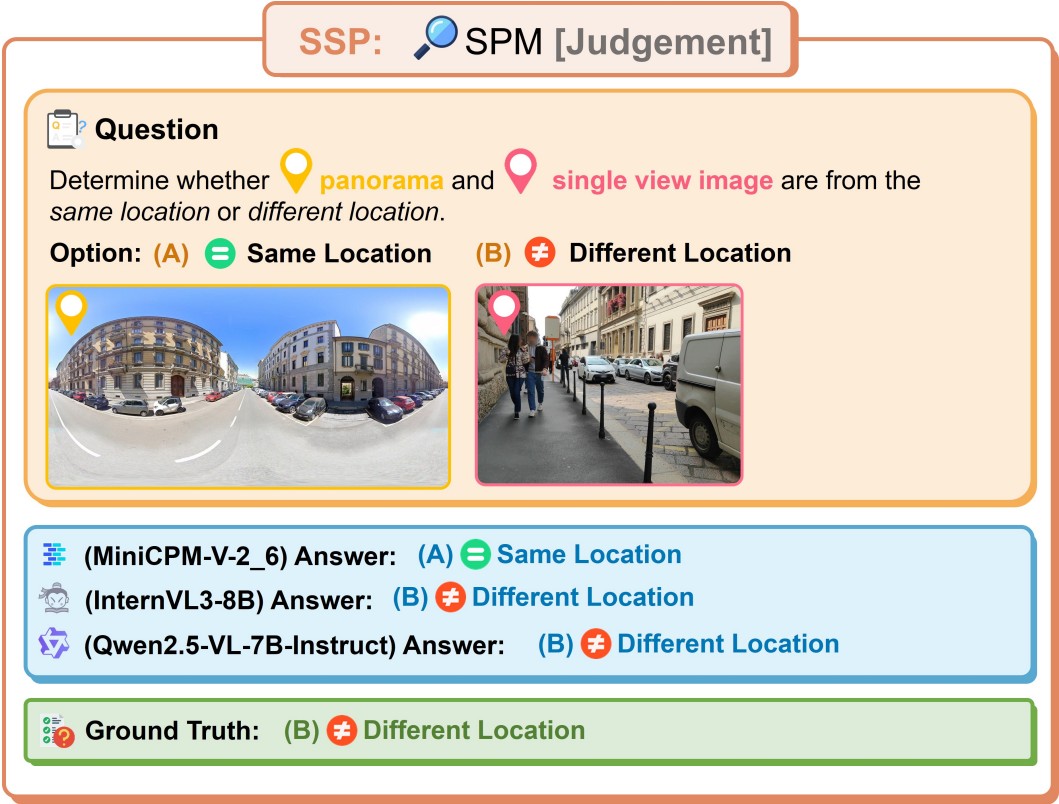

Figure 14: A question case of the **Single-to-Pano Matching(SPM)** task in UrbanFeel responses from *MiniCPM-V-2_6*, *InternVL3-8B*, *Qwen2.5-VL-7B-Instruct*

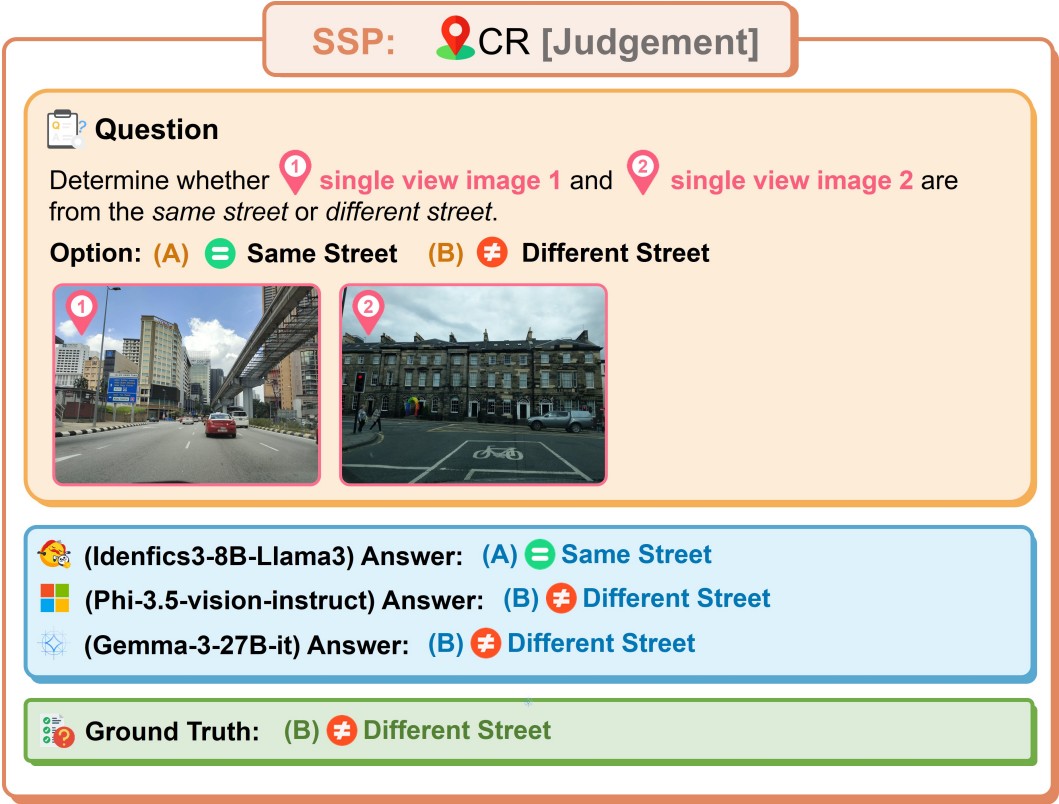

Figure 15: A question case of the **Co-location Recognition(CR)** task in UrbanFeel responses from *Idenfics3-8B-Llama3*, *Phi-3.5-vision-instruct*, *Gemma-3-27B-it*

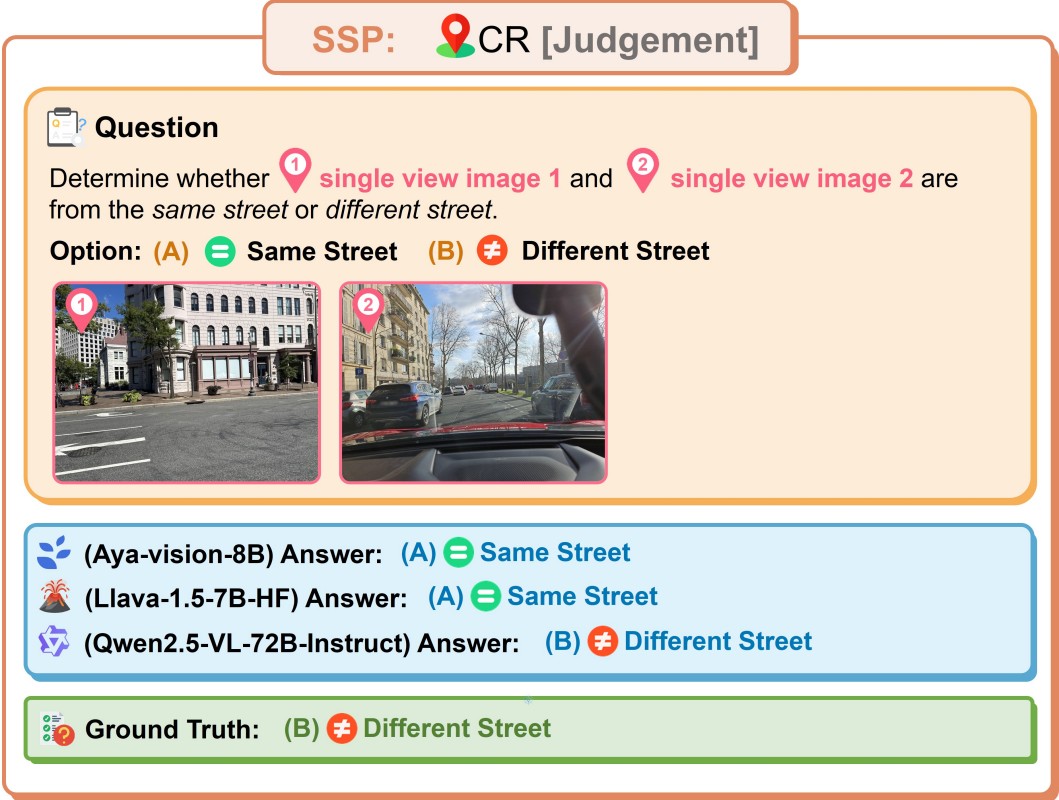

Figure 16: A question case of the **Co-location Recognition(CR)** task in UrbanFeel responses from *Aya-vision-8B*, *Llava-1.5-7B-HF*, *Qwen2.5-VL-72B-Instruct*

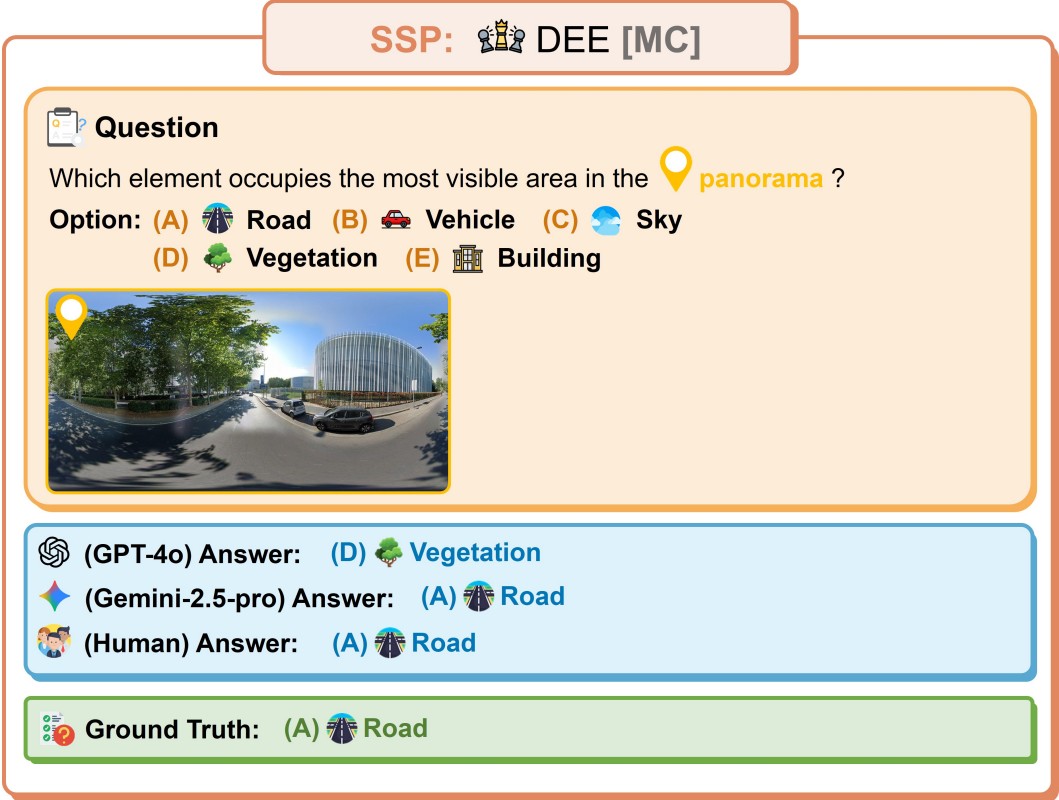

Figure 17: A question case of the **Dominant Element Extraction(DEE)** task in UrbanFeel responses from *GPT-4o*, *Gemini-2.5-pro*, *Human*

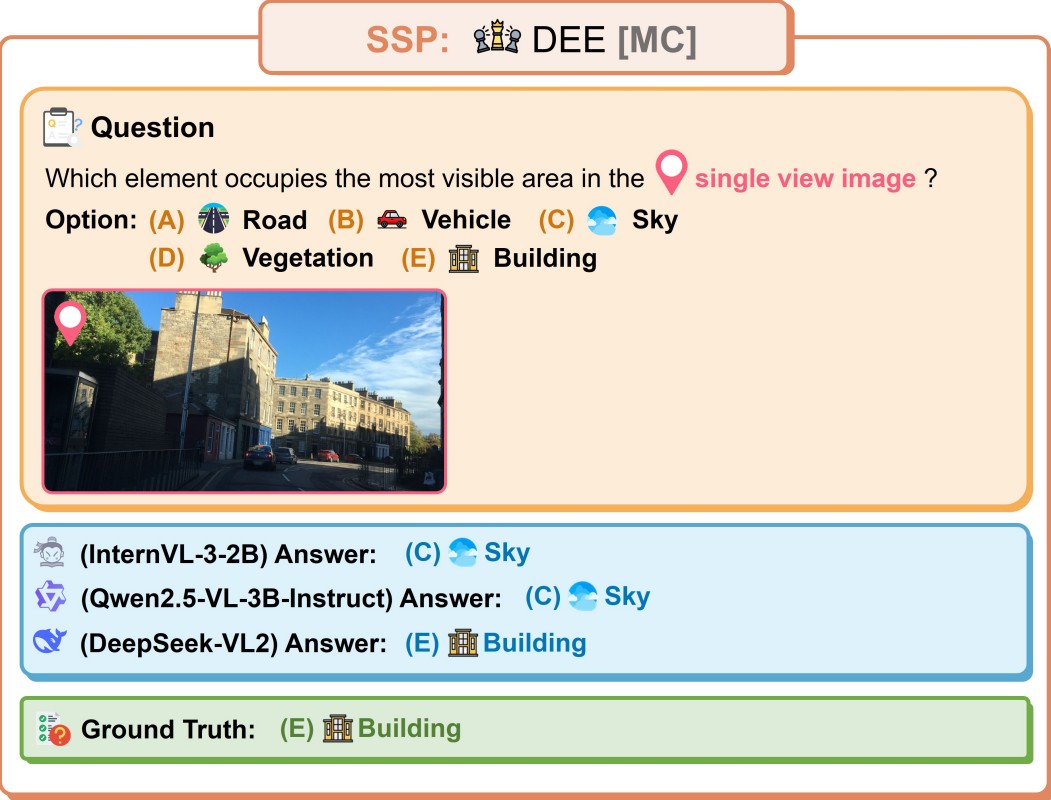

Figure 18: A question case of the **Dominant Element Extraction(DEE)** task in UrbanFeel responses from *InternVL-3-2B*, *Qwen2.5-VL-3B-Instruct*, *DeepSeek-VL2*

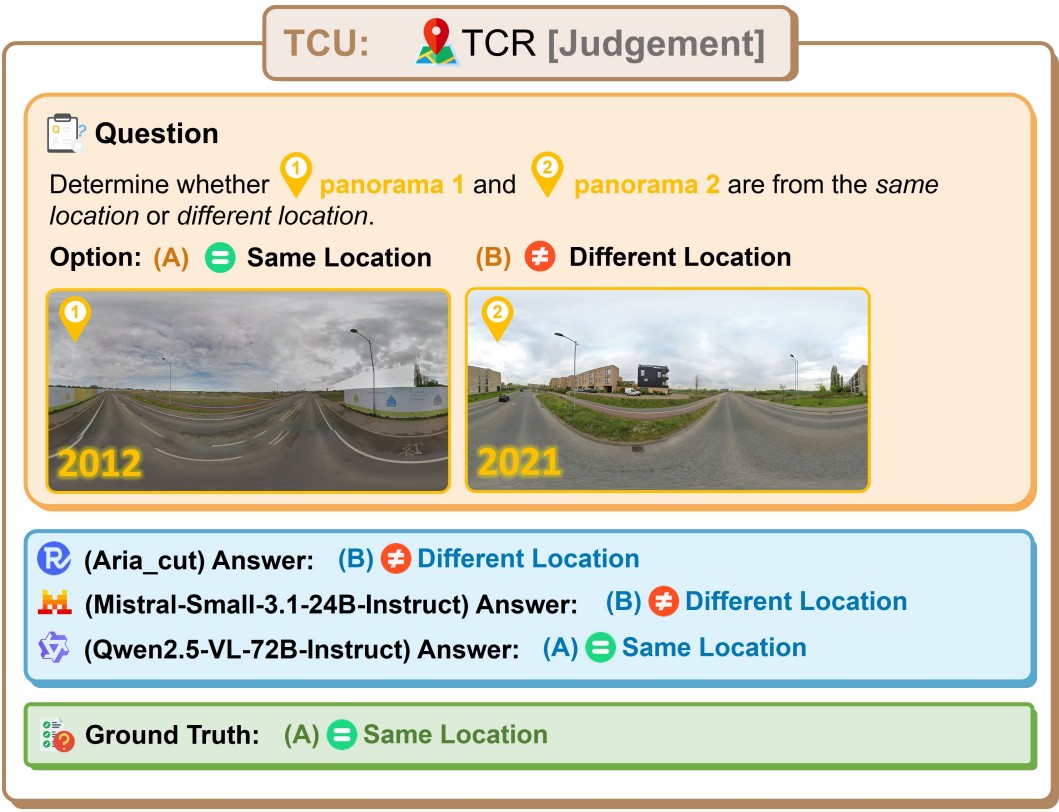

Figure 19: A question case of the **Temporal Co-location Recognition(TCR)** task in UrbanFeel responses from *Aria_cut*, *Mistral-Small-3.1-24B-Instruct*, *Qweb2.5-VL-72B-Instruct*

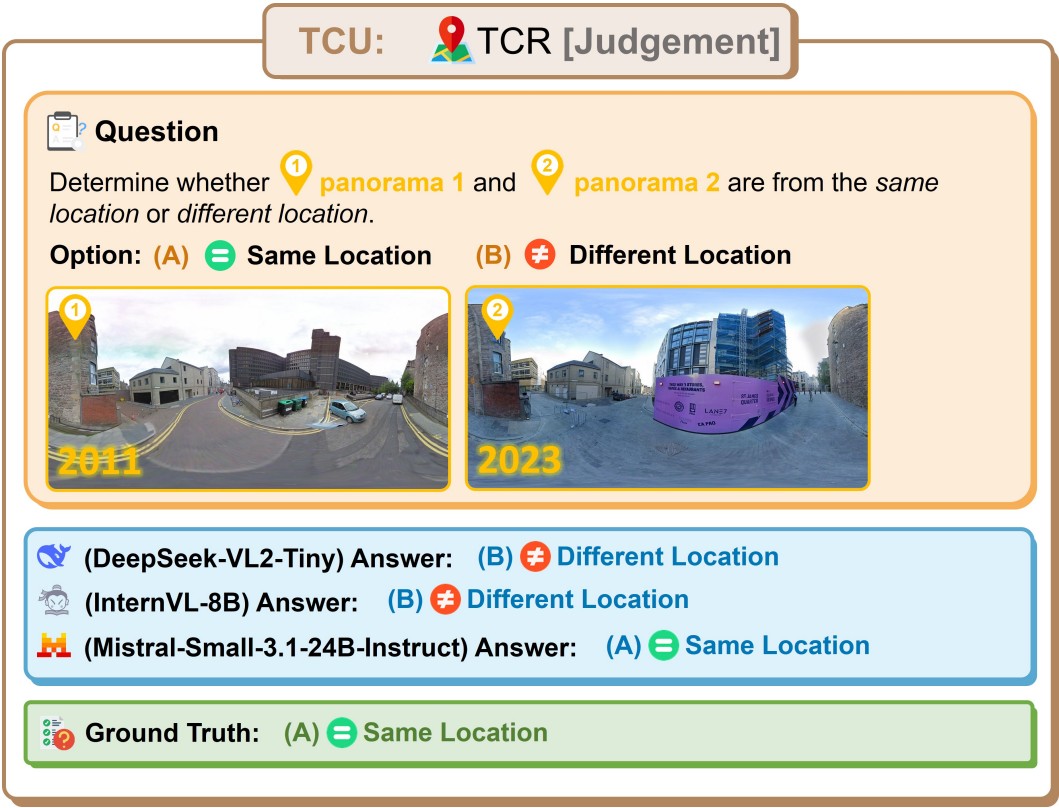

Figure 20: A question case of the **Temporal Co-location Recognition(TCR)** task in UrbanFeel responses from *DeepSeek-VL2-Tiny*, *InternVL-8B*, *Mistral-Small-3.1-24B-Instruct*

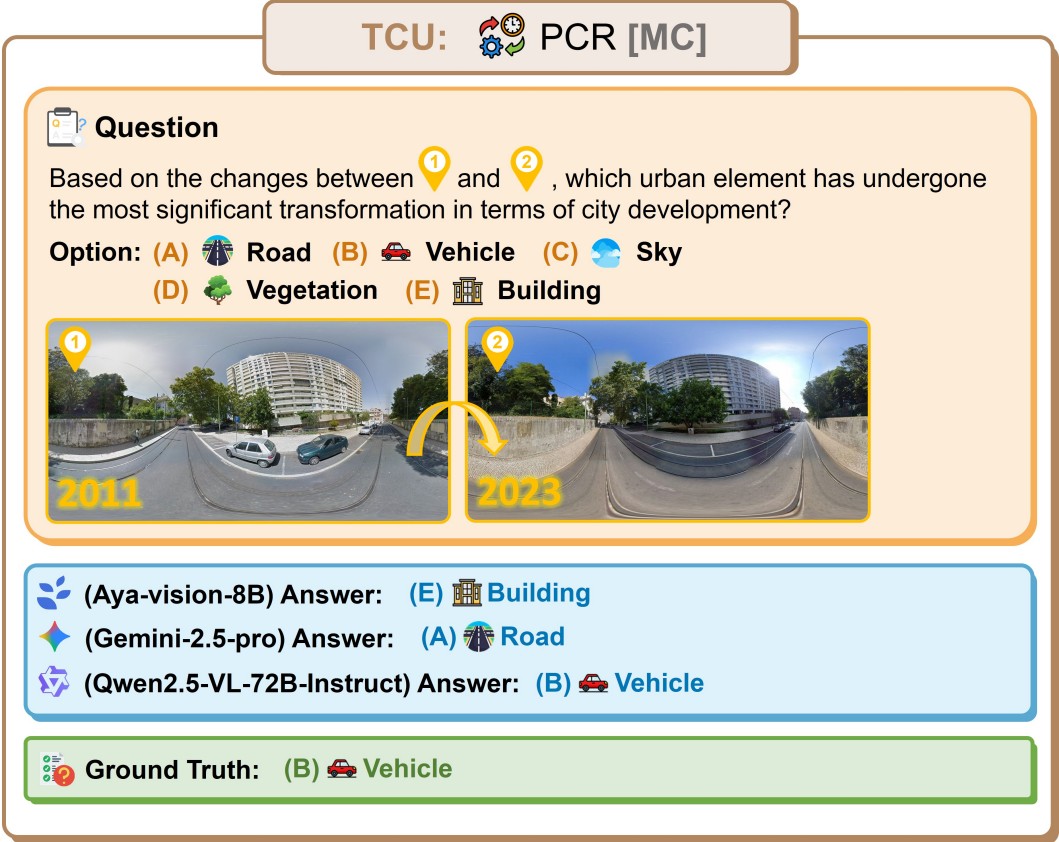

Figure 21: A question case of the **Pixel-level Change Recognition(PCR)** task in UrbanFeel responses from *Aya-vision-8B*, *Gemini-2.5-pro*, *Qwen2.5-VL-72B-Instruct*

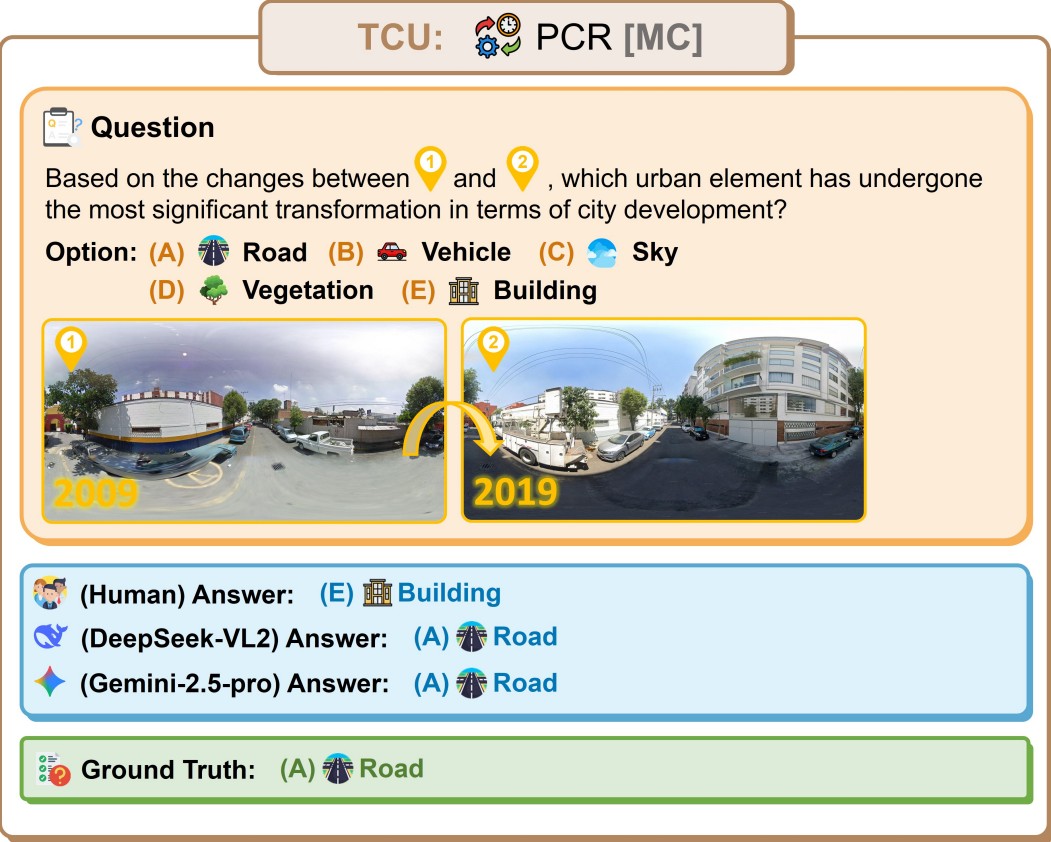

Figure 22: A question case of the **Pixel-level Change Recognition(PCR)** task in UrbanFeel responses from *Human*, *DeepSeek-VL2*, *Gemini-2.5-pro*

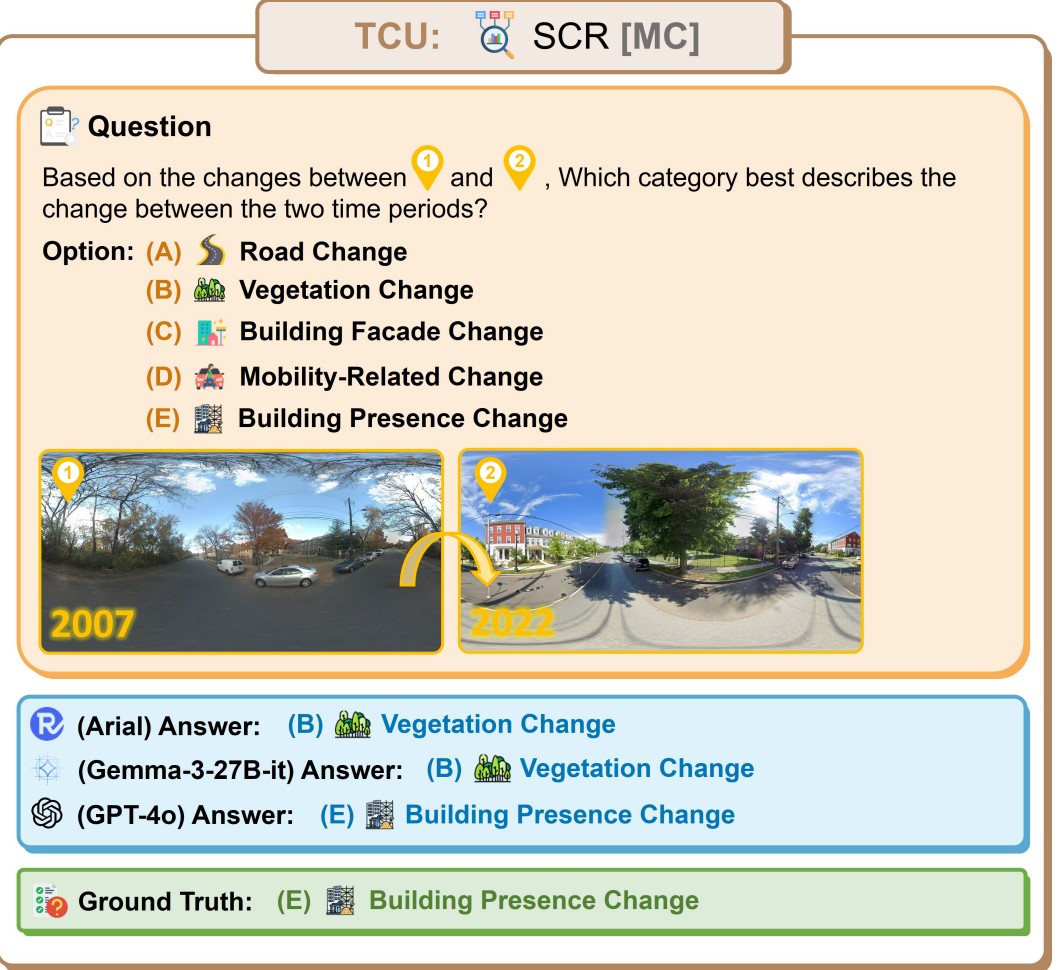

Figure 23: A question case of the **Scene-level Change Recognition(SCR)** task in UrbanFeel responses from *Arial*, *Gemma-3-27B-it*, *GPT-4o*

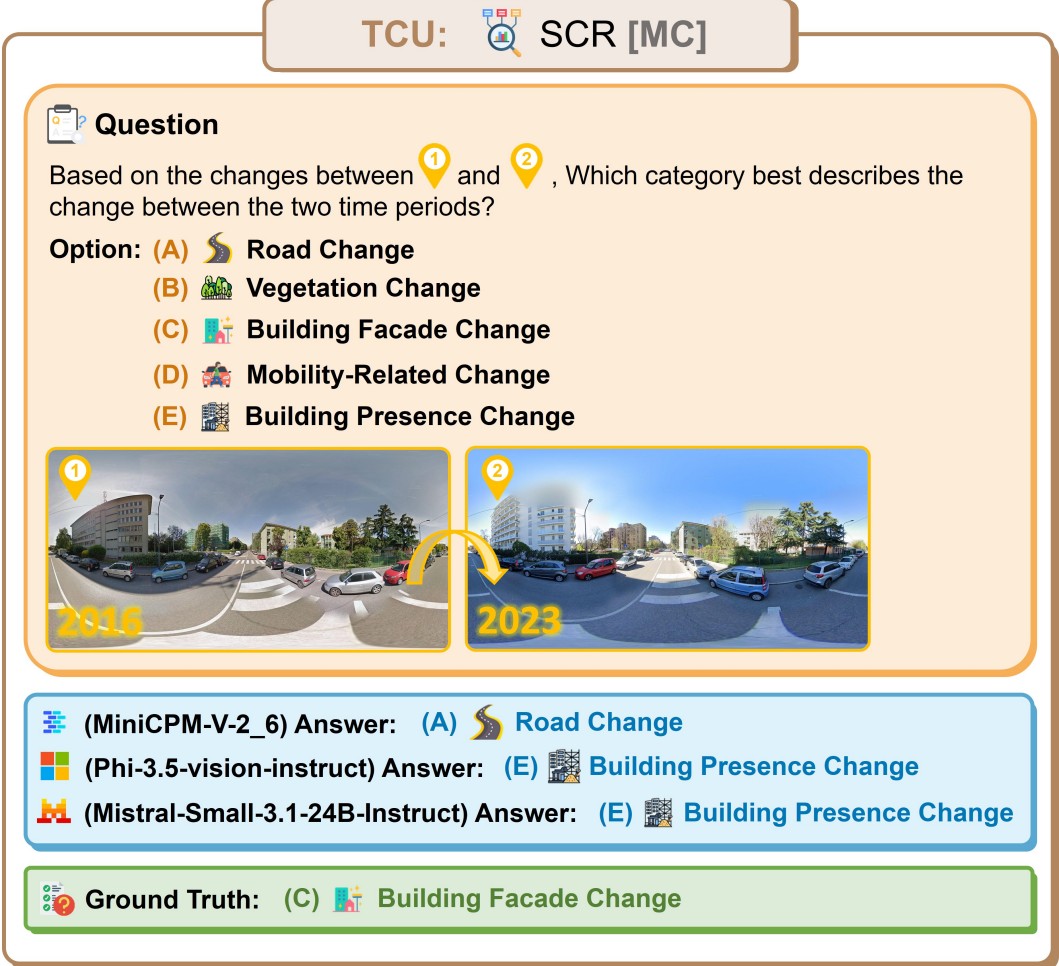

Figure 24: A question case of the **Scene-level Change Recognition(SCR)** task in UrbanFeel responses from *MiniCPM-V-2_6*, *Phi-3.5-vision-instruct*, *Mistral-Small-3.1-24B-Instruct*

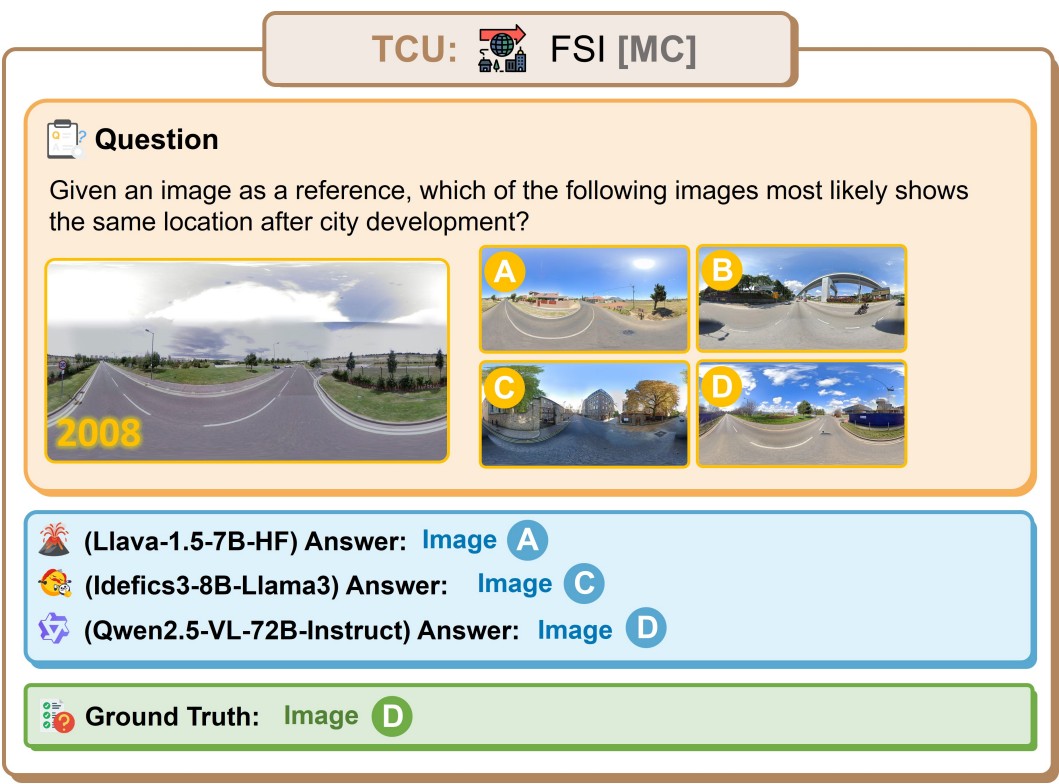

Figure 25: A question case of the **Future Scene Identification(FSI)** task in UrbanFeel responses from *Llava-1.5-7B-HF*, *Idenfics3-8B-Llama3*, *Qwen2.5-VL-72B-Instruct*

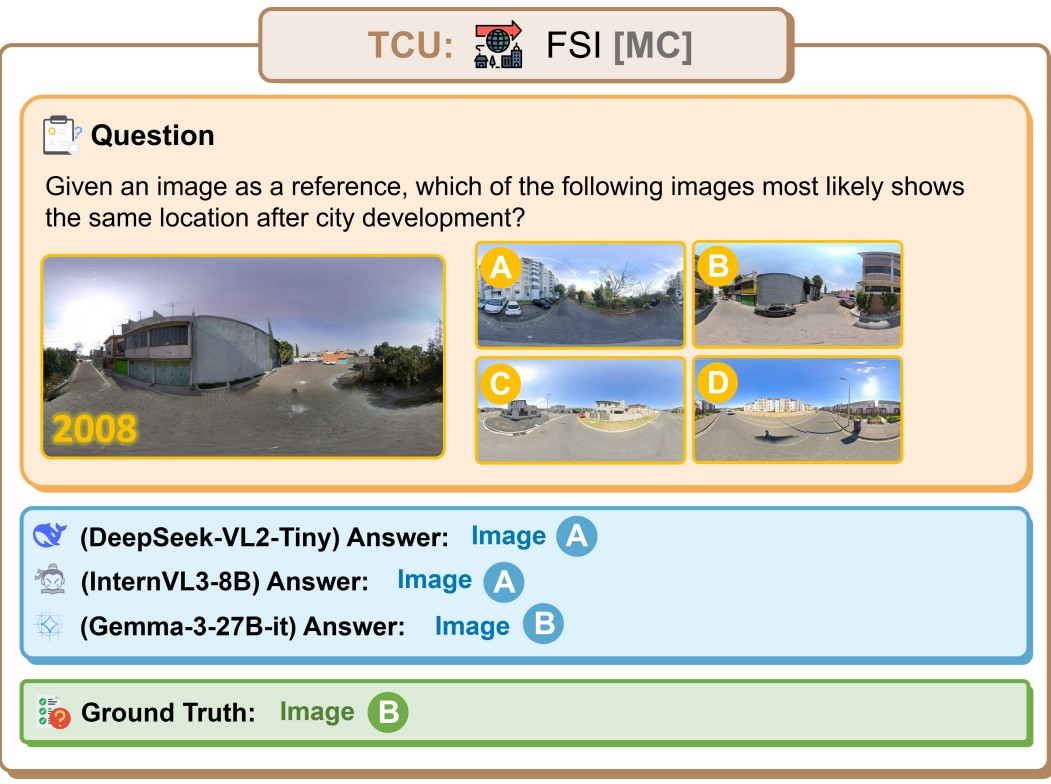

Figure 26: A question case of the **Future Scene Identification(FSI)** task in UrbanFeel responses from *DeepSeek-VL2-Tiny*, *InternVL3-8B*, *Gemma-3-27B-it*

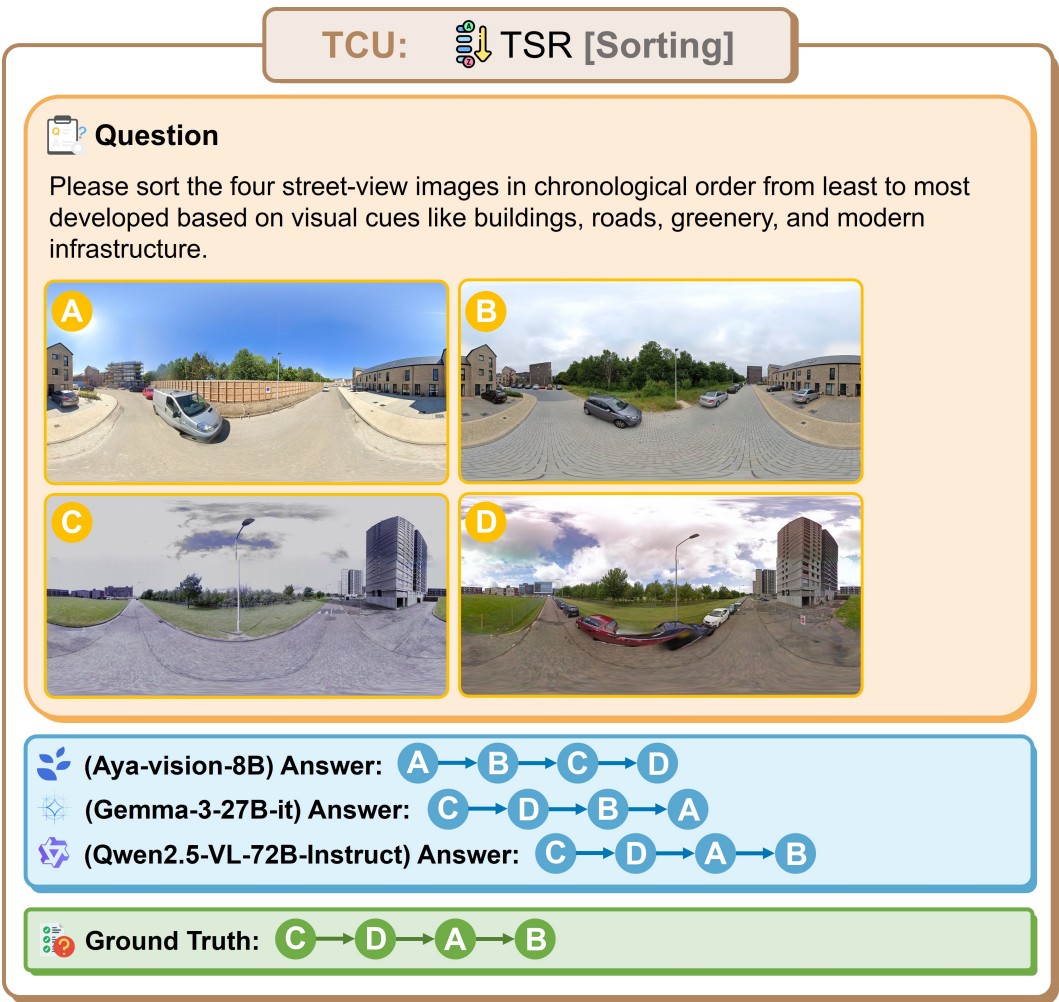

Figure 27: A question case of the **Temporal-Sequence Reasoning(TSR)** task in UrbanFeel responses from *Aya-vision-8B*, *Gemma-3-27B-it*, *Qwen2.5-VL-72B-Instruct*

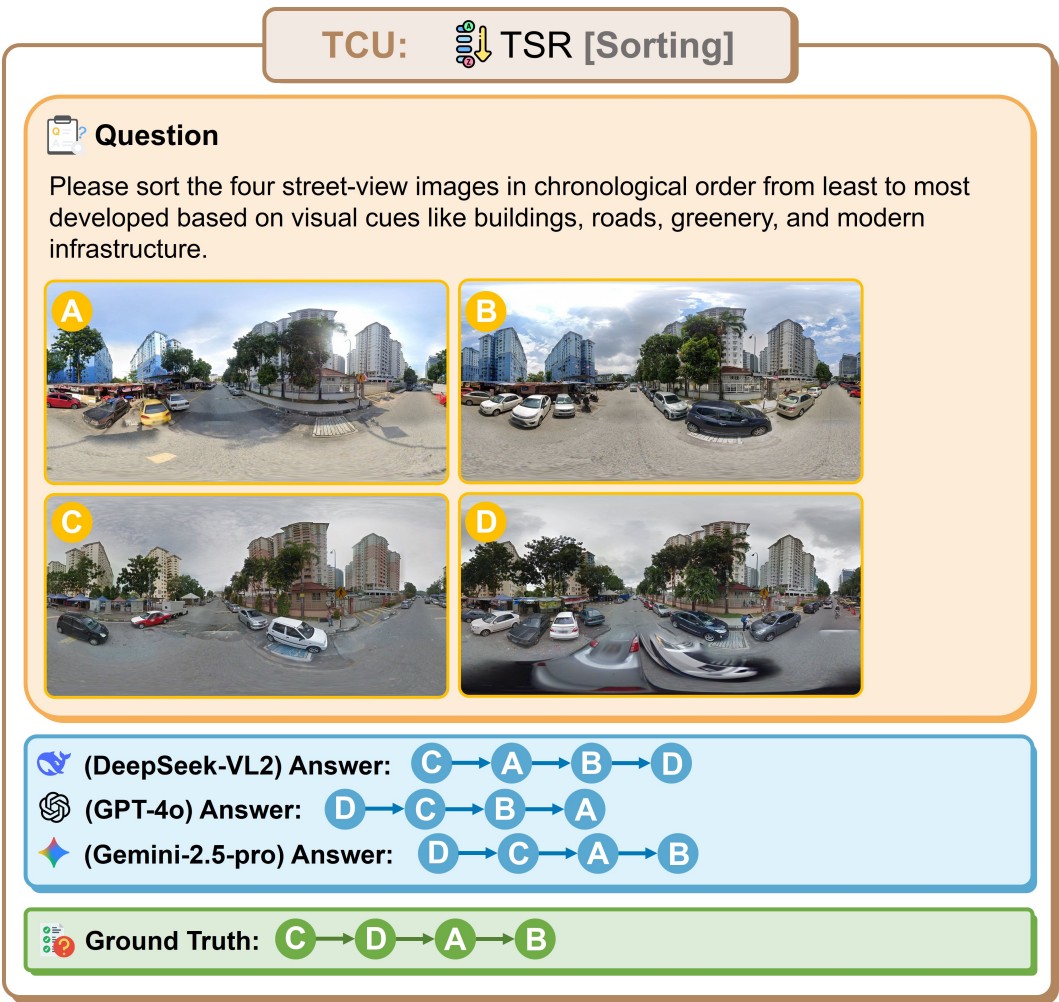

Figure 28: A question case of the **Temporal-Sequence Reasoning(TSR)** task in UrbanFeel responses from *DeepSeek-VL2*, *GPT-4o*, *Gemini-2.5-pro*

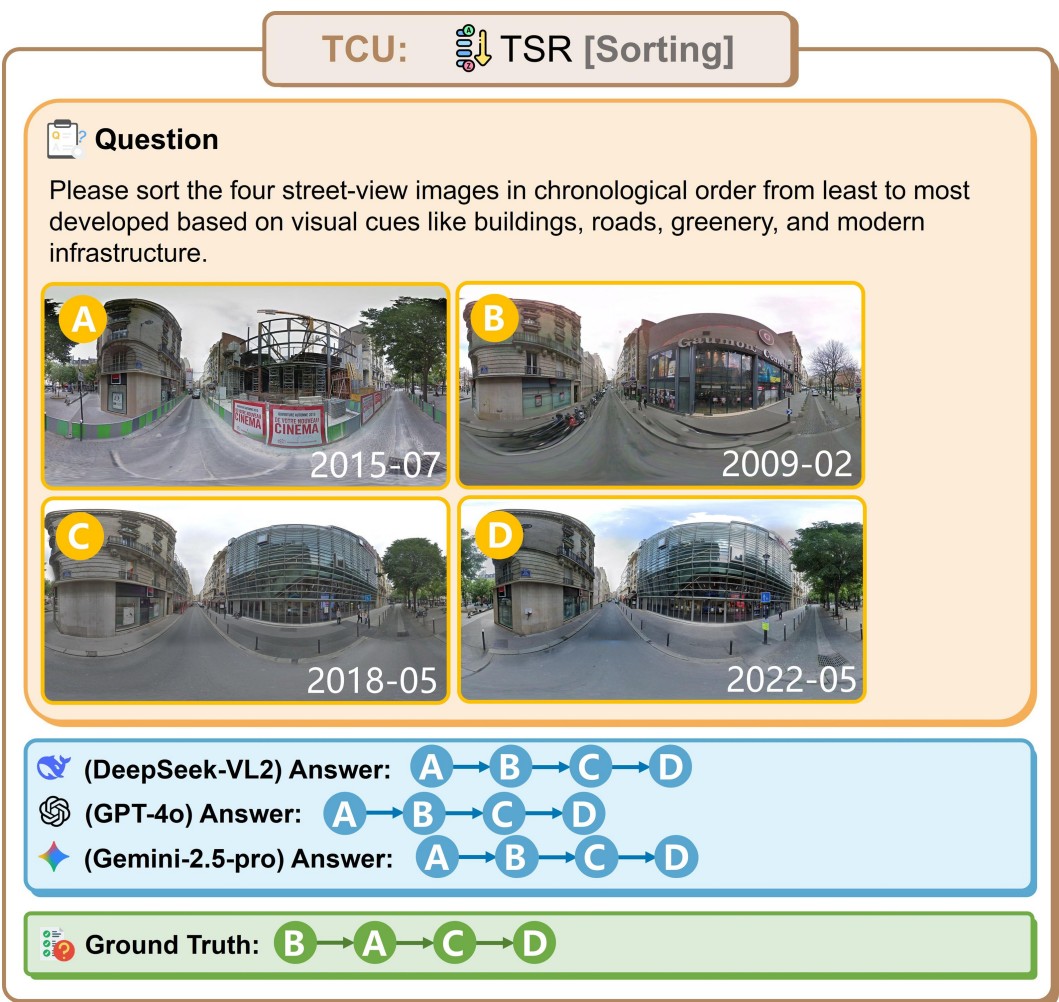

Figure 29: A question case of the **Temporal-Sequence Reasoning(TSR)** task in UrbanFeel responses from *DeepSeek-VL2*, *GPT-4o*, *Gemini-2.5-pro*

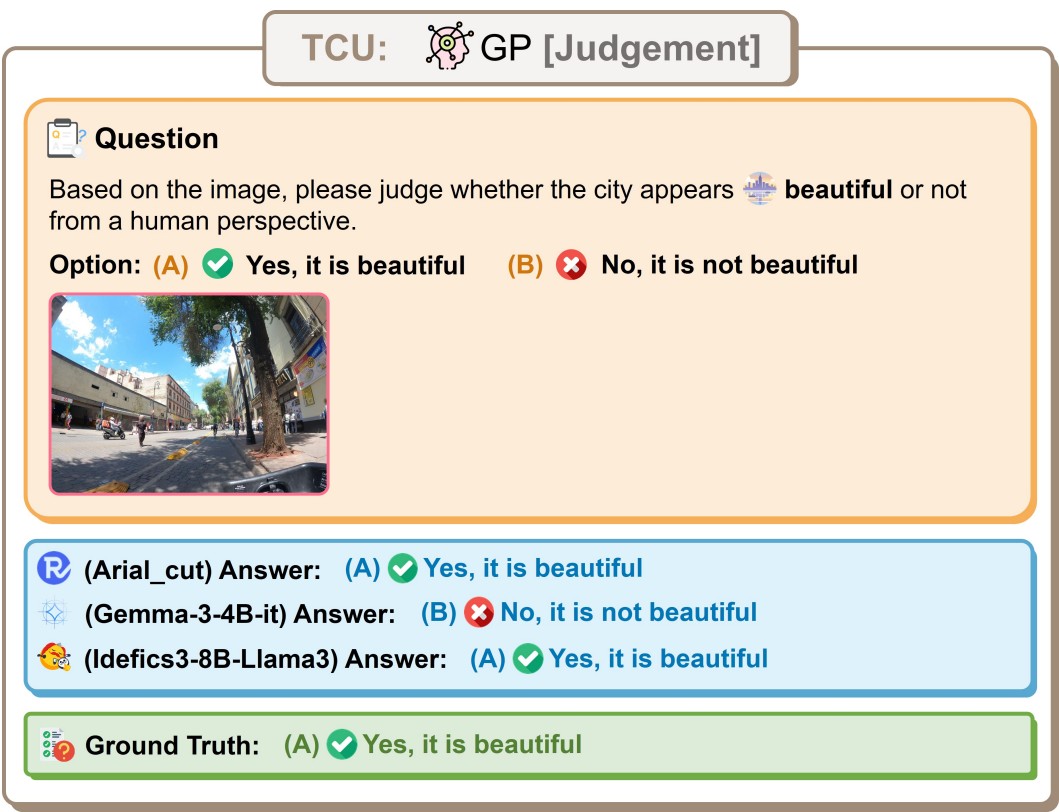

Figure 30: A question case of the **Global Perception(GP)** task in UrbanFeel responses from *Arial_cut*, *Gemma-3-4B-it*, *Idenfics3-8B-Llama3*

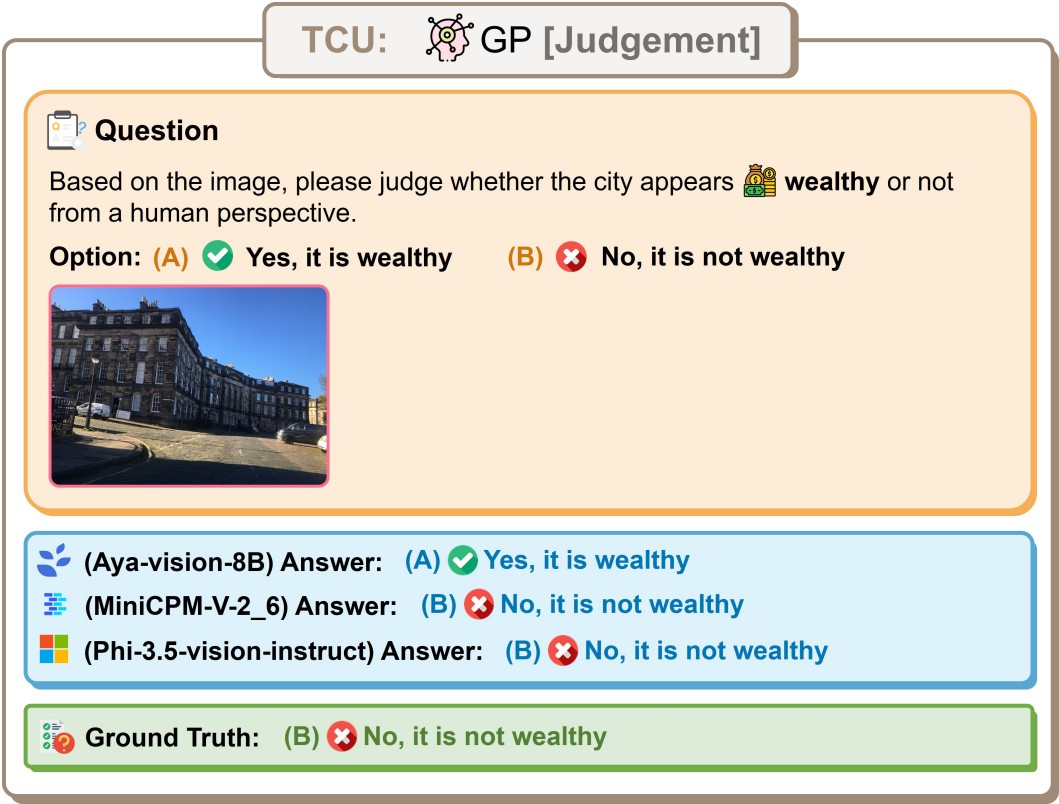

Figure 31: A question case of the **Global Perception(GP)** task in UrbanFeel responses from *Aya-vision-8B*, *MiniCPM-V-2_6*, *Phi-3.5-vision-instruct*

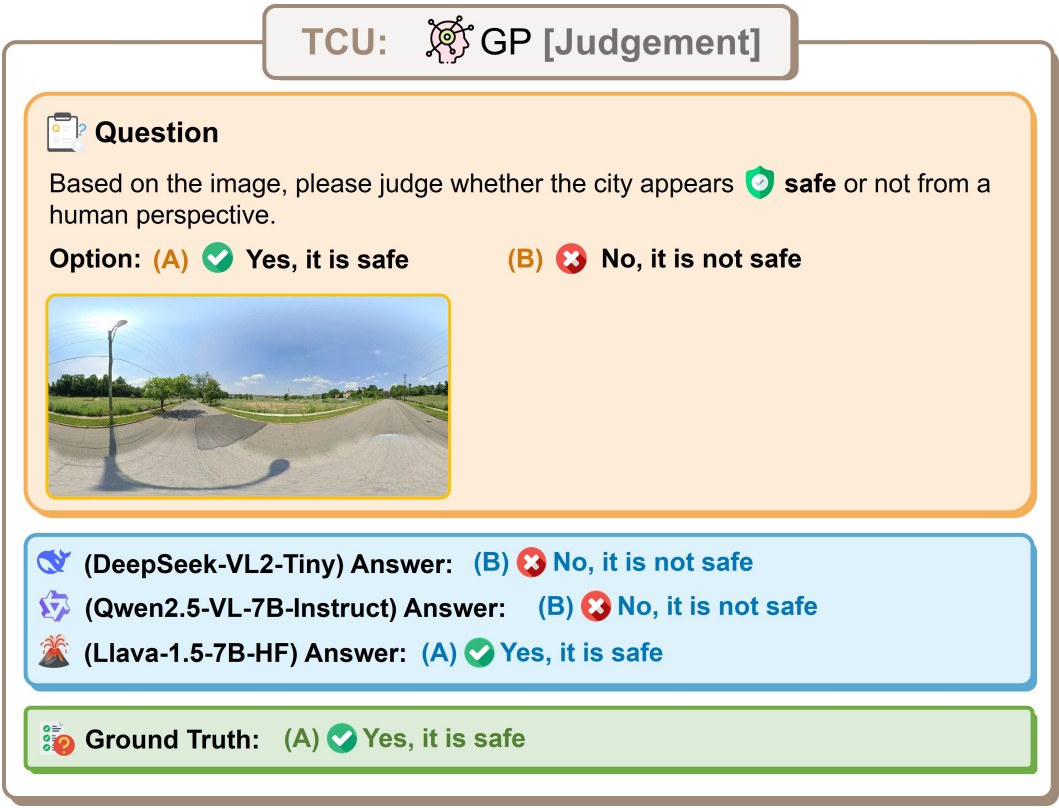

Figure 32: A question case of the **Global Perception(GP)** task in UrbanFeel responses from *DeepSeek-VL2-Tiny*, *Qwen2.5-VL-7B-Instruct*, *Llava-1.5-7B-HF*

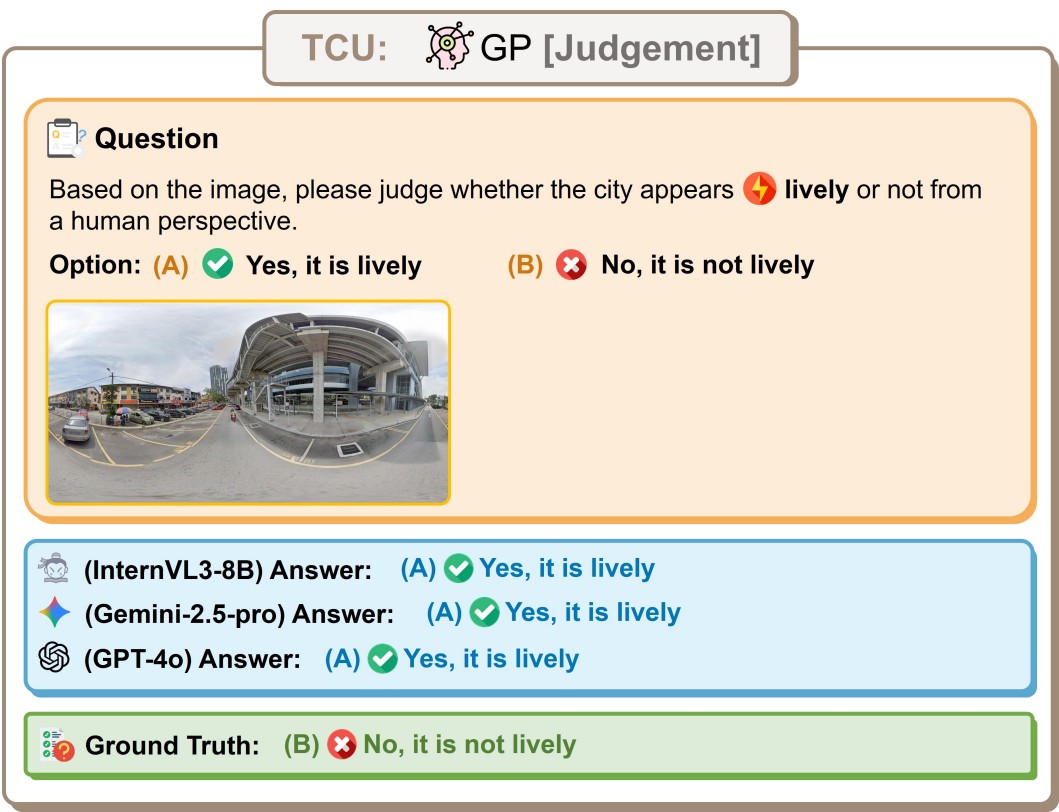

Figure 33: A question case of the **Global Perception(GP)** task in UrbanFeel responses from *InternVL3-8B*, *Gemini-2.5-pro*, *GPT-4o*

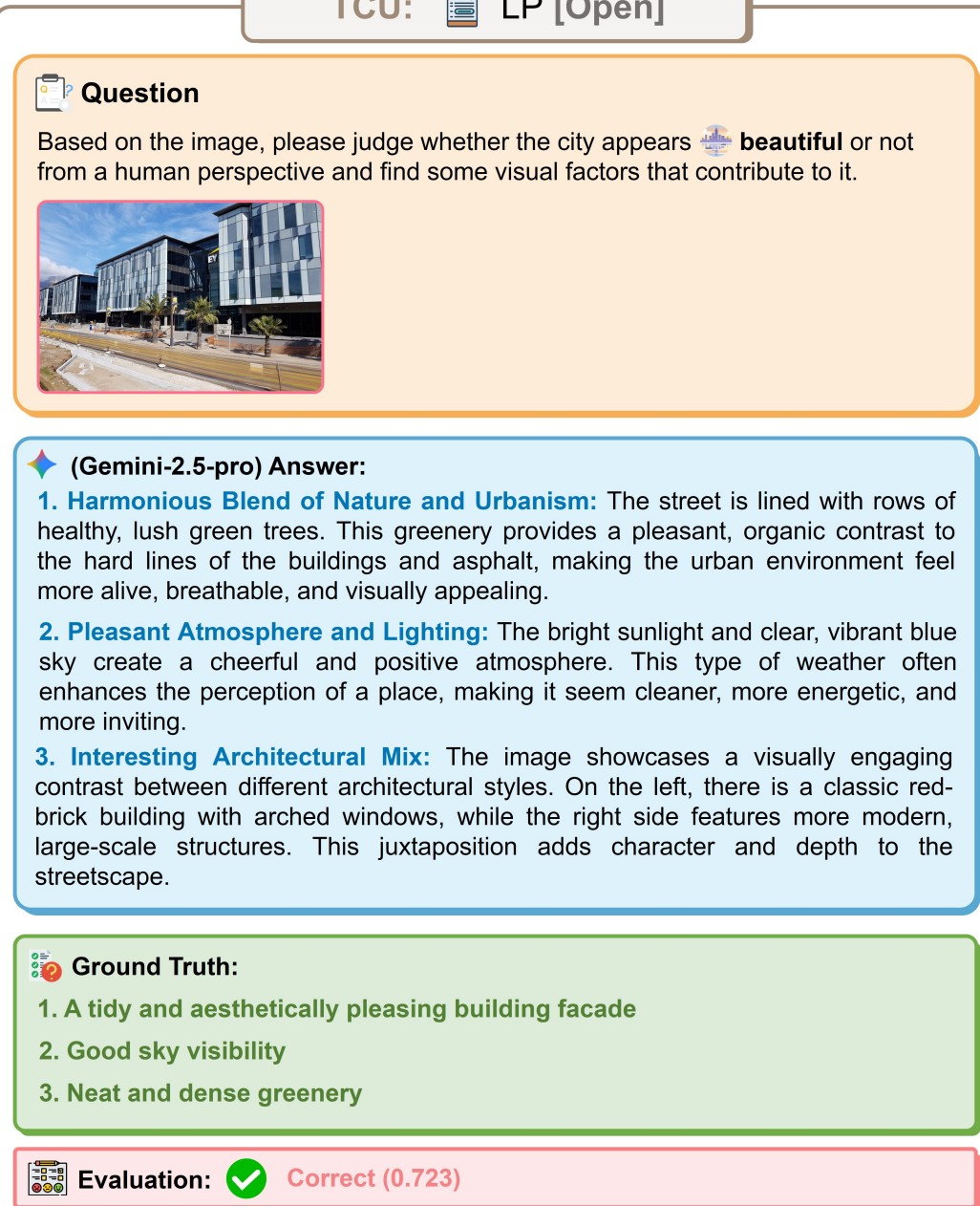

Figure 34: A question case of the **Local Perception(LP)** task in UrbanFeel responses from *Gemini-2.5-pro*

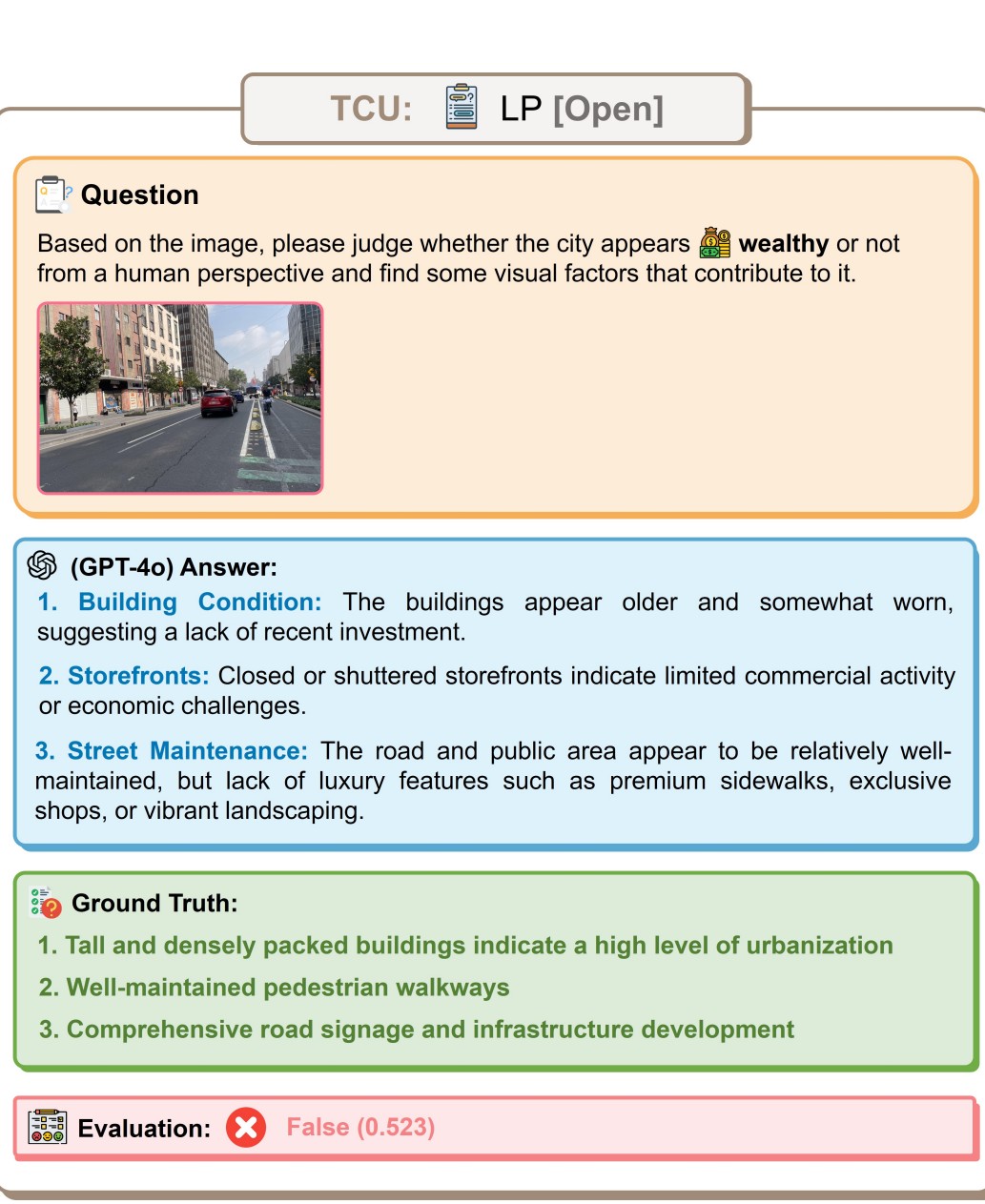

Figure 35: A question case of the **Local Perception(LP)** task in UrbanFeel responses from *GPT-4o*

**TCU:** 📋 LP **[Open]**

### 📋 Question

Based on the image, please judge whether the city appears 🛡 **safe** or not from a human perspective and find some visual factors that contribute to it.

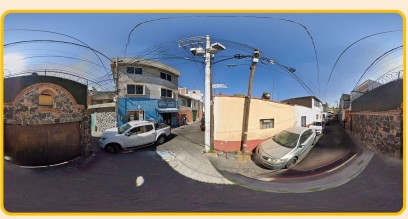

### 🐋 (DeepSeek-VL2) Answer:

**1.** The street appears narrow and congested, which can make it difficult for pedestrians to navigate safely

**2.** There are multiple cars parked along the street, which could obstruct visibility and create potential hazards for both drivers and pedestrians.

**3.** The road surface shows some wear, lacks lane markings, and the road signs are unclear, creating potential driving hazards

### 📋 Ground Truth:

**1. The high walls on both sides of the road obstruct visibility, posing safety hazards**

**2. The streetlight poles appear somewhat outdated, and the distribution of electrical wires is chaotic**

**3. The road surface shows some wear, lacks lane markings, and the road signs are unclear, creating potential driving hazards**

📋 **Evaluation:** ✅ **Correct (0.691)**

Figure 36: A question case of the **Local Perception(LP)** task in UrbanFeel responses from *DeepSeek-VL2*

**TCU:** 📋 LP **[Open]**

📋 **Question**

Based on the image, please judge whether the city appears ⚡ **lively** or not from a human perspective and find some visual factors that contribute to it.

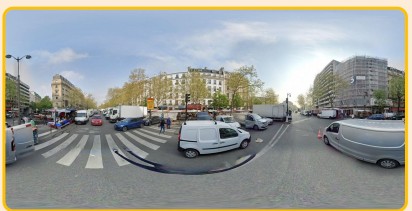

🔷 **(Qwen2.5-VL-72B-Instruct) Answer:**

**1. Presence of People and Activity:** There are several people visible in the image, walking across the crosswalk and along the sidewalks. This indicates active pedestrian movement, contributing to a lively atmosphere.

**2. Market Stalls and Vendors:** The image shows market stalls with vendors and customers, suggesting a bustling market scene. The presence of these stalls adds to the vibrancy and activity of the area.

**3. Traffic and Vehicles:** There are multiple vehicles, including cars, vans, and trucks, parked and moving along the street. This indicates a level of traffic activity, which is a sign of a busy and lively urban environment.

📋 **Ground Truth:**

**1. There are many pedestrians**

**2. There are many vehicles**

**3. There are many shops, and there are also many parked vehicles around, creating a lively atmosphere**

**4. The buildings consist of multiple residential blocks**

📋 **Evaluation:** ✅ **Correct (0.735)**

Figure 37: A question case of the **Local Perception(LP)** task in UrbanFeel responses from *Qwen2.5-VL-72B-Instruct*

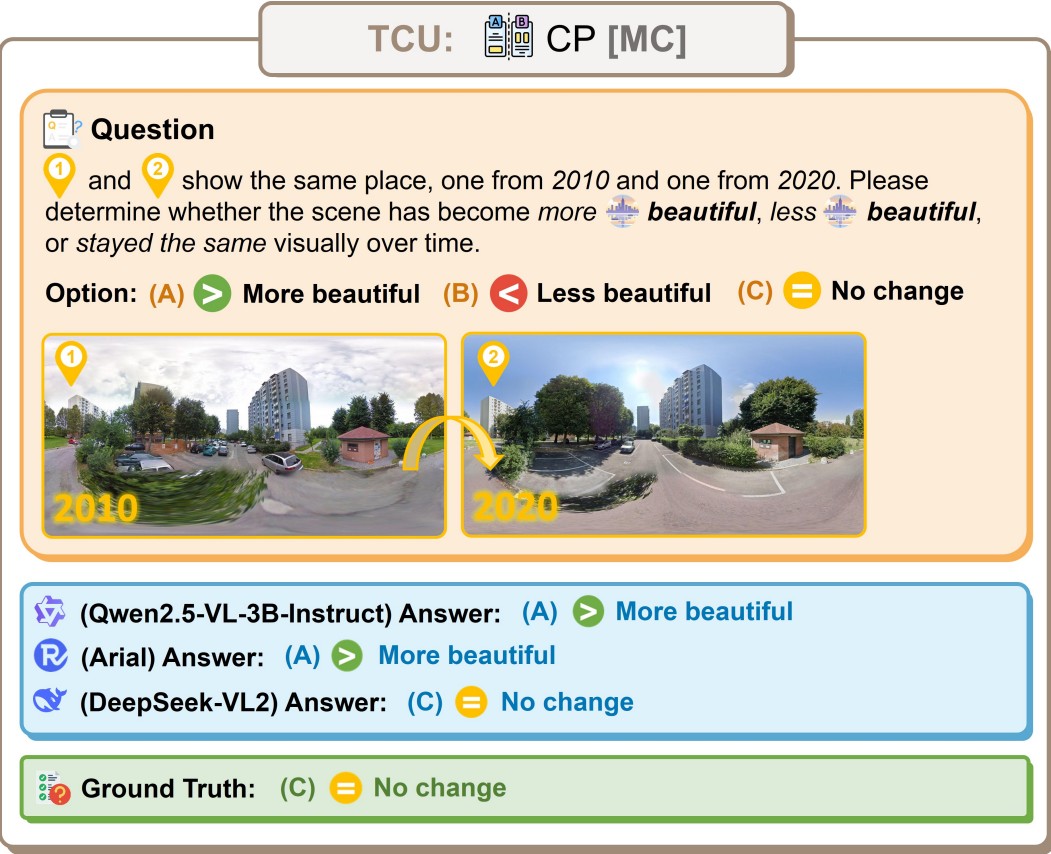

Figure 38: A question case of the **Comparative Perceptual analysis(CP)** task in UrbanFeel responses from *Qwen2.5-VL-3B-Instruct*, *Arial*, *DeepSeek-VL2*

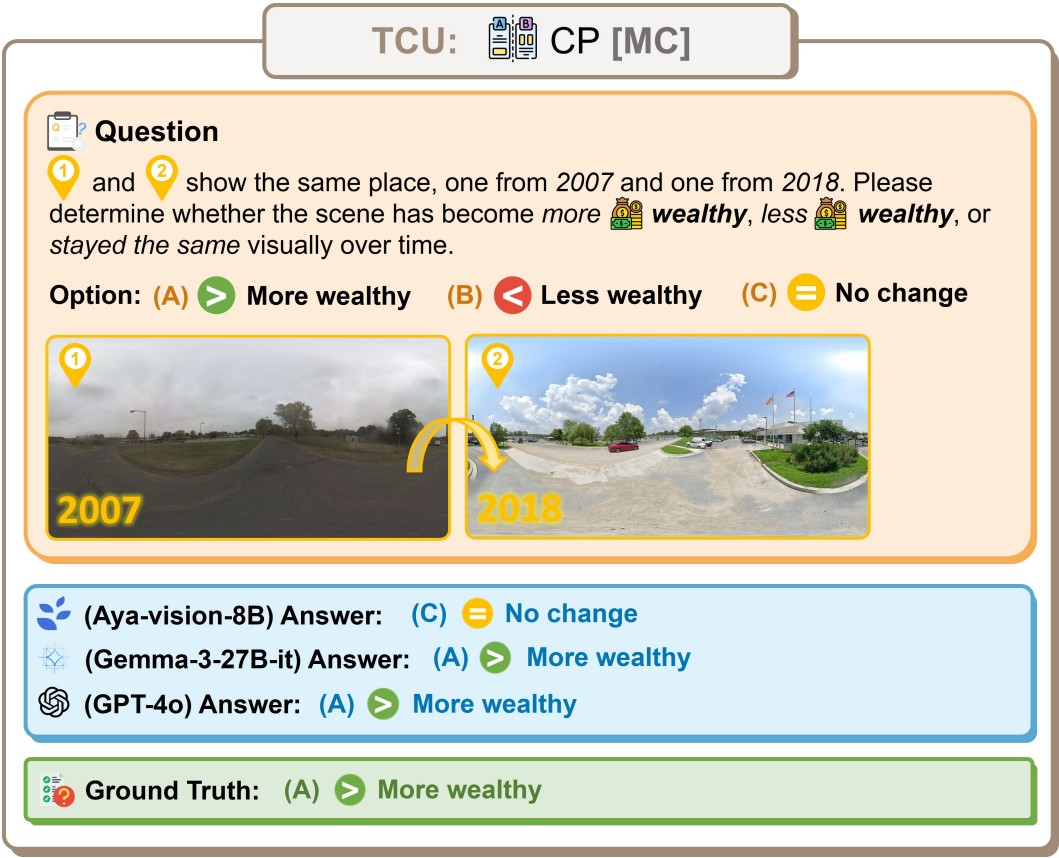

Figure 39: A question case of the **Comparative Perceptual analysis(CP)** task in UrbanFeel responses from *Aya-vision-8B*, *Gemma-3-27B-it*, *GPT-4o*

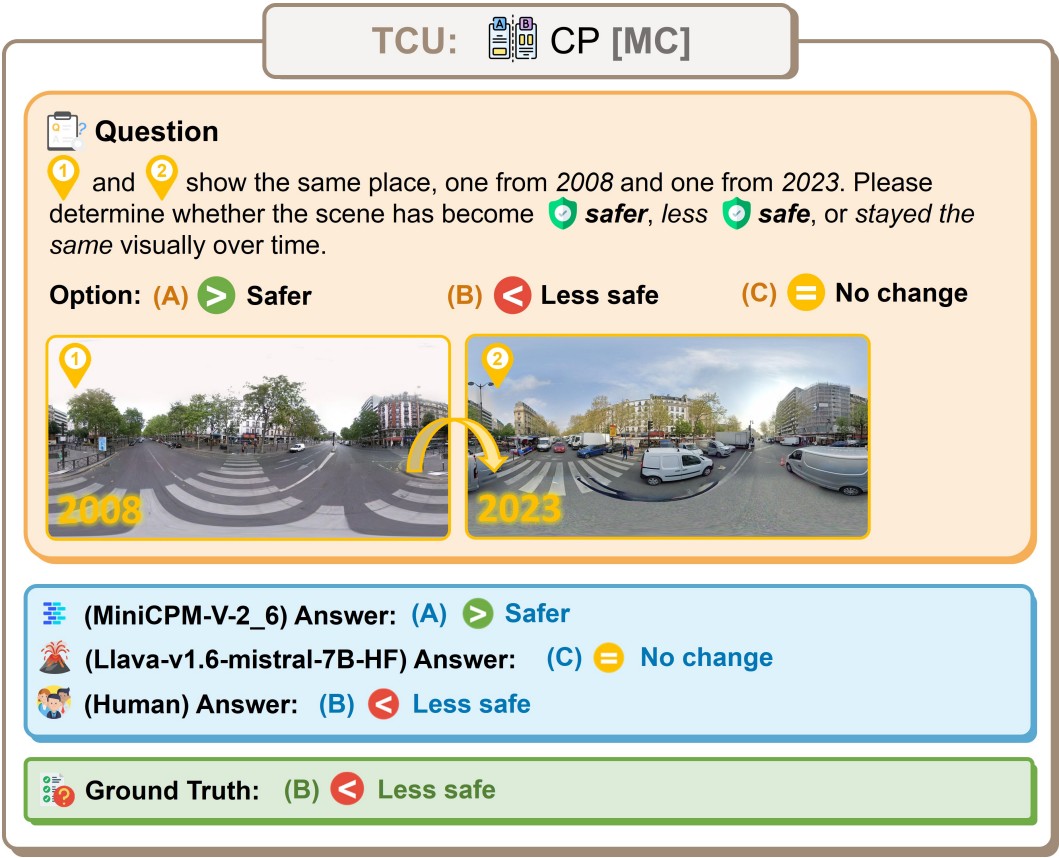

Figure 40: A question case of the **Comparative Perceptual analysis(CP)** task in UrbanFeel responses from *MiniCPM-V-2_6*, *Llava-v1.6-mistral-7B-HF*, *Human*

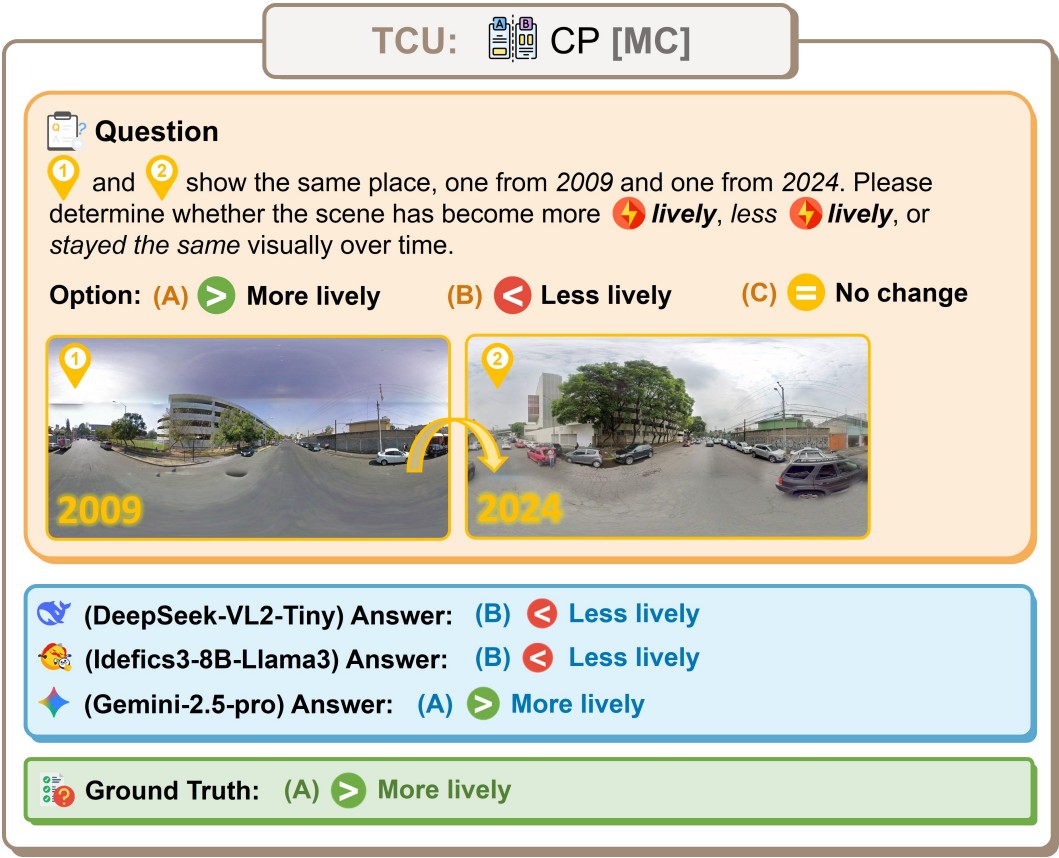

Figure 41: A question case of the **Comparative Perceptual analysis(CP)** task in UrbanFeel responses from *DeepSeek-VL2-Tiny*, *Idenfics3-8B-Llama3*, *Gemini-2.5-pro*

