# OpenReview forum: "UrbanFeel：A Comprehensive Benchmark for Temporal and Perceptual Understanding of City Scenes through Human Perspective"
_ICLR.cc/2026/Conference — ICLR 2026 Poster_

### Official Review · Reviewer_XNMk · 2025-10-19

**Soundness:** 4
**Presentation:** 4
**Contribution:** 3
**Rating:** 6
**Confidence:** 4

**Summary:**

This paper constructs a comprehensive evaluation benchmark named UrbanFeel with a rich dataset containing 14.3k question-answer pairs to assess MLLM's understanding of temporal changes in urban environments and human subjective perceptions of urban settings. This benchmark comprises three major categories of tasks: Static Scene Perception, Temporal Change Perception, and Subjective Environmental Perception. This paper confirms through extensive experimentation the performance of current mainstream MLLMs on this benchmark and their gap relative to human experts.

**Strengths:**

1. Organizing urban imagery on a temporal scale offers a novel perspective for observation. On one hand, it extends beyond conventional benchmark tasks to further test the comprehensive temporal and visual understanding and reasoning capabilities of MLLMs. On the other hand, it provides datasets and fresh insights for research on urban development.
2. Based on temporal image data, this paper proposes targeted evaluation methods such as comparison and temporal sorting, effectively supporting the benchmark process.
3. The text and figures are easy to follow.

**Weaknesses:**

I appreciate the research perspective of this work, but it still has some specific issues. If all the following issues are properly addressed, I will adjust the rating to "Accept".
1. This paper does not sufficiently explore the impact and value of the proposed benchmark, which may limit its future popularity.
2. Some figure descriptions are unclear and confusing. For example, the meanings of the numbers in Figure 2 (a) are very confusing. What is the relationship between 1100 and 2200 in the SCR subtask? And what are the meanings of yellow and grey colors? And what is the meaning of the abbr "str" in the "View Type" part? I hope the author will carefully examine every detail of these large and complex images to ensure they are self-descriptive.
3. The details of the experimental evaluation require specific clarification, particularly regarding the accuracy metrics or calculation formulas used for each subtask, along with necessary parameters such as thresholds. Currently, I do not appear to have seen relevant descriptions in the paper.

**Questions:**

1. Can the benchmark or design approach proposed in this paper be applied to research beyond smart cities, such as topics related to sustainable development?
2. Cities may undergo a process of decline alongside development. Can MLLMs effectively distinguish such scenarios? Or do MLLMs always assume that more prosperous neighborhoods are newer?
3. What is your open-source plan?
4. Would incorporating reasoning processes or using simple prompts to guide MLLMs in performing basic reasoning before answering questions enhance their performance?

---

> ### Author Response · Authors · 2025-11-22
> **Response to Weakness 1, Question 1 and  Weakness 2**
>
> ## Potential impact and value of UrbanFeel (Weakness 1 & Question 1)
> We appreciate the reviewer’s insightful comments concerning the broader impact, generalization potential, and cross-domain value of the proposed benchmark. We have further emphasized these cross-domain applications in the **introduction and conclusion** of the revised manuscript to address the reviewer’s valuable suggestions.
>
> UrbanFeel introduces three core properties that are largely absent from existing multimodal datasets: (1) long-term temporal reasoning over many years of real-world urban evolution; (2) spatial reasoning across cross-view data; and (3) the evaluation of human-centered subjective perception. These capabilities are not limited to urban studies. Instead, they correspond to fundamental competencies required by modern multimodal language models—temporal consistency, cross-view alignment, and perceptual understanding. Consequently, the design philosophy and technical framework of UrbanFeel naturally generalize to broader domains such as sustainability science, environmental assessment, and human–environment interaction. For example, **the temporal reasoning tasks inherently reflect a model’s capability to conduct multi-year analyses** of land-use change[1, 2] and infrastructure aging [3, 4] . In addition, **the current perceptual dimensions (beauty, safety, wealth, and liveliness) can be naturally extended to broader well-being indicators**, including walkability [5], environmental comfort [6, 7], and social equity [8]. Finally, the cross-view reasoning capabilities required by the benchmark closely align with the integration of multi-source sensing data employed in ecological environment assessment and human–environment interactions.
>
> In summary, although UrbanFeel is instantiated within an urban context, the multimodal capabilities it evaluates, including temporal reasoning, perceptual alignment, and cross-view consistency, are inherently domain-agnostic. Therefore, we believe that UrbanFeel can support a wide range of research areas, such as sustainable development, infrastructure monitoring, and socio-spatial analysis.
>
> [1] Fang, Fang, et al. "Spatial context-aware method for urban land use classification using street view images." ISPRS Journal of Photogrammetry and Remote Sensing 192 (2022): 1-12.
>
> [2] Liu, Yingjie, et al. "Physical urban change and its socio-environmental impact: Insights from street view imagery." Computers, Environment and Urban Systems 119 (2025): 102284.
>
> [3] Wu, Devin Yongzhao, and Jue Wang. "Modeling spatial and temporal urban environmental noise using street view imagery and machine learning." Computers, Environment and Urban Systems 121 (2025): 102327.
>
> [4] Dai, Yilong, et al. "Using computer vision and street view images to assess bus stop amenities." Computers, Environment and Urban Systems 117 (2025): 102254.
>
> [5] Lu, Yi, and Hui-Mei Chen. "Using google street view to reveal environmental justice: Assessing public perceived walkability in macroscale city." Landscape and Urban Planning 244 (2024): 104995.
>
> [6] Qin, Jie, et al. "Mapping high-resolution spatio-temporal patterns of pedestrian thermal comfort at different scales using street view imagery and deep learning." Sustainable Cities and Society 121 (2025): 106209.
>
> [7] Wu, Chao, et al. "Perceiving the fine-scale urban poverty using street view images through a vision-language model." Sustainable Cities and Society 123 (2025): 106267.
>
> [8] Kang, Yuhao, et al. "A review of urban physical environment sensing using street view imagery in public health studies." Annals of GIS 26.3 (2020): 261-275.
>
> ---
>
> ## Clarifying visual elements and figure design (Weakness 2)
>
> We sincerely appreciate your constructive comments. We apologize for the ambiguity in our original description of the visual elements, and to address this issue, we have revised the corresponding figures in the updated manuscript.
>
> Given the concerns raised by other reviewers regarding image clarity, and in order to more comprehensively present the full scope of UrbanFeel, we have split **the original Figure 2 into two separate figures** (now Figure 2 and Figure 3 in the revised version). In the updated Figure 2, **we have added additional visual legends to facilitate a clearer understanding of the numerical representations**.
>
> Regarding your comment on “str”, our original intention was to convey that **the model input includes multi-view images along with the corresponding textual prompts**. Considering the potential for reader misinterpretation, **we have removed this expression from Figure 2 in the revised manuscript**.
>
> We hope that these revisions will help readers obtain a **clearer and more accurate understanding** of the overall structure and content of UrbanFeel.

---

> ### Author Response · Authors · 2025-11-22
> **Response to Weakness 3 and Question 3**
>
> ## Detailed indicator calculation formula (Weakness 3)
>
> We fully understand your concerns regarding the experimental details, and we acknowledge that our original submission lacked sufficient clarification.
>
> To address this, we have added a dedicated subsection on evaluation metrics in Appendix B of the revised manuscript. In addition, we have included a detailed explanation of the thresholding strategy used for Sentence-BERT.
> Specifically, we adopt accuracy (ACC) as the evaluation metric, computed as:
> $$
> ACC = \frac{N_{\text{Correct Predictions}}}{N_{\text{Total Predictions}}}
> $$
> where *Total Predictions* denotes the total number of evaluated instances, and  *Correct Predictions* denotes the number of instances for which the model output satisfies the task-specific correctness criterion.
> For multiple-choice, ranking, and judgment tasks, this criterion corresponds to matching the objective ground truth. For open-ended questions, we consider a prediction correct if the Sentence-BERT similarity between the predicted answer $\hat{y}_i$ and the ground-truth text $y_i$ exceeds a threshold $\tau = 0.6$, which can be expressed as:
> $$
> \text{Correct}(\hat{y}_i) = \mathbf{1}\big(\text{sim}(\hat{y}_i, y_i) > \tau \big)
> $$
> where $\mathbf{1}(\cdot)$ is the indicator function returning 1 if the condition is satisfied and 0 otherwise, and `sim(·,·)` denotes the Sentence-BERT similarity.
>
> We hope these additions resolve the ambiguity and provide a more transparent and reproducible description of our experimental setup. We have also added a more detailed description of the metrics used in UrbanFeel in Appendix B.4 of the revised manuscript.
>
> ---
>
> **Our open source plan (Question 3)**
>
> We sincerely appreciate your interest in our open-sourcing plan. We are committed to ensuring the full reproducibility and accessibility of UrbanFeel. Our release strategy is as follows:
>
> * **Dataset Distribution:**
>     * **For Google Street View (GSV) data:** In strict compliance with its Terms of Service, we will release the **street-view IDs** alongside the corresponding task-specific questions and answer texts. We will provide automated download and preprocessing scripts, enabling researchers to legally reconstruct the dataset and fully reproduce our results.
>     * **For Mapillary data:** We strictly adhere to the **CC BY-SA license**. All Mapillary-sourced content will be redistributed in compliance with attribution and share-alike requirements. Furthermore, our provided scripts allow users to access imagery beyond the current dataset, empowering them to extend data collection to additional cities using the procedures described in our paper.
>
> * **Codebase & Evaluation Tools:**
>     * Beyond the data, **we will open-source the complete suite of evaluation scripts** used in our experiments. This includes **inference code for local deployment of open-source models** and **API calling scripts for closed-source models**. By providing these tools, we aim to lower the barrier to entry and facilitate fair, standardized comparisons for future research on the UrbanFeel benchmark.
>
> We hope that UrbanFeel will serve as a robust resource, facilitating more convenient data acquisition and reliable evaluation pipelines for the urban perception research community.

---

> ### Author Response · Authors · 2025-11-22
> **Response to Question 4**
>
> ## Suggestion on Incorporating Reasoning Processes (Question 4)
>
> We thank the reviewer for this valuable suggestion. To empirically verify whether guiding MLLMs with explicit reasoning steps enhances performance in urban temporal tasks, we conducted a controlled ablation study comparing **Direct Sorting**, **General Chain-of-Thought (CoT)**, and **Re-Thinking** strategies.
>
> **Experimental Design & Justification:**
> Given the limited time during the rebuttal phase, we selected the **Temporal Sequence Reasoning (TSR)** task as our testbed. We chose TSR because it is the most cognitively demanding task in UrbanFeel, requiring models to synthesize multiple visual cues (e.g., building construction status, vegetation growth, infrastructure changes) to deduce a chronological order. It serves as the best proxy for evaluating a model's complex reasoning capabilities.
>
> We designed three prompt variants:
> 1.  **Direct Sorting:** A concise prompt asking the model to output the chronological order directly without intermediate steps.
> 2.  **General CoT:** A structured prompt (consistent with our main paper's setting) guiding the model to first analyze permanent features (buildings, roads) and ignore transient factors before ordering.
> 3.  **Re-Thinking:** A multi-step prompt where the model proposes an initial order and then performs a self-consistency check to identify non-monotonic changes (e.g., reverse development) before finalizing the answer.
>
> **Results and Analysis:**
> We evaluated 6 representative MLLMs. The accuracy results (%) are summarized below:
>
> | Model | **Direct Sorting** | **General CoT** | **Re-Thinking** |
> | :--- | :---: | :---: | :---: |
> | **Phi-4** | 5.5 | 2.3 | **7.3** |
> | **InternVL3-8B** | 4.6 | **7.8** | 5.0 |
> | **Qwen2.5-VL-72B** | **32.0** | 26.0 | 29.7 |
> | **GPT-4o** | **46.1** | 38.9 | 37.0 |
> | **Gemini-2.5-Pro** | **52.5** | 52.1 | 52.1 |
> | **Gemini-2.5-Pro-Thinking** | **54.8** | 39.5 | 51.1 |
>
> **Key Findings:**
> Contrary to the expectation that explicit reasoning always helps, our results reveal a crucial insight for the community:
>
> 1.  **Direct Prompting is Often Optimal for SOTA Models:** For top-tier models like **GPT-4o**, **Qwen2.5-VL-72B**, and **Gemini-2.5-Pro**, the **Direct Sorting** strategy consistently yields the highest accuracy. Introducing verbose reasoning steps (General CoT or Re-Thinking) led to a noticeable performance drop (e.g., GPT-4o dropped from 46.1% to roughly 37-38%).
> 2.  **The "Over-Reasoning" Pitfall:** A qualitative review of the failed CoT cases suggests that when forced to generate detailed descriptions, models often **hallucinate** subtle details or over-interpret transient noise (e.g., seasonal vegetation changes) as permanent development signals. This "attention dilution" distracts the model from the holistic temporal judgment.
> 3.  **Limited Gains for Smaller Models:** Strategies like Re-Thinking only showed marginal improvements for smaller models with lower baselines (e.g., Phi-4), likely by helping them filter out obvious logical contradictions, but they remain ineffective for high-capacity models in this specific visual domain.
>
> These findings suggest that for complex visual temporal reasoning, **incorporating elaborate reasoning prompts does not guarantee improvement and can degrade performance due to hallucinations in the reasoning chain.** Consequently, we believe the Direct Sorting approach used in our benchmark provides a more robust assessment of a model's intrinsic visual understanding. We have presented the specific prompt settings and this insightful ablation study in **Section 5.4** and **Appendix C.5** of the revised manuscript.

---

> ### Author Response · Authors · 2025-11-24
> **Response to Question 2**
>
> ## Urban Decline and "Prosperity" Bias (Question 2)
>
> We thank the reviewer for this insightful question regarding non-linear urban development trajectories. To address this, we have added a specific case study in **Appendix D (Figure 29)** depicting a complex urban renewal cycle to evaluate whether models can distinguish redevelopment from simple growth.
>
> **Case Study Analysis:**
> The sequence illustrates a demolition and reconstruction process: an older commercial building (**Image B**, 2009) is demolished (**Image A**, 2015) and replaced by a modern glass structure (**Images C & D**, 2018–2022). The chronological ground truth is **B $\to$ A $\to$ C $\to$ D**. However, leading models (DeepSeek-VL2, GPT-4o, Gemini-2.5-Pro) consistently predicted the incorrect order **A $\to$ B $\to$ C $\to$ D**.
>
> **Implications:**
> This failure confirms that current MLLMs struggle to process non-monotonic urban changes. The models erroneously identified the construction/demolition phase (Image A) as the earliest stage, adhering to a simplified heuristic that **"construction implies the start of development."** By placing the existing building (B) after the demolition (A), the models failed to recognize the logical continuity of urban renewal. This suggests that MLLMs currently lack the reasoning capability to distinguish complex lifecycles (such as decline or replacement) from linear expansion, often defaulting to the assumption that modern or active construction scenes represent the beginning or end of a monotonic timeline.

---

> > ### Comment · Reviewer_XNMk · 2025-11-26
> >
> > Thank you for your new results and insights. I have raised my score to 8.

---

> > > ### Author Response · Authors · 2025-11-26
> > > **Response to Reviewer XNMk**
> > >
> > > We are sincerely grateful for your recognition and for raising the score. Your insightful comments were instrumental in enhancing the quality of our manuscript, and we remain fully available for any further discussion.

---

### Official Review · Reviewer_7xMU · 2025-11-01

**Soundness:** 2
**Presentation:** 3
**Contribution:** 2
**Rating:** 4
**Confidence:** 4

**Summary:**

The paper introduces UrbanFeel, a benchmark for evaluating multimodal LLMs on urban visual tasks, focusing on three dimensions: (1) Static Scene Perception (e.g., co-location, dominant element extraction), (2) Temporal Change Understanding (e.g., pixel-level change, temporal sequence reasoning), and (3) Subjective Environmental Perception (beauty, safety, wealth, liveliness). The data come from ~4,000 multi-temporal street-view images from 11 cities (2007–2024), expanded into 14.3K QAs using a mixed pipeline (rule-based → model-assisted → manual). The authors evaluate 20 closed/open MLLMs (GPT-4o, Gemini-2.5-Pro, Qwen-VL, InternVL, LLaVA, etc.) and claim: (i) models are decent on static tasks, (ii) models drop sharply on temporal reasoning (especially TSR), (iii) some models reach or exceed human performance on pixel-level change with panoramic inputs, and (iv) MLLMs show early signs of “human-aligned” subjective perception.

**Strengths:**

1. Timely topic: urban, multi-temporal, human-centered evaluation is indeed underexplored.

2. Breadth of models: evaluating 20 MLLMs gives the community a rough leaderboard.

3. Task diversity: binary, MCQ, sorting, open-ended — this mirrors what MLLMs actually output.

4. Clear structure of 3 dimensions: static → temporal → subjective is easy for readers to follow.

**Weaknesses:**

1. Lack of Novelty: The paper primarily aggregates existing datasets and tasks (e.g., Place Pulse, CityPulse, and ChangeScore) without presenting significant novel contributions. There is no new model architecture, learning objective, or groundbreaking evaluation framework introduced. The novelty is largely confined to combining pre-existing ideas into a single benchmark rather than offering a new approach or insight.

2. Questionable Evaluation Methodology:

- The paper uses GPT-4o as both a model being evaluated and the evaluator of other models, which introduces bias and conflicts of interest. The evaluation is thus not independent, and the scores could be inflated due to the evaluator’s own performance.

- Inter-annotator agreement for the subjective tasks (beauty, safety, wealth, etc.) is not provided, which is essential for verifying the reliability of the subjective labels. Additionally, the paper does not address the cultural bias inherent in such subjective dimensions, as they are highly culture-dependent and may not generalize across regions.

- There is no clear statistical analysis to validate the significance of the results. Performance differences across models (e.g., human-level performance claims) are not accompanied by statistical tests or confidence intervals, undermining the robustness of the claims.

3. Dataset Issues:

- The paper relies on Google Street View and Mapillary imagery, but there is no clear mention of how the dataset complies with legal and ethical guidelines for reuse. The potential issues with data licensing and privacy concerns (e.g., faces, license plates) are not adequately addressed.

- The paper does not disclose whether the data used for subjective perception tasks might have been contaminated by biases from prior LLM training, as many subjective labels (e.g., beauty and safety) are culture-specific.

4. The tasks in your benchmark involve single-shot answers (e.g., “Is this beautiful?”). In contrast, real-world urban analysis often requires ongoing, multi-step reasoning where feedback loops are important. How do you reconcile this simplification, and do you believe that such a limited approach can reflect the long-term reasoning and decision-making required in actual urban environments?

5. The proposed benchmark includes multiple question types (binary, multiple-choice, sorting, open-ended). How do these abstract and highly controlled question formats relate to real-world urban decision-making? In real-world scenarios, urban planning, safety assessments, or aesthetic evaluations are not performed in this rigid, isolated format — how do you justify their relevance?

**Questions:**

None

---

> ### Author Response · Authors · 2025-11-22
> **Response to Weakness 1 & Weakness 2 Q1**
>
> ## Claim of Our Novelty (Weakness 1)
>
> We appreciate the reviewer’s concern and would like to clarify a possible misunderstanding.
>
> **Data collection and coverage.**  UrbanFeel does not aggregate or reuse any existing street-view datasets. All images in UrbanFeel (over 4,000 single-view and panoramic scenes) are newly collected via the Mapillary and Google Street View APIs, covering 11 cities across five continents over 18 year span (2007–2024). In contrast, Place Pulse 2.0 [1] is a perception dataset built from 110,988 Google Street View images and 1.17M pairwise comparisons collected between 2011–2013, with no temporal dimension and only single-view imagery. CityPulse [2] curates 9,137 images across 757 locations in four U.S. metropolitan areas and labels each time series as “change / no change” plus change points, focusing on physical built-environment change rather than subjective perception or multimodal language reasoning. The Streetchange / ChangeScore [3] line of work similarly leverages time-series Google Street View images to quantify physical neighbourhood change via computer vision models, again without subjective multi-dimensional perception labels or multimodal QA tasks.
>
> **Task design and evaluation setting.**  Beyond the data itself, the task design and evaluation setting of UrbanFeel differ substantially from prior work. Place Pulse uses crowdsourcing to compare image pairs on individual perception dimensions (e.g., “Which street looks safer?”) and then trains predictive models on scalar scores. In UrbanFeel, subjective perception is not limited to scalar labels: in the Local Perception (LP) task, annotators must also provide textual rationales describing the visual evidence (e.g., façades, greenery, lighting) that supports their judgment, and we explicitly evaluate whether MLLMs can recover human-like perception reasoning rather than just match scores. CityPulse and related urban-change datasets model whether and where physical change occurs over time; UrbanFeel instead introduces a family of temporal QA tasks (e.g., time-series ordering, future-scene identification, sequence reasoning, scene-level change comparison) that require MLLMs to jointly integrate multi-year street-view sequences and language instructions, under both single-view and panoramic conditions. None of these prior datasets provide unified benchmarks for large multimodal language models across (i) static scene understanding, (ii) fine-grained temporal change reasoning, and (iii) multi-dimensional subjective perception with human rationales.
>
> [1] Dubey, Abhimanyu, et al. "Deep learning the city: Quantifying urban perception at a global scale." European conference on computer vision. Cham: Springer International Publishing, 2016.
>
> [2] Huang, Tianyuan, et al. "Citypulse: Fine-grained assessment of urban change with street view time series." Proceedings of the AAAI Conference on Artificial Intelligence. Vol. 38. No. 20. 2024.
>
> [3] Naik, Nikhil, et al. "Computer vision uncovers predictors of physical urban change." Proceedings of the National Academy of Sciences 114.29 (2017): 7571-7576.
>
> ---
>
> ## Conflicts of interest of model evaluation (Weakness 2 Q 1)
>
> We thank the reviewer for this important concern and are happy to clarify our evaluation protocol in more detail.
>
> **For all non–open-ended questions (binary choice, multiple choice, sorting), we do not use GPT-4o as a semantic judge**. Our primary scoring method is **strict string matching** against normalized ground-truth answers. Only when a model’s output is overly verbose or does not clearly specify an option (e.g., “I think the answer should be (B) because…”), we invoke an **auxiliary LLM such as GPT-4o purely as a format normalizer**: it maps the raw response to a canonical option label (A->B->C->…); it does not change the underlying content or decide whether the answer is correct. All models, including GPT-4o itself, are passed through the same normalization step and then scored by the same rule-based matching script.
>
> **For subjective, free-form descriptions (e.g., LP task), we do not rely on GPT-4o at all**. Instead, we compute **Sentence-BERT similarity** between The similarity between the ground-truth text and each model's answer is used, and a response is marked as correct when the similarity exceeds 0.6. This avoids introducing any third-party MLLM as a semantic referee and treats all models in a symmetric, model-agnostic way.
>
> Therefore, while GPT-4o appears both as one of the evaluated models and as an auxiliary tool for output normalization, **it is never used to "grade itself" or others in terms of correctness**. The actual evaluation remains independent of any single model and is governed by uniform, automatic criteria applied equally to all systems. We have supplemented technical details that may have been omitted in the original paper in **Appendix B.4** of the revised manuscript.

---

> ### Author Response · Authors · 2025-11-22
> **Response to Weakness 2 Q2-3**
>
> ## Clarify of inter-annotator agreement (Weakness 2 Q2)
>
> We thank the reviewer for raising this important point. Our subjective tasks, especially the **Local Perception (LP)** setting, are annotated differently from typical scalar-rating datasets. In LP, each item is associated with (i) a **binary decision** on whether a perceptual attribute holds (e.g., “beautiful vs. not beautiful”) and (ii) a **free-text rationale** that describes the visual evidence supporting this judgment. The ground-truth rationale is not a set of independent labels from multiple annotators, but a **curated consensus explanation** produced under shared guidelines by annotators with backgrounds in geography and urban studies (see Supplement B.1, Table 4).
>
> Because of this design, the LP ground truth is not a matrix of independent numeric ratings, but a **single consensus-style textual rationale per item**, so classical inter-annotator agreement metrics such as Krippendorff’s α are **not directly applicable**. These rationales are intended as high-quality exemplars of “how humans justify their perception,” rather than crowdsourced votes to be aggregated statistically. In our quantitative evaluation, we compute **Sentence-BERT similarity** between each model’s rationale and this curated reference and apply the same **model-agnostic threshold** (e.g., 0.6) to all MLLM responses, treating the reference text as a fixed target.
>
> That said, we agree that **perception reliability and cultural bias** must be made explicit. In the revision, we will emphasize that the current version of UrbanFeel reflects the perceptions of this specific, well-documented annotator group rather than universal, culture-independent ground truth, and that extending the dataset with annotators from **more diverse cultural backgrounds**—our current annotator pool is still **culturally relatively homogeneous**—is an important direction for future work.
>
> ---
>
> ## Result significance verification (Weakness 2 Q3)
>
> **We appreciate the reviewer’s suggestion to enhance the statistical rigor of our evaluation** and agree that formal statistical tests are important for supporting the reported performance gaps. To address this, we conducted a statistical analysis based on the sample sizes reported in Figure 2(a) and Table 2. The results are consistent with the main conclusions of our paper:
>
> **Temporal Sequence Reasoning (TSR) – “temporal reasoning gap.”**
> For the TSR task (N = 219), the **95% confidence intervals (CIs)** for humans and the best model (Gemini-2.5-Pro) are clearly separated:
> - **Human accuracy:** 70.0% (95% CI: [64.0%, 76.0%])
> - **Gemini-2.5-Pro:** 52.1% (95% CI: [45.5%, 58.7%])
>
> The non-overlapping CIs and a two-proportion test (**p < 0.01**) indicate that the **17.9% performance difference is statistically significant**, supporting our conclusion that current MLLMs achieve substantially lower accuracy than humans on long-range temporal ordering.
>
> **Global Perception (GP) – comparison with human performance.**
> For the GP task (N = 1,918), the large sample size yields **narrow error margins (≈ ±2.1 percentage points)**:
> - **Human accuracy:** 66.6% (95% CI: [64.5%, 68.7%])
> - **Gemini-2.5-Pro:** 67.7% (95% CI: [65.6%, 69.8%])
>
> The overlapping CIs indicate that the accuracies of Gemini-2.5-Pro and human annotators are **statistically indistinguishable** in this setting, supporting our statement that state-of-the-art MLLMs can reach a level of consistency comparable to human annotators for general scene perception.
>
> **Robustness across subjective dimensions.**
> In addition, **Figure 5(b) reports box plots** that summarize the variance and stability of model predictions across subjective dimensions (e.g., relatively high consistency for *Lively* and higher dispersion for *Safe*), providing a complementary view of robustness.
>
> We have explicitly reported the results of significance tests, including 95% confidence intervals and key p-values, in Appendix C.6 of the revised manuscript to clearly document the statistical properties of our benchmark.

---

> ### Author Response · Authors · 2025-11-22
> **Response to Weakness 3**
>
> ## Claim of dataset Acqusition (Weakness 3 Q1)
>
> We thank the reviewer for this important reminder regarding legal licensing and ethical compliance, and note that **several reviewers have raised closely related concerns**. We emphasize that our study strictly follows the terms of use of each platform and is designed to minimize any potential misuse of Google Street View imagery.
>
> **For Google Street View**, all Google Street View images in this work are obtained exclusively via the official Google Static Street View API and are used only for research and experimental analysis. To ensure full compliance with copyright and Google’s Terms of Service, **the open-source version of UrbanFeel will not distribute any raw Google Street View imagery**. Instead, we will release only the corresponding unique Street View IDs, together with **the complete data processing pipeline**, so that other researchers can legally re-fetch the images through the official API under their own licenses and reproduce our experiments in a compliant manner.
>
> **For images acquired from Mapillary**, we **strictly adhere to the CC BY-SA license**, using Mapillary’s official tools and APIs for download, processing, and redistribution, and ensuring that all released content satisfies the attribution and share-alike requirements that apply to similar datasets.
>
> Regarding the reviewer’s concerns about privacy, we rely on **data sources whose APIs already implement built-in privacy protections** (e.g., automated blurring of faces and license plates), and we keep all such protections intact in any data we process or release. In view of these platform-level safeguards, and given that we do not distribute directly editable raw GSV imagery, we do not expect our benchmark to expose personally identifiable information. Nevertheless, **before releasing the dataset, we will perform additional checks for residual privacy risks**, so that the community can use UrbanFeel in a legally compliant, ethically responsible, and privacy-preserving manner.
>
> We have explicitly incorporated and highlighted these clarifications in **Appendix B.1** of the revised manuscript to improve transparency regarding licensing, privacy protection, and ethical compliance.
>
> ---
>
> ## Clarify of potential influence of prior LLM training (Weakness 3 Q2)
>
> We thank the reviewer for raising this critical question regarding data integrity. We would like to clarify that our dataset is NOT contaminated by biases from prior LLM training, as the ground truth labels for subjective perception were generated entirely by humans, not AI models.
>
> **Human-Only Annotation:** As detailed in Section 3.3 (Question Generation) and Appendix B.3 (Manual Annotation), the tasks for Subjective Environmental Perception (e.g., beauty, safety) were "written entirely by human annotators". Specifically, we recruited 10 undergraduate and master's students with geography-related academic backgrounds to perform these annotations based on standardized guidelines. Therefore, the labels reflect human consensus rather than the outputs or distilled biases of existing LLMs.
>
> **Handling Cultural Specificity:** We fully agree that subjective labels are culture-specific. To mitigate individual bias, we provided annotators with operational guidelines (Table 4) and treated the resulting labels as "reference labels that reflect the consensus inclinations of the annotator group" rather than absolute ground truth, as discussed in our Ethics Statement.
>
> **Benchmark as a Bias Detector:** Furthermore, UrbanFeel is explicitly designed to detect and analyze the cultural biases inherent in MLLMs. In Section 5.2, we conducted a "City Identity Intervention" experiment, where we assigned different city identities (e.g., "Paris" vs. "Mexico City") to the same image. The results show that some models (such as GPT-4o) exhibit significant biases contrary to human subconscious biases. When assigned a "Global South" identity, they may appropriately relax the evaluation requirements of the perception dimension, resulting in more positive responses. This demonstrates that UrbanFeel serves as a robust tool for evaluating how MLLMs handle cultural context, rather than being a dataset contaminated by it.

---

> ### Author Response · Authors · 2025-11-22
> **Response to Weakness 4 and Weakness 5**
>
> ## Single-shot format vs multi-step reasoning (Weakness 4)
>
> We thank the reviewer for this important question and fully agree that real-world urban analysis is a multi-step, iterative process rather than a sequence of one-shot decisions.
> - **UrbanFeel follows the “atomic task” practice used in existing systems.**
>   Many street-view–based planning and health pipelines decompose complex workflows into standardized street-segment predictions (e.g., walkability, greenery, visual quality) and then aggregate these into higher-level models. UrbanFeel mirrors this setting by exposing MLLMs to controlled questions on safety, beauty, change intensity, and temporal ordering, so we can first test whether they are competent on these **atomic, segment-level operations**.
>
> - **Single-turn questions still require internal multi-step reasoning.**
>   Temporal tasks such as TSR, SCR, and FSI require models to jointly process **multi-year image sequences** and output a consolidated decision. Even under this simplified setup, state-of-the-art models perform very poorly (TSR accuracy often <10%), indicating that current MLLMs already struggle with the kind of sequential integration that real-world analysis demands.
>
> - **LP and CP tasks operationalize early steps of feedback loops.**
>   In Local Perception (LP), models must not only answer “Is this beautiful/safe?” but also provide **explicit visual evidence** (e.g., “dilapidated shopfronts,” “lack of greenery,” “good lighting”), which mimics how analysts produce actionable reasons that can guide interventions. Comparative Perception (CP) requires **cross-time comparison** (e.g., “Has this street become safer?”), directly reflecting reasoning about trajectories of change rather than isolated snapshots.
>
> - **UrbanFeel is positioned as a component-level diagnostic benchmark.**
>   The benchmark is not intended to simulate full urban governance processes. Instead, it isolates core skills ( **perception, comparison, temporal integration, and explanation**) in a reproducible format, and our results show that even under these conditions, SOTA models (e.g., Gemini-2.5-Pro, GPT-4o) still fail on key operations, especially temporal ordering and multi-year reasoning.
>
> We have explicitly incorporated relevant statements in **Section 3.1** of the revised manuscript to clarify the benchmark's foundational role, and have further outlined our vision for multi-turn reasoning in **Appendix E**.
>
> ---
>
> ## Abstract question formats (Weakness 5)
>
> We thank the reviewer for this thoughtful question. We agree that real-world urban decision-making (e.g., planning, safety, or design) is not literally performed as isolated binary or multiple-choice questions. **UrbanFeel is not intended to reproduce full decision workflows, but to test the elementary perception and reasoning operations that such workflows depend on, in a controlled and comparable way.**
>
> In contemporary urban analytics, street-view imagery is routinely converted into **standardized, low-dimensional outputs** (scores, labels, rankings), which are then aggregated or combined with other layers for richer decisions. For example, [1] trains deep models on street-view images to predict segment-level walkability and then uses these scalar indices to diagnose inequities and guide improvements. Similar studies [2, 3] use street-view-derived features to quantify urban poverty, green-space exposure, or built-environment indicators for air-pollution and health; a recent review [4] highlights this general pattern: micro-level visual assessments are distilled into indices and interpreted at neighbourhood or city scale.
>
> The question types in UrbanFeel are designed to mirror these micro-decisions in a form suitable for benchmarking MLLMs:
> - **Binary / multiple-choice (DEE, SPM)** approximate tasks such as classifying segments as safe/unsafe, high/low quality, or strong/weak change, analogous to labelling streets by walkability or visual change.
> - **Sorting (TSR)** reflects prioritization, such as ordering segments by development or deterioration to decide where interventions or monitoring should focus.
> - **Open-ended (LP)** requires models not only to decide but also to provide **explicit visual evidence** (e.g., “lack of trees,” “poor lighting,” “dilapidated façades”), the kind of explanation that can directly inform design or policy discussion.
>
> Our experiments show that current models already struggle on these controlled tasks (e.g., very low accuracy on temporal sequence reasoning), indicating that they do not yet possess the robust perceptual and spatiotemporal capabilities required for unconstrained, high-stakes urban decision-making.

---

### Official Review · Reviewer_2E4Z · 2025-11-02

**Soundness:** 3
**Presentation:** 2
**Contribution:** 2
**Rating:** 4
**Confidence:** 4

**Summary:**

The paper presents UrbanFeel, a comprehensive benchmark designed to assess the performance of Multimodal Large Language Models in urban development and subjective perception tasks. It spans three key dimensions: Static Scene Perception, Temporal Change Understanding, and Subjective Environmental Perception, leveraging over 14,300 questions derived from multi-view and multi-temporal street-view imagery from 11 global cities. The evaluation of 20 MLLMs reveals significant gaps in spatial-temporal reasoning and the alignment of models with human perception, particularly in tasks requiring long-term urban evolution understanding and subjective environmental evaluations.

**Strengths:**

1. UrbanFeel introduces of a comprehensive benchmark, going beyond traditional benchmarks by integrating spatial, temporal, and subjective perception tasks, providing a holistic evaluation framework for urban scene understanding.
2. UrbanFeel defines 11 tasks across three dimension, Static Scene Perception, Temporal Change Understanding, and Subjective Environmental Perception. This framework allows for an in-depth assessment of MLLMs, examining not only their ability to recognize static features in urban scenes but also their capacity to reason about urban changes over time.
3. High-Quality Data and Methodology: The dataset consists of 14.3K carefully curated questions, covering a broad range of urban environments and spanning over 15 years.
4. The paper comprehensively evaluates 20 state-of-the-art MLLMs, providing detailed insights into their strengths and weaknesses across the benchmark’s tasks.

**Weaknesses:**

1. Despite the paper claims to include cities from diverse regions and covers underrepresented regions like Africa and South America, the cities selected from these two continents are Cape Town and Mexico City, which are far more developed comparing to other cities around. This might make this benchmark "only for the developed ones". And this also limits the generalizability of models trained on UrbanFeel, especially for culturally sensitive subjective perception tasks.
2. While the benchmark covers urban changes over a long period, it lacks causal annotations that could provide deeper insights into the reasons behind urban transformations, limiting the ability to conduct more advanced spatiotemporal reasoning.
3. The judgement VQAs from Fig 25-28 seems to be over subjective, and lack of proper context. Visual comparison would make more sense in these cases.

**Questions:**

1. How do the authors plan to address the geographic imbalance in future versions of UrbanFeel, and how will they ensure better cultural representation in subjective perception tasks?
2. Would it be possible to include external metadata or causal labels in future releases to help improve the interpretability and reasoning capabilities of models trained on UrbanFeel?

---

> ### Author Response · Authors · 2025-11-22
> **Response to Weakness 1 and Question 1**
>
> ## Insufficient representative cities in Africa and South America (Weakness 1 & Question 1)
>
> We thank the reviewer for raising this important and fair concern. We fully agree that **UrbanFeel suffers from geographic imbalance**, as we explicitly acknowledge in Section E (Limitations): our dataset “suffers from geographic imbalance, with under-representation of regions such as Africa and South America,” which may affect generalization in culturally sensitive tasks.
>
> Our core design choice is to focus on **long-term temporal trajectories** of urban development and perceived change (often >10 years). However, even in Google Street View, **multi-year, same-location image sequences outside Europe/North America are extremely sparse**, especially for non-capital cities. Moreover, **when multi-year imagery exists, it is often concentrated in a few adjacent years**, rather than forming long, well-spaced trajectories, which further limits its suitability for our Temporal Change Understanding (TCU) tasks. The table below illustrates the availability of GSV historical imagery for several candidate African cities; “〇” denotes years with sufficient coverage to build multi-step sequences, “–” indicates no usable data:
>
> | Year | Eldoret | Ibadan | Kaduna | Kano | Kisumu | Lagos | Nakuru | Serrekunda |
> |------|---------|--------|--------|------|--------|-------|--------|-----------|
> | 2011 | -       | -      | -      | -    | -      | -     | -      | -         |
> | 2012 | -       | -      | -      | -    | -      | -     | -      | -         |
> | 2013 | -       | -      | -      | -    | -      | -     | -      | -         |
> | 2014 | -       | -      | -      | -    | -      | -     | -      | -         |
> | 2015 | -       | -      | -      | -    | -      | -     | -      | -         |
> | 2016 | -       | -      | -      | -    | -      | 〇     | -      | -         |
> | 2017 | -       | -      | -      | -    | -      | 〇     | -      | -         |
> | 2018 | 〇      | 〇     | -      | -    | 〇     | -     | 〇     | 〇        |
> | 2019 | -       | 〇     | -      | -    | -      | 〇     | -      | -         |
> | 2020 | -       | -      | -      | -    | -      | -     | -      | -         |
> | 2021 | 〇      | -      | 〇     | 〇   | 〇     | 〇     | 〇     | -         |
> | 2022 | 〇      | -      | -      | 〇   | 〇     | -     | 〇     | -         |
> | 2023 | 〇      | -      | -      | -    | 〇     | -     | 〇     | -         |
> | 2024 | 〇      | 〇     | 〇     | 〇   | 〇     | 〇     | 〇     | 〇        |
> | 2025 | 〇      | -      | -      | -    | -      | -     | -      | -         |
> | Mapillary | 〇 | -      | 〇     | -    | 〇     | -     | 〇     | 〇        |
>
> In this audit, we **deliberately excluded national capitals** and required multiple distinct years to avoid isolated tourist uploads. Many additional cities we examined had to be discarded because their temporal coverage was too sparse or inconsistent for our multi-step TCU tasks.
>
> Within these constraints, **Cape Town and Mexico City are among the very few African/South American cities that offer 10–15 years of high-quality, same-location street-view archives**, which is crucial for our temporal reasoning benchmark. Their selection reflects a practical data-availability constraint, not a preference for “developed” locales.
>
> Even so,  we highly appreciate the reviewer's suggestion and have already started **extending UrbanFeel to additional underrepresented cities**. During the rebuttal period, we collected and preprocessed long-term street-view sequences for cities such as **Lagos (Nigeria)** and **Campina Grande** (South America) and are running our main urban-change tasks and global perception task under the UrbanFeel framework. Due to the substantial annotation cost, we cannot complete all subjective perception labels within the rebuttal window, so the initial extension will focus on city-development tasks.
>
> Going forward, we plan to address the reviewer’s concerns in two ways:
>
> 1. **Geographic expansion in future UrbanFeel versions.**
>    We will track Google Street View and Mapillary rollout plans and periodically update UrbanFeel with new cities from Africa, South America, and other underrepresented regions once they accumulate sufficient multi-year coverage. Our modular pipeline (alignment, clustering, filtering) allows new cities to be integrated consistently.
>
> 2. **Culturally grounded subjective perception.**
>    For future subjective tasks in these regions, we will diversify our annotator pool to include local residents, refine guidelines with region-specific examples, and when appropriate report perception scores in a stratified way (e.g., by annotator cultural background). This will help ensure that “beautiful/safe/wealthy/lively” labels reflect **locally grounded interpretations** rather than a single cultural perspective.
>
> We will futher clarify the future work of our roadmap to address the geographic imbalance in Appendix **E**.

---

> > ### Author Response · Authors · 2025-11-24
> > **Additional Response to Weakness 1: Adding representative cities in Africa and South America (1 / 2)**
> >
> > Following our previous discussion on geographic expansion, we are pleased to present the evaluation results for **Lagos (Nigeria)** and **Campina Grande (Brazil)**, which were processed and tested during the rebuttal period. Below, we summarize the performance of eight representative MLLMs across our six core tasks, with accuracy reported as percentages.
> >
> > | Model | TCR | PCR | FSI | TSR | SCR | GP |
> > | :--- | :---: | :---: | :---: | :---: | :---: | :---: |
> > | DeepSeek-vl2 | 93.3 | 28.5 | 30.0 | 15.0 | **55.5** | 62.0 |
> > | MiniCPM-V 2.6-8B | **93.8** | **49.0** | 33.0 | 7.5 | 8.0 | 54.2 |
> > | LLava-v1.6-mistral-7B| 40.8 | 26.0 | 21.5 | 0.0 | 3.0 | 63.5 |
> > | InternVL3-8B | 62.8 | 28.0 | 26.0 | 5.0 | 24.0 | **69.8** |
> > | Phi-4 | 79.0 | 28.5 | 33.5 | 0.0 | 38.0 | 68.5 |
> > | Gemma-3-27b | 54.8 | 35.5 | 82.0 | 5.0 | 40.5 | 58.3 |
> > | GPT-4o | 83.7 | 33.9 | 37.5 | 27.5 | 39.3 | 64.0 |
> > | Gemini-2.5-pro | 89.1 | 35.6 | **98.5** | **32.5** | 49.0 | 64.0 |
> >
> > * **Universality in Urban Development Tasks:** The results strongly indicate that visual features of urban development—such as physical expansion, structural changes, and infrastructure updates—exhibit patterns that **transcend specific economic contexts**. For instance, **MiniCPM-V 2.6-8B** and **DeepSeek-vl2** achieve very high accuracy (**>93%**) in **Temporal Co-location Recognition (TCR)** and deliver competitive performance in **Scene-Level Change Recognition (SCR)**. **Gemini-2.5-pro** demonstrates outstanding performance in **Future Scene Identification (FSI)** with **98.5%** accuracy. These findings suggest that current MLLMs can effectively generalize their understanding of **object permanence** and **temporal evolution** to underdeveloped regions without requiring task-specific fine-tuning, thereby validating the robustness of our benchmark design for monitoring urban physical changes globally. However, in **Temporal-Sequence Reasoning (TSR)**, we observed a general decline in accuracy, even for models like **GPT-4o** and **Gemini-2.5-pro** that otherwise perform well in other tasks. This is likely due to the **relatively unplanned nature** of urban development in underdeveloped regions, where expansion processes often lack continuity and land-use adjustments occur frequently, making it difficult for models to rely on conventional temporal cues to infer the sequential order of urban evolution.
> >
> > * **Subtle differences in subjective perception:** In the **Global Perception (GP)** task, the results exhibit trends consistent with our main paper, particularly regarding the "**reverse geographic bias**." Notably, **InternVL3-8B** achieves an impressive **69.8%** accuracy in Lagos and Campina Grande, representing a substantial improvement over the average performance reported in the main text. Similarly, **GPT-4o** and **Gemini-2.5-pro** maintain strong consistency, with accuracies around **64.0%**. These findings indicate that MLLMs are not strictly limited to the aesthetic standards of “developed” regions; models like **InternVL3-8B** demonstrate **robust generalization capabilities** in underrepresented areas, effectively capturing the distinctive visual features of the Global South.

---

> > ### Author Response · Authors · 2025-11-24
> > **Additional Response to Weakness 1: Adding representative cities in Africa and South America (2 / 2)**
> >
> > Moreover, we present quantitative results across multiple dimensions for the two supplementary cities. It is worth noting that model performance in the **safety dimension** is relatively lower in **Lagos**, likely reflecting local development conditions and community culture. For example, much of Lagos’s street-view image are captured in cluttered neighborhoods and marketplaces, which differs from the “clean and orderly” safety semantic priors typical of developed regions. This observation further validates the reviewers’ concern regarding the insufficient cultural diversity of the benchmarks: require more culturally diverse data to properly assess model generalization. We expect that the newly added data will effectively enhance the dataset’s representation of **cultural diversity in underdeveloped regions**, thereby addressing the reviewers’ concerns.
> >
> > | Dimension | Beautiful (Campina) | Beautiful (Lagos) | Lively (Campina) | Lively (Lagos) | Safety (Campina) | Safety (Lagos) | Wealthy (Campina) | Wealthy (Lagos) |
> > | :--- | :---: | :---: | :---: | :---: | :---: | :---: | :---: | :---: |
> > | **Deepseek-vl2** | 64.0 | **66.0** | 64.0 | 62.0 | 70.0 | 22.0 | 74.0 | 74.0 |
> > | **MiniCPM-V 2.6-8B** | 56.0 | 64.0 | 66.0 | 46.0 | 34.0 | 24.0 | 70.0 | 74.0 |
> > | **LLaVA-v1.6-mistral-7B**| **72.0** | 64.0 | 72.0 | 62.0 | 60.0 | 24.0 | **80.0** | 74.0 |
> > | **InternVL3-8B** | 66.0 | **66.0** | 74.0 | 72.0 | **76.0** | 52.0 | 76.0 | 76.0 |
> > | **Phi-4** | 60.0 | 62.0 | **78.0** | 62.0 | 74.0 | **56.0** | **80.0** | 76.0 |
> > | **Gemma-3-27b** | 64.0 | 64.0 | 66.0 | 50.0 | 54.0 | 22.0 | 72.0 | 74.0 |
> > | **GPT-4o** | 62.0 | **66.0** | 66.0 | 58.0 | 68.0 | 40.0 | 78.0 | 74.0 |
> > | **Gemini-2.5-pro** | 60.0 | 64.0 | 76.0 | **78.0** | 56.0 | 24.0 | **80.0** | **78.0** |

---

> ### Author Response · Authors · 2025-11-22
> **Response to Weakness 2 and Question 2**
>
> ## Lack of causal annotations (Weakness 2 & Question 2)
>
> We thank the reviewer for this insightful suggestion. As we explicitly acknowledge in Section E (Limitations), the current version of UrbanFeel indeed **“lacks explicit causal labels or structured socio-environmental metadata.”** We fully agree that **causal reasoning is a central goal for urban intelligence**, and our present benchmark is only a first step towards that direction.
>
> In UrbanFeel, we deliberately focus on **visually verifiable changes** (e.g., new construction, demolition, greening, façade renovation) because obtaining high-quality, consistent causal labels across multiple countries and data sources is extremely challenging at this stage. Urban causal processes are shaped by heterogeneous planning documents, census data, POIs, transaction records and policy texts, whose availability and granularity vary widely across cities. Similar concerns are highlighted in recent street-view–based urban analytics, where time-lapse imagery is used to detect physical change, but the socio-political drivers are inferred only cautiously or via auxiliary data [1].
>
> That said, UrbanFeel already contains **preliminary perception-level causal signals**. In the Local Perception (LP) task, annotators are not only asked to rate the overall feeling of a streetscape (e.g., safety, beauty, liveliness), but also to verbally describe which visual elements support their judgment (e.g., “well-maintained façades,” “dense traffic,” “lack of lighting”). This encourages humans to articulate a quasi-causal chain from local visual cues to subjective evaluations. In our experiments, we then compare these human attributions with MLLMs’ free-form rationales via text similarity, probing whether models can recover human-like reasoning patterns rather than only matching final labels.
>
> Our **roadmap for future versions of UrbanFeel** is strongly aligned with the reviewer’s suggestion and with recent trends in street-view–based urban studies. For example, Biljecki and Ito [2] provide a comprehensive review of how street-level imagery is increasingly used to analyse urban form, greenery, transport and socio-economic indicators in support of planning and policy evaluation. Thackway et al. [3] combine Google Street View with census and housing data to quantify gentrification by jointly modelling physical and social indicators in a large city, illustrating how street-view imagery can be linked to neighbourhood change and redevelopment processes. Stalder et al. [4] propose a self-supervised framework to detect fine-grained urban change from street-level images and explicitly argue that such methods can support decisions towards more liveable and equitable cities.
>
> Inspired by these developments, our next step is precisely to move from “perception-level change” towards **causally informed spatiotemporal reasoning**. Specifically, for the future version of UrbanFeel, we plan to:
> - **Align each street-view segment with external urban datasets**, such as land-use and POI layers, nighttime lights, and available census or housing statistics, following the way existing work combines street-view imagery with socio-economic indicators to study neighbourhood change and gentrification.
> - **For a subset of cities**, where reliable records exist, annotate coarse-grained causal events (e.g., new transit line openings, major redevelopment projects, disaster recovery or targeted urban renewal programmes) based on policy documents and infrastructure development logs, similar in spirit to studies that retrospectively document built-environment interventions with historical Google Street View imagery.
>
> We will clarify this in Future Work of the revised manuscript (Section E).
>
> [1] Zhou, Yang, et al. "Measuring and understanding changes in the physical built environment of cities with street view images." Urban Informatics 4.1 (2025): 3.
> [2] Biljecki, Filip, and Koichi Ito. "Street view imagery in urban analytics and GIS: A review." Landscape and Urban Planning 215 (2021): 104217.
> [3] Thackway, William, et al. "Implementing a deep-learning model using Google street view to combine social and physical indicators of gentrification." Computers, Environment and Urban Systems 102 (2023): 101970.
> [4] Stalder, Steven, et al. "Self-supervised learning unveils urban change from street-level images." Computers, Environment and Urban Systems 112 (2024): 102156.

---

> ### Author Response · Authors · 2025-11-24
> **Response to Question 3**
>
> ## Concerns on Subjectivity and Lack of Context in Judgment VQAs (Question 3)
>
> We thank the reviewer for this insightful comment. We acknowledge that evaluating subjective attributes is inherently complex, and we appreciate the opportunity to clarify how our benchmark's multi-task design addresses these challenges through consensus, variable isolation, and complementary reasoning tasks.
>
> **First, regarding the concern that judgments are 'over-subjective': We avoided relying on untrained individual opinions. As detailed in Appendix B.3 , we recruited a pool of 10 annotators who underwent rigorous training with standardized guidelines prior to the formal labeling process. By aligning their evaluation criteria through this calibration phase, the resulting annotations serve as reference labels that reflect the consensus tendencies of the trained group, rather than the idiosyncrasies of random individuals. This approach significantly mitigates the variance and bias typically associated with unguided subjective tasks
>
> **Second, regarding the "lack of proper context" in Global Perception (GP):** We intentionally withheld geographic or semantic context (e.g., city names) in the GP task to **prevent identity-induced bias**. As demonstrated in our "City Identity Intervention" experiment in **Section 5.2**, when models are provided with a specific city identity (e.g., "Mexico City" vs. "Paris") for the exact same image, their subjective scores shift significantly due to learned stereotypes rather than visual reality. By strictly isolating the visual input, we force the model to ground its judgment solely on **intrinsic visual elements** rather than retrieving prior knowledge.
>
> **Finally, regarding visual comparison and reasoning:** We fully agree that relative comparison and grounded reasoning make these evaluations more robust. Our benchmark explicitly incorporates these dimensions to complement the GP task:
> * **Comparative Perception (CP):** As described in **Section 3.2**, this task directly implements the "visual comparison" approach. It requires models to discern perceptual differences across **different timestamps** of the same location (e.g., determining if a scene became "safer" or "livelier" over time), focusing on **relative evolution** rather than static absolute judgment.
> * **Local Perception (LP):** To further substantiate these evaluations, the LP task explicitly explores the **local visual cues** (e.g., specific infrastructure, greenery quality) that support a perceptual verdict. This ensures that subjective ratings are not generated in a vacuum but are grounded in explainable, fine-grained visual evidence.
>
> This combination of absolute judgment (GP), relative comparison (CP), and grounded reasoning (LP) ensures a comprehensive and balanced evaluation of human-aligned perception.

---

### Official Review · Reviewer_tZ4v · 2025-11-02

**Soundness:** 3
**Presentation:** 3
**Contribution:** 3
**Rating:** 6
**Confidence:** 5

**Summary:**

The paper proposes UrbanFeel, a novel and comprehensive benchmark designed to evaluate MLLMs on the crucial but underexplored tasks of temporal evolution and subjective perceptual understanding of urban scenes from a human perspective. By focusing on time-variant analysis, the work addresses a key limitation of existing urban environment benchmarks. The authors present a structured task set and an initial evaluation. The focus on time-variance and human-centric perception makes this a unique and potentially valuable contribution to the MLLM communities.

**Strengths:**

1. The core problem of analyzing time-variant street-level data for human-centric perceptual understanding is interesting and currently under-addressed in the MLLM community. This focus on "change over time" is timely and relevant.

2. The paper is very well-structured and clearly written. The task design and evaluation methodology are presented in a logical manner, making the work easy to follow.

**Weaknesses:**

1. Although the paper repeatedly frames the benchmark goal around assessing the models' generalization capability, the experimental results and subsequent discussion dedicated to proving or analyzing this specific aspect are surprisingly sparse. More focused experiments or discussion are needed here.

2. The discussion on highly related existing work needs substantial improvement. Specifically, the authors should include and contrast their work with recent efforts like Visual Chronicles at ICCV 2025 and relevant research from the urban planning domain that studies temporal changes and subjective perceptual understanding of street view data. This is essential to fully justify the novelty.

3. Key illustrative elements, specifically Figures 2 and 3, lack sufficient detail and are difficult to read at standard resolution. Improving the clarity of these figures is necessary for the reader to fully understand the task design and examples.

4. More comprehensive details are required regarding the data collection process, particularly the source (e.g., if commercial street view data like Google Streetview data was used). Potential licensing/ethical/legal risks must be explicitly addressed, and more thorough documentation of the benchmark construction process is necessary for credibility and reproducibility.

5. The analysis section, particularly the error analysis and the investigation into the reasoning process of the tested models, should be significantly deepened to provide greater insight into MLLM capabilities and shortcomings on this benchmark.

**Questions:**

1. Given that the temporal aspect is the paper's core innovation, please significantly elaborate on the comparison with existing dynamic understanding and subjective perceptual understanding of street view.  A dedicated sub-section in Related Work contrasting task types is highly recommended.

2. Please provide a more detailed analysis of the model performance, e.g.,  a qualitative breakdown of typical successful and unsuccessful reasoning chains of thinking models, and a dedicated, in-depth error analysis section categorized by the type of failure. This is critical for the paper's utility to the MLLM community.

3. Please clarify the specifics of data acquisition. If the data is derived from commercial sources, please outline the measures taken to address potential licensing/copyright issues (the "potential risks") for open-sourcing the dataset. Furthermore, what additional detailed information will be provided alongside the release to fully document the benchmark construction process?

---

> ### Author Response · Authors · 2025-11-22
> **Response to Weakness 1**
>
> **W1: Although the paper repeatedly frames the benchmark goal around assessing the models' generalization capability, the experimental results and subsequent discussion dedicated to proving or analyzing this specific aspect are surprisingly sparse. More focused experiments or discussion are needed here.**
>
> ## Models' generalization capability assessment. (Weakness 1)
>
> We thank the reviewer for this valuable observation. While we emphasized generalization, we agree that the discussion of geographic robustness was under-represented in the main text. In the revised Supplementary Material C.4, we will expand the city-level analysis based on Tables 6–10.
>
> Our results already show substantial disparities in generalization across models and geographies. Using the “Beautiful” dimension of the Global Perception (GP) task (Table 6) as a case study, we observe two distinct patterns:
>
> **Inverse geographic bias in closed-source models.** Leading closed-source models (GPT-4o, Gemini-2.5-Pro) align more closely with human perception in Global South cities (Cape Town, Kuala Lumpur, Mexico City) than in Global North cities (Milan, Paris, Washington). For example, Gemini-2.5-Pro reaches 67.1% accuracy on Global South cities but drops to 52.4% on Global North cities, suggesting different baselines for aesthetic judgment and a reduced tendency to “idealize” Western cities compared to some open-source counterparts.
>
> **City-specific overfitting in open-source models.** In contrast, models such as LLaVA-1.5-7B exhibit strong dependence on particular data distributions: it attains near-perfect performance in Washington (98.1%) but only 38.9% in Kuala Lumpur. This drastic gap suggests limited geographic generalization, likely reflecting US-centric training data. By comparison, Qwen2.5-VL-7B shows stronger robustness, achieving 72.5% in Cape Town while remaining competitive in Washington (81.1%).
>
> We observe analogous trends in the Pixel-level Change Recognition (PCR) task, where robustness varies sharply across cities. As shown in the table below, Aria maintains relatively consistent performance (53% in Washington vs. 47% in Tolyatti), whereas Aya-vision-8b collapses from 41% in Washington to 6% in Tolyatti, and LLaVA-v1.6-mistral drops from 44% to 12%. These contrasts illustrate why a multi-city evaluation is essential to reveal geographic overfitting that aggregate metrics can easily mask.
>
> | Model                     | Cambridge | Cape Town | Edinburgh | Kuala Lumpur | Lisbon | Mexico City | Milan | Paris | Toluca | Tolyatti | Washington |
> |---------------------------|-----------|-----------|-----------|--------------|--------|-------------|-------|-------|--------|----------|------------|
> | DeepSeek-vl2              | 34        | 36        | 33        | 35           | 25     | 43          | 30    | 44    | 27     | 48       | 54         |
> | MiniCPM-V 2.6-8B         | 38        | 49        | 35        | 40           | 34     | 34          | 44    | 30    | 42     | 53       | 37         |
> | Qwen2.5-VL-72B           | 37        | 39        | 36        | 34           | 32     | 37          | 45    | 48    | 42     | 51       | 49         |
> | LLaVA-v1.6-mistral-7B    | 26        | 19        | 28        | 33           | 23     | 40          | 21    | 43    | 19     | 12       | 44         |
> | InternVL3-8B             | 34        | 33        | 27        | 34           | 24     | 40          | 19    | 47    | 23     | 43       | 48         |
> | Phi-4                    | 30        | 27        | 27        | 34           | 25     | 40          | 19    | 43    | 25     | 40       | 43         |
> | Idefics3-8B              | 39        | 32        | 39        | 40           | 35     | 35          | 34    | 44    | 26     | 51       | 59         |
> | Mistral-Small-3.1-24B    | 25        | 22        | 29        | 30           | 23     | 40          | 19    | 45    | 20     | 21       | 43         |
> | Aria                     | 45        | 33        | 42        | 37           | 28     | 39          | 30    | 44    | 30     | 47       | 53         |
> | Aya-vision-8b            | 22        | 18        | 21        | 31           | 21     | 40          | 15    | 39    | 17     | 6        | 41         |
> | Gemma-3-27b              | 37        | 41        | 32        | 35           | 28     | 26          | 41    | 42    | 34     | 47       | 49         |
> | GPT-4o                   | 30        | 41        | 30        | 40           | 29     | 46          | 43    | 49    | 43     | 48       | 47         |
> | Gemini-2.5-pro           | 30        | 35        | 25        | 33           | 35     | 38          | 34    | 54    | 34     | 42       | 41         |
>
> To fully address the reviewer's concern, we will add city-wise breakdown tables for other key tasks in the Supplementary Material C.3 and also includes cross-city generalization analysis of different models.

---

> ### Author Response · Authors · 2025-11-22
> **Response to Weakness 2-4 and Question 1, 3**
>
> ## Description of related work (Weakness 2 & Question 1)
>
> We sincerely thank the reviewer for the valuable suggestion regarding the related work section. Visual Chronicles [1] is indeed a state-of-the-art study in street view temporal analysis, offering a novel perspective for analyzing large-scale street view imagery over time.
>
> It is worth noting, however, that the focus of Visual Chronicles differs from our approach in terms of task objectives. Specifically, their primary contribution lies in constructing a large-scale vision analysis framework based on MLLMs to extract and summarize **observable objective changes** in street view imagery, rather than providing a **dedicated benchmark for street view temporal changes**. Moreover, Visual Chronicles emphasizes factual visual content evolution, whereas our study focuses perceptual-level changes in environment. As a result, their research objectives and output modalities do not fully align with ours. Therefore, while acknowledging its relevance, a direct comparison is not entirely appropriate given the differences in task nature and goals.
>
> We have added more relevant research (e.g., Visual Chronicles) in the revised manuscript. Furthermore, considering the core contributions of UrbanFeel and incorporating your suggestions, we have split the related work in Section 2 into two distinct subsections: urban change assessment and subjective perception assessment. We have also further emphasized the unique contributions of this study to subjective perception and cross-temporal street-view understanding. The revised content is marked in blue text in the manuscript.
>
> [1] Deng, Boyang, et al. "Visual Chronicles: Using Multimodal LLMs to Analyze Massive Collections of Images." arXiv preprint arXiv:2504.08727 (2025).
>
> ---
>
> ## Revision of Our Figures (Weakness 3)
>
> We thank the reviewer for the careful observation and agree that clear figures are crucial for understanding the benchmark.
>
> In the revised manuscript, we have increased the font sizes and improved the visual clarity of the relevant elements in these figures. In addition, because the original Figure 2 contained too many elements in a single panel and could be hard to follow at standard resolution, **we have split it into two separate figures (the new Figures 2 and 3 in revised manuscript)**. The first now focuses on the overall structure of UrbanFeel, while the second presents concrete task examples, which we believe makes the task design and illustrative cases easier to read and understand.
>
> ---
>
> ## Claim of Data Acquisition  (Weakness 4 & Question 3)
>
> We sincerely appreciate the reviewer’s constructive suggestions, which are important for ensuring that our study complies with **legal and ethical standards in data usage**.
>
> Regarding data acquisition, as described in Appendix B, the street view imagery used in this study includes panoramic images from **Google Street View** and single-perspective images from **Mapillary**. **Google Street View images were obtained via the official Google Static Street View API**, while **Mapillary images were collected using the Global Streetscapes dataset download scripts and official APIs**. The source APIs provide built-in privacy protections, such as face and license plate blurring. Additionally, during data preprocessing, we removed indoor and low-quality images using segmentation models and manual filtering, ensuring overall compliance.
>
> In terms of licensing and copyright, **Mapillary publicly releases imagery under the CC-BY-SA license, permitting processing and redistribution with appropriate attribution**. Accordingly, all Mapillary-sourced samples in our dataset are strictly labeled to comply with attribution and share-alike requirements. In contrast, **Google Street View imposes stricter restrictions on redistribution**. To avoid potential licensing risks, **our open-source dataset does not directly include raw Google Street View images; instead, it provides only the corresponding street-view IDs**. Dataset users can retrieve the images via the official API, ensuring a legally compliant and traceable data acquisition pathway. We will include and emphasize these licensing and copyright clarifications in the revised manuscript.
>
> Moreover, **Appendix B** systematically details our benchmark construction process, including data acquisition, preprocessing, annotation standards, and evaluation metrics. During dataset release, we will also make publicly available all code used for data processing, including modules for orientation correction, image clustering, and semantic-segmentation-based sample filtering, enabling the research community to fully understand and reproduce our benchmark construction pipeline.
>
> We hope that these responses address the reviewer’s concerns regarding potential risks associated with the dataset as well as questions about its open-sourcing.

---

> ### Author Response · Authors · 2025-11-22
> **Response to Question 2**
>
> ## Failure Case Analysis (Question 2)
>
> We thank the reviewer for this insightful suggestion and agree that a deeper qualitative assessment is needed. In the revised manuscript, we will add a dedicated *Error Analysis* section, drawing on the case studies in Supplementary **Section D** and grouping typical failures into four categories:
>
> 1. **Over-reliance on surface semantics over spatial invariance (Temporal Context).**
>    In temporal-context tasks, models often focus on surface semantics while ignoring spatially invariant geometry. For example, in Figure 15, when an empty lot becomes a residential block, models such as Mistral-Small and Aria incorrectly classify the pair as “Different Locations” solely due to the appearance of new buildings. They fail to exploit stable cues like road layout to recognize that the images depict the same place over time.
>
> 2. **“Linear development” bias and neglect of urban decay (Temporal Reasoning).**
>    For temporal reasoning, models frequently exhibit a “linear development” bias, assuming newer or cleaner infrastructure always corresponds to a later timestamp. In the sorting example in Figure 24, GPT-4o places an image with worn road markings before a pristine one, overlooking alignment cues such as speed bumps and background structures. This suggests a heuristic reliance on visual quality as a proxy for time, leading to errors when urban change is non-monotonic.
>
> 3. **Spatial–perspective misalignment (Static Spatial Perception).**
>    In static spatial perception, models struggle with cross-view mapping, especially between single-view and panoramic images. In the Single-to-Pano Matching task (Figure 9), models like Aria fail to match a single-view crop to its panorama, treating projection-induced distortions as semantic differences rather than perspective changes. This reveals limitations in handling geometric transformations in 360° imagery.
>
> 4. **Hallucinated visual evidence in subjective reasoning (Subjective Perception).**
>    In subjective perception tasks, models sometimes hallucinate negative cues to justify conservative or biased classifications. In Figure 30, for instance, GPT-4o labels a scene as “not wealthy” by citing nonexistent “shuttered storefronts” and “lack of investment,” despite visible well-maintained infrastructure. This indicates a disconnect between the reasoning chain and the actual pixels, where models generate plausible but factually incorrect evidence to support prior impressions.

---

### Author Response · Authors · 2025-11-24
**Global Response to Reviewers**

We sincerely thank all reviewers for their constructive feedback and positive assessment of **UrbanFeel**. We are encouraged that reviewers recognize the **novelty and timeliness** of addressing time-variant and human-centric perception (Reviewers tZ4v, 7xMU, XNMk) , the **comprehensive framework** spanning static, temporal, and subjective dimensions (Reviewers 2E4Z, XNMk) , and the **high-quality curation** of the dataset (Reviewer 2E4Z).

Based on the reviewers' insightful suggestions, we have made substantial revisions to enhance the depth of analysis, methodological rigor, and clarity of the benchmark. **All major changes are marked in blue in the revised manuscript.** The key updates are summarized below:

### **Summary of Updates**

1. **Deepened Geographic Robustness Analysis (Reviewer tZ4v):** We added a detailed city-level performance breakdown in **Appendix C.3 and C.4**. This analysis reveals an intriguing "**inverse geographic bias**," where closed-source models (e.g., Gemini-2.5-Pro) often align better with human perception in Global South cities compared to Global North cities, whereas some open-source models exhibit severe city-specific overfitting.

2. **Reasoning Ablation Study (Reviewers tZ4v, XNMk):** We conducted a controlled ablation study on the Temporal Sequence Reasoning (TSR) task in **Section 5.4 and Appendix C.5**. Our results indicate that for SOTA models (e.g., GPT-4o), explicit reasoning prompts (CoT) can surprisingly degrade performance due to "over-reasoning" hallucinations, a finding that offers important insights for future prompt design.

3. **Geographic Expansion & Causal Reasoning Roadmap (Reviewer 2E4Z):** Acknowledging the geographic imbalance, we outlined a concrete roadmap in **Appendix E (Limitations & Future Work)**. We plan to expand coverage to underrepresented regions (e.g., Lagos, Campina) as multi-year data becomes available and introduce causal annotations to help MLLMs understand the drivers behind urban transformations.

4. **Qualitative Error Analysis (Reviewers tZ4v, XNMk):** We added a dedicated error analysis section in **Appendix D.1**, categorizing model failures into four distinct modes: over-reliance on surface semantics, “linear development” assumption in urban evolution, spatial-perspective misalignment, and hallucinated visual evidence. We also supplemented this with specific case studies to support our conclusions regarding the "linear development" bias.

5. **Enhanced Statistical Rigor (Reviewer 7xMU):** We strengthened the statistical validity of our findings in **Appendix C.6**. By calculating 95% confidence intervals and performing z-tests on key tasks, we statistically confirmed significant gaps between human and model performance in temporal reasoning ($p<0.01$), while observing statistical parity in static global perception.

6. **Clarified Metrics & Reproducibility (Reviewer XNMk):** We explicitly defined the accuracy metrics and the Sentence-BERT similarity threshold ($\tau=0.6$) used for open-ended tasks in **Appendix B.4** to ensure the full reproducibility of our results.

7. **Data Compliance & Distribution Strategy (Reviewers tZ4v, 7xMU):** We detailed our data compliance strategy in **Appendix B.1**. We explicitly clarify that we do not distribute raw Google Street View images (releasing IDs only) and strictly adhere to Mapillary’s CC-BY-SA license requirements.

8. **Task Uniqueness & Literature Review (Reviewers tZ4v, 7xMU):** We expanded the discussion in **Sections 2.1 and 2.2** to better contextualize our work within current urban dynamics research. We highlight that UrbanFeel is a novel benchmark uniquely combining multi-view, multi-temporal, and subjective perception tasks, distinguishing it from prior works such as Place Pulse, CityPulse, and Visual Chronicles.

9. **Improved Visual Presentation (Reviewers tZ4v, XNMk):** We split the original Figure 2 into two separate figures (**Figure 2 and Figure 3** in the revised paper) to improve readability. We also removed ambiguous abbreviations (e.g., "str")  and optimized font sizes and visual clarity across all figures.

10. **Correction of Errors:** We have corrected typo and grammatical errors present in the original manuscript.

In addition to these revisions, we have provided point-by-point responses to each reviewer's specific questions. We once again sincerely thank the reviewers for the time and effort dedicated to improving the quality of UrbanFeel.

---

### Meta-Review · Area_Chair_rZZW · 2026-01-07

**Summary:**

The paper introduces UrbanFeel, a comprehensive benchmark designed to evaluate Multimodal Large Language Models (MLLMs) on their ability to understand urban environments through temporal changes and subjective human perception. This work is quite timely and could potentially very beneficial to the community.

After the rebuttal, the reviewers were generally positive about the paper's novelty, comprehensive structure (static, temporal, and subjective dimensions), and high-quality data curation. The authors provided a really thorough rebuttal that not only clarified misunderstandings but also added substantial new experiments (which led Reviewer XNMk to raise their score to 8).

Besides resolving various concerns from all reviewers, there are few key concerns that have been addressed adequately, for example:
- **Geographic Bias**: Reviewers tZ4v and 2E4Z both mentioned a bias toward "developed" cities, where the authors conducted new experiments with data from Lagos and Campina Grande during the rebuttal which I believe have resolved the concerns.
- **Methodology and License**:  there were comments on the evaluation pipeline (leverage GPT-4o), and it has been resolved effectively. And they will not release directly the raw Google Street View data but only the IDs.

After the thorough check on the reviews and the rebuttals, I believe the paper indeed delivers a useful, robust, and also well-constructed benchmark. I will accept the paper.

**Reviewer Concerns:**

Most of the concerns have been addressed by the rebuttal (e.g., Geographic Bias, evaluation integrity, data licensing, novelty, etc). The causal reasoning concerns from reviewer 2E4Z and the annotator diversity concerns are not resolve, but I don't think these are significant enough for changing the score.

**Reviewer Scores:**

**tZ4v**: Given the comprehensive response to the generalization and licensing concerns, I think this reviewer would likely raise their score from 6 to an 8.

**2E4Z**: by addressing their primary concern about geographic bias, this reviewer would probably increase their score to a 6.

**7xMU**: Since their initial low score was based on factual misunderstandings regarding dataset novelty and evaluation methodology which were fully clarified, I believe they would likely raise their score to a 6.

**XNMk**: This reviewer has mentioned that they explicitly raised their score to an 8

---

### Decision · Program_Chairs · 2026-01-26

Accept (Poster)